

# Low-energy effective description of dark $Sp(4)$ theories

**Suchita Kulkarni**[1*], **Axel Maas**[1†], **Seán Mee**[1‡], **Marco Nikolic**[2∘],
**Josef Pradler**[2§] **and Fabian Zierler**[1△]

**1** Institute of Physics, NAWI Graz, University of Graz,
Universitätsplatz 5, A-8010 Graz, Austria
**2** Institute of High Energy Physics, Austrian Academy of Sciences,
Nikolsdorfergasse 18, 1050 Vienna, Austria

★ suchita.kulkarni@uni-graz.at , † axel.maas@uni-graz.at , ‡ sean.mee@uni-graz.at ,
∘ marco.nikolic@oeaw.ac.at , § josef.pradler@oeaw.ac.at , △ fabian.zierler@uni-graz.at

## Abstract

Strongly interacting massive particles are viable dark matter candidates. We consider a dark $Sp(4)$ gauge theory with $N_f = 2$ fermions in the pseudo-real fundamental representation and construct the chiral low-energy effective theory. We determine the flavour multiplet structure and the chiral Lagrangian, including the Wess-Zumino-Witten term for mass-degenerate and non-degenerate flavours. We then study the possible charge assignments under a $U(1)'$ gauge symmetry, emphasizing on dark state stability, and provide the full Lagrangian description for Goldstone bosons and vector resonances, including the Wess-Zumino-Witten term. Finally, we use dedicated lattice simulations to determine the chiral low-energy effective theory's validity and low-energy constants. This work represents a self-consistent study of this non-Abelian theory. It thereby provides a framework for future phenomenological exploration in connection to the dark matter problem.

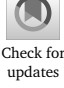

# 1  Introduction

The absence of concrete, non-gravitational experimental signals for dark matter (DM) necessitates the unbroken exploration of a wide variety of scenarios. Among these are extensions of the Standard Model (SM) featuring a dark non-Abelian sector in the ultra-violet (UV) regime

with confinement below some scale $\Lambda$ in the infra-red (IR) regime. Within these scenarios, confinement leads to bound states, containing potentially stable particles and opens up a new class of DM candidates. The stability of bound states in such theories is either ensured by means of introducing additional symmetries or, more naturally, the result of accidental global symmetries of the theory. The composite states can then be considered DM candidates and the viability of the theory can be explored further; for an overview on composite dynamics see [1] and references therein.

The construction and exploration of associated phenomenology of such theories is however far from obvious. While the UV Lagrangian is governed by a microscopic theory, the resulting low-energy Lagrangian in the IR is an effective theory. As such, it has a limited range of validity, and is parameterized by a number of low-energy constants (LECs) [2,3]. The range of validity and the LECs are determined by the underlying theory, and can, e.g., be obtained from lattice simulations [4,5]. In general, the formulation in the IR depends on the UV details such as the gauge group, the particle content and the representations of participating fields. For example, the chiral effective theory in isolation for the mass-degenerate $Sp(4)$ theory, including a determination of its range of validity and LECs, has been analysed on the lattice in [6–8].

It is certainly possible that such theories live in isolation from the SM. Indeed, DM can adjust its abundance in the early Universe free from SM interactions, e.g., by "cannibalizing" through number-changing 3-to-2 or 4-to-2 processes in the dark sector [9–18]. However, couplings to the SM additionally allow to regulate the dark sector temperature in relation to the SM one, and, of course, open the door to experimental exploration. In either case, strongly interacting dark sectors, see, *e.g.* [11,12,19–47] among many other works, are of high interest in such scenarios as their low energy states are often endowed with sizeable interactions. Particular interest have received strongly-interacting massive particles (SIMPs) that are the Goldstone bosons of a dark confining gauge group and where 3-to-2 interactions are enabled by the Wess-Zumino-Witten(WZW) term [11,12,29]. The presence of these $3 \rightarrow 2$ interactions in association with $2 \rightarrow 2$ self-interactions has generated sizeable further activity. For example, [32] explores Abelian gauged SIMPs within $SU(N)$ theories, [48] demonstrates the importance of higher order corrections to the chiral Lagrangian for self-scattering in $Sp(2N)$ theories, [33,35] include dark vector resonances into the SIMP paradigm, while [36,37] have analysed effects of isospin symmetry breaking (dark quark mass non-degeneracy) for dark matter phenomenology. In most of these works, however, either a specific subset of the low-energy effective theory was studied or a SM-like, i. e. $SU(N)$ gauge group and fundamental fermions, dark sector was considered. In addition, the masses and other parameters of the theory were varied independently, instead of deriving them from the underlying UV-complete theory. Only in a few cases contact to the UV theory was made by means of lattice simulations, see [22,30,31,34,49,50].

It is the purpose of this paper to study one particular example of a confining gauge group self-consistently, from the UV-complete Lagrangian to the low-energy description in terms of the chiral Lagrangian. Concretely, we consider a $Sp(2N)$ gauge theory with $N_f = 2$ fermions in the pseudo-real fundamental representation. We allow for mass non-degeneracy of the two flavours, study possible gauge assignments under a new dark $U(1)$ symmetry and obtain the multiplet structure of resulting low energy spectrum of mesons and vector resonances. Importantly, our findings are supported and complemented by lattice simulations. Such unifying effort is a first of its kind in the study of a non-Abelian dark sector. It provides the grounds for a comprehensive and systematically controlled and extensible framework for future phenomenological studies of this theory. $Sp(2N)$ theories as we consider in this work have also been a longstanding topic of interest in the composite Higgs community [1,51–58]. While in isolation these theories are identical to our case, the main difference is the portal phenomenol-

ogy to the Standard Model. While in the case of composite Higgs theories, some of the $Sp(2N)$ quarks transform under the Standard Model gauge group, in our theory, the $Sp(4)$ quarks are singlets under the Standard Model. This leads to different decay patterns and requirements on the theory construction compared to those considered in composite Higgs literature. We also note that in our theory, there is no tree level mixing between $Sp(4)$ spectrum particles and the SM e.g. composite sigma – SM Higgs mixing.

The paper is organized as follows: in Sec. 2 we introduce the global flavour symmetries and clarify the quantum numbers and multiplet structure of states in the low-energy description. In Sec. 3 we systematically develop the chiral Lagrangian and WZW term for the Goldstone bosons, the pseudoscalar singlet and vector resonances for mass-degenerate quarks. In Sec. 4 we couple this theory to an Abelian gauge field, discuss possible charge assignments, study the stability of states against decay into gauge bosons, provide the gauged WZW term and comment on the radiatively induced mass splitting from such interactions. In Sec. 5 we allow for mass non-degeneracy, and determine how the spectrum and interactions change. In Sec. 6 we extend previous lattice simulations to cover the mass non-degenerate case, providing concrete results for bound state mass spectra and decay constants; preliminary results on this effort have been reported in [59]. We infer the range of validity of the chiral effective theory as a function of the mass splitting, delineating where the theory changes its character to a hierarchical "heavy-light" system. In addition, we determine the relevant LEC for the coupling to the $U(1)'$ gauge boson from lattice results as a function of the mediator sector. We conclude in Sec. 7 and compile in four appendices a host of technical details, including pertinent Feynman rules, useful expressions for operators and generators and states, and details of the lattice simulations including studies of the systematics.

## 2 Symmetry breaking patterns

### 2.1 Lagrangian and global symmetries

The construction of the low energy effective theory starts by recognizing the symmetries of the underlying microscopic theory in the ultraviolet (UV). In this chapter we first discuss the symmetries of the UV Lagrangian for massless fermions. We then consider spontaneous chiral symmetry breaking by the fermion condensate and show that this results in the same global symmetry as an explicit breaking induced by *degenerate* fermion masses. Finally, we show the resulting breaking patterns that correspond to *non-degenerate* fermion masses in the UV Lagrangian or fermions under different charge assignments when coupled to a $U(1)$ gauge group.

The UV Lagrangian of a $Sp(4)_c$ gauge theory — where the subscript $c$ for "colour" on the group is used to highlight its gauge nature — with $N_f = 2$ fundamental Dirac fermions $u$ and $d$ is given by,

$$\mathcal{L}^{\text{UV}} = -\frac{1}{2}\text{Tr}\left[G_{\mu\nu}G^{\mu\nu}\right] + \bar{u}\left(\gamma_\mu D_\mu + m_u\right)u + \bar{d}\left(\gamma_\mu D_\mu + m_d\right)d\,, \tag{1}$$

where spinor, and colour indices are suppressed. We will call $u$ and $d$ (dark) "quarks" in analogy to QCD, even if they are singlets under the SM gauge group; $D_\mu = \partial_\mu + igA_\mu$ is the non-Abelian covariant derivative with gauge coupling $g$ and associated (dark) "gluon" fields $A_\mu = A_\mu^a\tau^a$. We denote the generators of $Sp(4)_c$ by $\tau^a$ and the generators of a global $Sp(4)$ group by $T^a$. The generators of $Sp(4)$ are explicitly given in App. A. The Yang-Mills field strength tensor is given by $G_{\mu\nu} = \partial_\mu A_\nu - \partial_\nu A_\mu + ig\left[A_\mu, A_\nu\right]$ as usual.

The fermions in (1) are in a pseudo-real representation. Compared to a theory where fermions are in the complex representation such as in QCD, the global symmetry of the La-

grangian (1) is enlarged to a so-called Pauli-Gürsey symmetry [60,61]. Following [60], this can be made explicit, by first introducing the left-handed ($L$) and right-handed ($R$) chiral Weyl components of the Dirac spinors $u$ and $d$,

$$u = \begin{pmatrix} u_L \\ u_R \end{pmatrix}, \qquad d = \begin{pmatrix} d_L \\ d_R \end{pmatrix}. \tag{2}$$

One may subsequently group the left- and right-handed components,

$$\psi_L = \begin{pmatrix} u_L \\ d_L \end{pmatrix}, \qquad \psi_R = \begin{pmatrix} u_R \\ d_R \end{pmatrix}. \tag{3}$$

Using the chiral representation of the Dirac gamma matrices, we may now rewrite the fermionic kinetic term of the Lagrangian as

$$\mathcal{L}_{\text{fermionic}} = i \begin{pmatrix} \psi_L^* \\ \psi_R^* \end{pmatrix}^T \begin{pmatrix} \bar{\sigma}_\mu D^\mu & 0 \\ 0 & \sigma_\mu D^\mu \end{pmatrix} \begin{pmatrix} \psi_L \\ \psi_R \end{pmatrix}. \tag{4}$$

For this we have introduced the four-component notation $\sigma_\mu = (1, \vec{\sigma})$ and $\bar{\sigma} = (1, -\vec{\sigma})$ where $\sigma_j$ are the usual Pauli matrices. We may now use the pseudo-reality condition of the colour group, i.e., the existence of a colour matrix $S$ for which the relation $(\tau^a)^T = S\tau^a S$ holds for all generators $\tau^a$, as well as the relation $\sigma_2 \sigma_\mu \sigma_2 = \bar{\sigma}_\mu^T$ and rewrite the fermionic kinetic term as

$$\mathcal{L}_{\text{kin}}^{\text{UV,f}} = i \begin{pmatrix} \psi_L^* \\ \sigma_2 S \psi_R \end{pmatrix}^T \begin{pmatrix} \bar{\sigma}_\mu D^\mu & 0 \\ 0 & \bar{\sigma}_\mu D^\mu \end{pmatrix} \begin{pmatrix} \psi_L \\ \sigma_2 S \psi_R^* \end{pmatrix} = i \Psi^\dagger \bar{\sigma}_\mu D^\mu \Psi. \tag{5}$$

In the last equality we introduced the notation where $\Psi$ is a vector consisting of the four Weyl fermions in this theory, which is known as the Nambu-Gorkov formalism,

$$\Psi \equiv \begin{pmatrix} \psi_L \\ \tilde{\psi}_R \end{pmatrix} = \begin{pmatrix} u_L \\ d_L \\ \sigma_2 S u_R^* \\ \sigma_2 S d_R^* \end{pmatrix}, \tag{6}$$

from where one may appreciate a global $U(4)$ symmetry of the kinetic term. We observe that $\tilde{\psi}_R \equiv \sigma_2 S \psi_R^*$ transforms under the same representation of this global symmetry as $\psi_L$. In case of massless fermions this is the symmetry of the entire fermionic Lagrangian. The $U(4)$ symmetry is then broken to $SU(4)$ by the axial anomaly analogously as to in QCD. For a theory with $N_f$ fermions the global flavour symmetry becomes $SU(2N_f)$.

In the massless fermion limit the remaining $SU(4)$ symmetry is subsequently broken spontaneously by the chiral condensate $\langle \bar{u}u + \bar{d}d \rangle \neq 0$. This can be seen by rewriting the chiral condensate in terms of the generalized vector of spinors $\Psi$ as we did for the kinetic term of the Lagrangian [60],

$$\bar{u}u + \bar{d}d = -\frac{1}{2} \Psi^T \sigma_2 S E \Psi + \text{h.c.}, \tag{7}$$

$$E = \begin{pmatrix} 0 & \mathbb{1}_{N_f} \\ -\mathbb{1}_{N_f} & 0 \end{pmatrix}. \tag{8}$$

Here, $E$ is a matrix in flavour space. Under a global $SU(4)$ transformation $\Psi \to U\Psi$ this expression transforms as

$$-\frac{1}{2} \Psi^T \sigma_2 S E \Psi \to -\frac{1}{2} \Psi^T U^T \sigma_2 S E U \Psi, \tag{9}$$

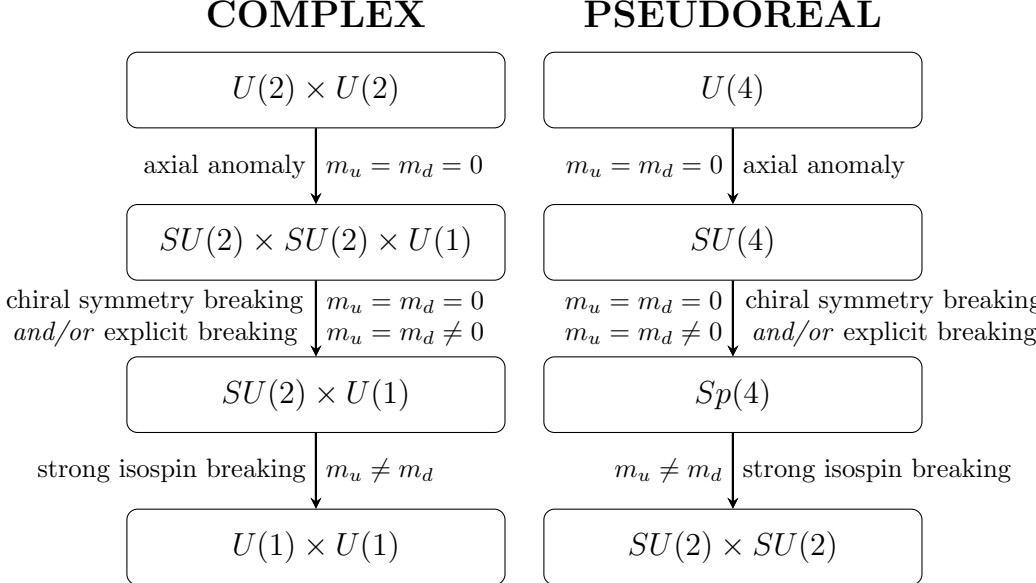

Figure 1: Comparison of breaking patterns in a theory with $N_f = 2$ fermions in pseudo-real (right) and complex (left) representation. In any step, the symmetry for pseudo-real representations is larger than for complex ones.

The condensate is thus invariant under all $SU(4)$ transformations $U$ which fulfil $U^T E U = E$. This is exactly the $Sp(4)$ subgroup of the global $SU(4)$. The matrix $E$ is an invariant tensor of $Sp(4)$ and the flavour space-equivalent of the colour matrix $S$. By counting the generators of $SU(4)$ and $Sp(4)$ we thereby obtain $15 - 10 = 5$ Goldstone bosons in this theory. In turn, for general $N_f$, the remaining symmetry group after chiral symmetry breaking is $Sp(2N_f)$.

*Degenerate* UV masses can be introduced by the term $m(\bar{u}u + \bar{d}d)$. They yield an additional, explicit breaking of the flavour symmetry from $SU(4)$ to $Sp(4)$. For a general *non-degenerate* mass term we may write

$$m_u \bar{u}u + m_d \bar{d}d = -\frac{1}{2}\Psi^T \sigma_2 M E \Psi + \text{h.c.} = -\frac{1}{2}\Psi^T \sigma_2 \begin{pmatrix} 0 & 0 & m_u & 0 \\ 0 & 0 & 0 & m_d \\ -m_u & 0 & 0 & 0 \\ 0 & -m_d & 0 & 0 \end{pmatrix} \Psi + \text{h.c.} \quad (10)$$

This leaves only the $N_f = 1$, $Sp(2) = SU(2)$ symmetry for each flavour separately,[1] and thus a global $Sp(2) \times Sp(2)$ flavour symmetry. The breaking patterns are summarized in Fig. 1. The mass matrix $M = \text{diag}(m_u, m_d, m_u, m_d)$ containing the non-degenerate dark quark masses can then be read from (10),

$$ME = \begin{pmatrix} 0 & 0 & m_u & 0 \\ 0 & 0 & 0 & m_d \\ -m_u & 0 & 0 & 0 \\ 0 & -m_d & 0 & 0 \end{pmatrix},$$

where $E$ is given in Eq. (8).

Another source of explicit symmetry breaking are gauge couplings to an external $U(1)$ vector field $V^\mu$. Depending on the charge assignments of the fermions, this yields distinct

---

[1] The actual group is $U(2) \times U(2)$, but one $U(1)$ is broken by the axial anomaly, and the other $U(1)$ is just fermion number conservation. The global symmetry of a theory of $N_1$ fermions of mass $m_{N_1}$ and $N_2$ fermions of mass $m_{N_2}$ is $Sp(2N_1) \times Sp(2N_2)$. See e.g. [62] for a dark matter model with a $Sp(4) \times Sp(2)$ symmetry of the fermionic mass terms.

explicit symmetry breaking patterns parameterized by a matrix $\mathcal{Q}$ in the Weyl flavour space,

$$\mathcal{L}_{\text{break}} \sim V^\mu \Psi^\dagger \mathcal{Q} \partial_\mu \Psi \,. \tag{11}$$

For example, a breaking pattern $Sp(4) \to SU(2) \times U(1)$ can be realized, assuming that $\mathcal{Q}$ is diagonal. The choices of $\mathcal{Q}$ which preserve this nontrivial flavour subgroup are those for which $\mathcal{Q}^2 \propto \mathbb{1}$. This is incidentally the same condition usually proposed for anomaly cancellation [35]. We show in what follows that this is no accident, and that the nontrivial remaining flavour group ensures the anomalous vertex responsible for pion decay to vanish to all orders (in contrast to SM QCD.) We can also realize a further breakdown to a $Sp(4) \to U(1) \times U(1)$, which has some significant implications for the stability of composite states of the theory, allowing some of them to decay. An in-depth discussion of this is deferred to Sec. 4.

## 2.2 Symmetries of the low energy mesonic spectrum

We now turn to the symmetries of the low energy mesonic spectrum for *degenerate* and *non-degenerate* dark quarks. As can be seen in Fig. 1 the flavour symmetry of the $Sp(4)_c$ gauge theory is enlarged compared to a theory with a complex fermion representation such as QCD due to the pseudo-reality of the fundamental representation of $Sp(4)$. This entails that also the meson multiplets are enlarged in both the *degenerate* and *non-degenerate* case. There are now five instead of three Goldstone bosons of two flavour QCD. We denote the Goldstone bosons in this theory by $\pi^{A,\dots,E}$.

The extra states are quark-quark and antiquark-antiquark states. The extra operators corresponding to the Goldstone bosons are given by [20, 22]

$$\pi^D = \bar{d}\gamma_5 SC\bar{u}^T \,, \tag{12}$$

$$\pi^E = d^T SC\gamma_5 u \,. \tag{13}$$

For convenience, we will call these states "diquark" states. In this case, only diquarks of differing flavour are possible; other operators of this form vanish identically for spin-0 composite states. They can occur, however, for other spin states such as $J = 1$ [8].

The multiplet structure of mesons for *mass-degenerate* fermions without coupling to any other symmetry is discussed in [6–8] as well as in studies of $SU(2)_c$, which has the same flavour symmetry as $Sp(4)_c$, with two fundamental fermions [63]. Besides the 5 Goldstone bosons there is another pseudoscalar singlet under $Sp(4)$ which is the analogue of the $\eta'$ meson of QCD. In turn, the multiplet of the vector mesons is enlarged to a 10-plet (whose states we call $\rho^{F,\dots,O}$) which includes the state that sources the analogue of the $\omega$ meson of QCD [8]. Due to the structure of $Sp(4)$, mesons are either in a 10-plet, 5-plet or a singlet representation of the global $Sp(4)$ flavour symmetry [63]. In appendix B.1 we give explicitly the operators that source the $\pi$ and $\rho$ mesons.

In case of *non-degenerate fermions*, the $Sp(4)$ flavour symmetry is broken to $SU(2)_u \times SU(2)_d$. It can be shown that the 5-plet splits into a degenerate 4-plet[2] and a singlet under $SU(2)_u \times SU(2)_d$. The 10-plet containing the $\rho$-mesons decomposes into a 4-plet of similar structure as the 4-plet of Goldstones and two degenerate triplets under each of the $SU(2)_{u/d}$ groups, *i.e.*, the remaining 6 states will have identical properties. The 4-plets contain in both cases the "open-flavour" mesons, *i.e.*, two meson operators of the form $\bar{u}\Gamma d$ – where $\Gamma$ is either $\gamma_5$ for the Goldstones or $\gamma_\mu$ for the $\rho$-mesons – and two corresponding diquark states. In appendix B.2 we state the explicit transformations of the operators under $SU(2)_u \times SU(2)_d$.

---

[2]Note that $SU(2)_u \times SU(2)_d \sim O(4)$, and thus the 4-plet can be considered to be in the 4-dimensional fundamental vector representation of $O(4)$. This 4-plet can be decomposed into its two $SU(2)$ representations explicitly in a suitable two-dimensional tensor representation [64].

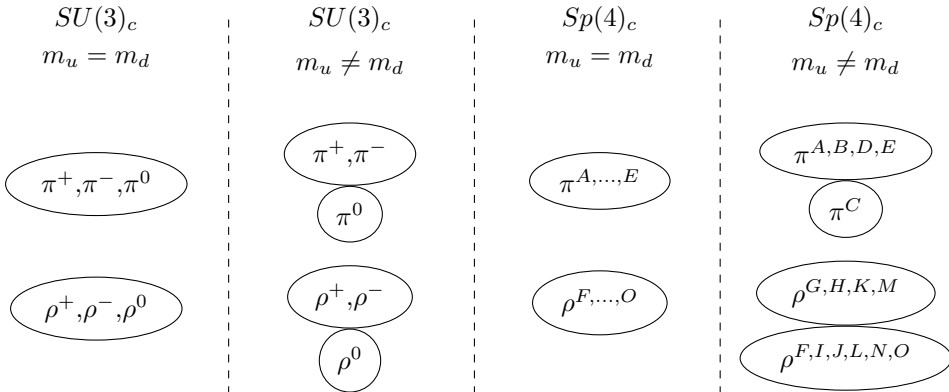

Figure 2: The flavour structure of the meson multiplets for an $Sp(4)_c$ gauge theory with $N_f = 2$, compared to QCD with the same number of quarks. The multiplets of $Sp(4)_c$ are enlarged due to additional diquark states. When the flavour symmetry is explicitly broken by non-degenerate fermion masses, $m_u \neq m_d$, the pseudoscalar and vector multiplets split further into smaller flavour-multiplets under the flavour group. The unflavoured $\pi^C$ becomes a singlet.

The six other spin-1 states are given by linear combinations of $\bar{u}\Gamma u$, $\bar{d}\Gamma d$ and corresponding diquark operators. They can be written in the eigenbases of the two $SU(2)_{u/d}$.

Figure 2 groups mesons into degenerate and non-degenerate sets and compares them to QCD with two degenerate fundamental fermions. In the case of different UV fermion masses $m_u \neq m_d$, $\pi^C$ becomes a singlet under the global flavour symmetry, whereas before it was part of a multiplet. This has consequences for its viability as a dark matter candidate: it is no longer protected by a flavour symmetry and, equipped with further interactions, may in principle decay. Finally, as already alluded to above, coupling the strongly interacting dark sector to a $U(1)$ gauge symmetry can further break the remaining flavour symmetry and affect the multiplet structure of the mesons. This will be addressed in Sec. 4.

## 2.3 Parity and diquarks

Conventionally, the transformation of a Dirac fermion under parity is defined as [65]

$$P : \psi(\mathbf{x}, t) \to \gamma_0 \psi(-\mathbf{x}, t). \tag{14}$$

This implies that parity mixes left-handed and right-handed components. This can be made explicit by going to the chiral representation of the $\gamma$-matrices,

$$P : \begin{aligned} \psi_L(\mathbf{x}, t) &\to \psi_R(-\mathbf{x}, t), \\ \psi_R(\mathbf{x}, t) &\to \psi_L(-\mathbf{x}, t). \end{aligned} \tag{15}$$

This discrete transformation, which we shall refer to as "ordinary parity" leaves the Lagrangian invariant and is thus a symmetry. However, it is important to note that the global flavour symmetry mixes left-handed and right-handed components, and such transformation does not in general commute with the parity transformation law given above. In other words, the flavour eigenstates are not eigenstates of $P$. This has the consequence that a diquark Goldstone state is rather a scalar than a pseudoscalar under $P$:

$$\pi^E(\mathbf{x}, t) = d^T(\mathbf{x}, t) SC\gamma_5 u(\mathbf{x}, t) \xrightarrow{P} d^T(-\mathbf{x}, t) \gamma_0^T SC\gamma_5 \gamma^0 u(-\mathbf{x}, t)$$
$$= d^T(-\mathbf{x}, t) SC\gamma_5 u(-\mathbf{x}, t) = +\pi^E(-\mathbf{x}, t). \tag{16}$$

Table 1: Parity assignments of the quark-antiquark bound-states and the additional diquark states. Under ordinary $P$-parity different parity-eigenstates occur in the same meson multiplet. Only under $D$-parity all Goldstones can be classified as pseudoscalars and all particles in the $\rho$-multiplets as vectors. An extension of this table is given in App. B.1.

|  | $J^P$ | $J^D$ |
|---|---|---|
| $\pi^A, \pi^B, \pi^C$ | $0^-$ | $0^-$ |
| $\pi^D, \pi^E$ | $0^+$ | $0^-$ |
| $\rho^H, \rho^M, \rho^N, \rho^O$ | $1^-$ | $1^-$ |
| $\rho^F, \rho^G, \rho^I, \ldots, \rho^L$ | $1^+$ | $1^-$ |

The reason for it is the occurrence of the charge conjugation matrix $C$ in the diquark states for which $\gamma_0^T C \gamma_0 = -C$ holds. Concretely, we find that the multiplet of the Goldstones bosons consists of 3 pseudoscalar mesons and 2 scalar diquarks and the multiplet containing the $\rho$ is made up of 4 vectors and 6 axialvectors under $P$; these states may then change their ordinary parity under flavour transformations.

We are, however, free to obtain a better definition of parity by combining it with any other internal symmetry present in our Lagrangian, see e.g. [66]. Introducing an additional phase in the transformation properties of the spinors (14), we may choose a new parity $D$ that now commutes with all flavour transformations [63]. It is given by

$$D : \psi(\mathbf{x}, t) \to \pm i P \psi(\mathbf{x}, t) P^{-1} = \pm i \gamma_0 \psi(-x, t). \tag{17}$$

The extra phase cancels in all operators of the form $\bar{u}\Gamma d$ but produces an extra minus sign in the diquark operators. The new parity $D$ is again a symmetry of the Lagrangian introduced below and all members of a meson multiplet share the same parity assignment under $D$. In this way, all Goldstones become pseudoscalars and all members of the $\rho$-multiplet become vectors under $D$; see Tab. 1 for an overview of $P$- and $D$-parity of the mesons considered in this work. For instance, the diquark $\pi^E$ transforms under the parity assignment $D$ as follows,

$$\pi^E(\mathbf{x}, t) = d^T(\mathbf{x}, t) S C \gamma_5 u(\mathbf{x}, t) \xrightarrow{D} i^2 d^T(-\mathbf{x}, t) \gamma_0^T S C \gamma_5 \gamma^0 u(-\mathbf{x}, t)$$
$$= -d^T(-\mathbf{x}, t) S C \gamma_5 u(-\mathbf{x}, t) = -\pi^E(-\mathbf{x}, t). \tag{18}$$

## 3 Chiral Lagrangian for degenerate fermion masses

In this section, we assume that the two fermion states $u, d$ carry degenerate masses $m_u = m_d = m$ in the UV. First, we analyse the mass spectrum of Goldstone bosons in isolation from additional, external interactions. Second, we construct the chiral Lagrangian for vector and axial-vector excitations of the theory.

### 3.1 Low energy effective theory for the Goldstone bosons

In this section, we recap the construction of the chiral Lagrangian for the (pseudo-Nambu-) Goldstone bosons (PNGB), the "pions" of the theory. The final results agree with the literature [11]. Following [60], the effective theory of Goldstone fields can be written as fluctuations of the orientation of the chiral condensate $\Sigma$,

$$\Sigma = e^{i\pi/f_\pi} \Sigma_0 e^{i\pi^T/f_\pi}, \tag{19}$$

which, under the action of $U \in \mathrm{SU}(4)$, transforms as

$$\Sigma \to U\Sigma U^T \, . \tag{20}$$

Here, $f_\pi$ is the Goldstone decay constant and $\Sigma_0$ is the orientation of the chiral condensate. The latter depends on the remaining symmetry group after spontaneous or explicit symmetry breaking. The Goldstone bosons are determined by the broken generators of the coset space $SU(2N_f)/Sp(2N_f)$. For our case of interest, with $N_f = 2$ flavours in the pseudo-real representation of the gauge group $Sp(4)_c$, there are five Goldstone bosons $\pi^{1,\dots,5}$ corresponding to the broken generators $T^n$ ($n = 1, \dots, 5$) of the coset space $SU(4)/Sp(4)$ given by (A.6); the change of basis to $\pi^{A,\dots,E}$ used above is given towards the end of this subsection.

The Goldstone boson matrix takes the form,

$$\pi \equiv \sum_{n=1}^5 \pi_n T^n = \frac{1}{2\sqrt{2}} \begin{pmatrix} \pi_3 & \pi_1 - i\pi_2 & 0 & \pi_5 - i\pi_4 \\ \pi_1 + i\pi_2 & -\pi_3 & -\pi_5 + i\pi_4 & 0 \\ 0 & -\pi_5 - i\pi_4 & \pi_3 & \pi_1 + i\pi_2 \\ \pi_5 + i\pi_4 & 0 & \pi_1 - i\pi_2 & -\pi_3 \end{pmatrix} . \tag{21}$$

The effective Lagrangian needs to preserve the reality condition, Lorentz invariance, chiral symmetry invariance (with vanishing UV masses), as well as parity and charge conjugation symmetry. This yields[3]

$$\mathcal{L} = \frac{f_\pi^2}{4} \mathrm{Tr}\left[\partial_\mu \Sigma \partial^\mu \Sigma^\dagger\right] - \frac{\mu^3}{2}\left(\mathrm{Tr}[M\Sigma] + \mathrm{Tr}\left[\Sigma^\dagger M^\dagger\right]\right) + \dots \, . \tag{22}$$

Here, $\mu$ has mass dimension one and is related to the chiral condensate as $\langle \bar{u}u + \bar{d}d \rangle = \mu^3 \Sigma_0$; the prefactor $f_\pi^2/4$ in front of the first term ensures a canonically normalized kinetic term.[4] The ellipses stand for contributions, such as, e.g., $\mathrm{Tr}\left[\partial_\mu \Sigma \partial^\mu \Sigma^\dagger\right]^2$ with prefactors of accordingly higher mass-dimension.

In order to find the vacuum alignment $\Sigma_0$, we have to minimize the potential of the chiral Lagrangian, which in our case amounts to $\min_{\pi=0}\left(\mathrm{Tr}\left[M\Sigma + \Sigma^\dagger M^\dagger\right]\right)$. The minimum is then given by $\Sigma_0 = E \in Sp(4)$. Expanding the chiral field (19) in the "kinetic" and "mass" terms of the chiral Lagrangian in terms of Goldstone fields $\pi$ yields the ordinary kinetic, mass, and interaction terms of even numbers of the Goldstone bosons,

$$\mathcal{L}_{\mathrm{kin}} = \mathrm{Tr}\left[\partial_\mu \pi \partial^\mu \pi\right] - \frac{2}{3f_\pi^2}\mathrm{Tr}\left[\pi^2 \partial^\mu \pi \partial_\mu \pi - \pi \partial^\mu \pi \pi \partial_\mu \pi\right] + \mathcal{O}\left(\frac{\pi^6}{f_\pi^4}\right)$$

$$= \frac{1}{2}\sum_{n=1}^5 \partial_\mu \pi_n \partial^\mu \pi_n - \frac{1}{12f_\pi^2}\sum_{k,n=1}^5\left(\pi_k \pi_k \partial_\mu \pi_n \partial^\mu \pi_n - \pi_k \partial_\mu \pi_k \pi_n \partial^\mu \pi_n\right) + \mathcal{O}\left(\frac{\pi^6}{f_\pi^4}\right), \tag{23}$$

$$\mathcal{L}_{\mathrm{mass}} = -m_\pi^2 \mathrm{Tr}\left[\pi^2\right] + \frac{m_\pi^2}{3f_\pi^2}\mathrm{Tr}\left[\pi^4\right] + \mathcal{O}\left(\frac{\pi^6}{f_\pi^6}\right) = -\frac{m_\pi^2}{2}\sum_{k=1}^5 \pi_k^2 + \frac{m_\pi^2}{48f_\pi^2}\left(\sum_{k=1}^5 \pi_k^2\right)^2 + \mathcal{O}\left(\frac{\pi^6}{f_\pi^6}\right). \tag{24}$$

The universal Goldstone mass is given by $m_\pi^2 = 2\mu^3 m/f_\pi^2$. By construction, only interactions of an even number of Goldstone bosons are present; $\mathrm{Tr}[\pi^n] = 0$ for $n$ odd. We note in passing

---

[3]The mass term may be derived by treating $M$ as a spurion field, as the global flavour symmetry should also be manifest in the effective theory. $M$ has to transform as $M \to U^* M U^\dagger$ with $U \in SU(4)$ and $\Psi \to U\Psi$ in order to ensure full chiral/flavour symmetry in (10), see [60] for more details. This leads to the chiral-invariant mass term at lowest order in $M$ in (22) respecting the transformation of $\Sigma$ in (20).

[4]The prefactor depends on the normalization of the generators as they enter exponential of (19). For instance, $f_\pi^{\mathrm{Ref.\,[11]}} = 2f_\pi$ where $f_\pi^{\mathrm{Ref.\,[11]}}$ is the decay constant used in [11] and differs from ours by a factor 2.

that the four-point interactions of same-flavour Goldstone bosons only emerge from the "mass term" (24).

A five-point interaction is induced by the Wess-Zumino-Witten (WZW) action [67–69]. The latter is written as an integral over the five-dimensional disc $Q_5$,

$$
\begin{aligned}
\mathcal{S}_{\text{WZW}} &= \frac{-iN_c}{240\pi^2} \int_{Q_5} \text{Tr}\left[(\Sigma^\dagger d\Sigma)^5\right] = \frac{-iN_c}{240\pi^2} \int_{Q_5} \text{Tr}\left[\left(\Sigma^\dagger \frac{\partial\Sigma}{\partial x} dx\right)^5\right] \\
&= \frac{-iN_c}{240\pi^2} \int_{Q_5} dx^\mu dx^\nu dx^\rho dx^\sigma dx^\lambda \text{Tr}\left[\Sigma^\dagger \partial_\mu \Sigma \Sigma^\dagger \partial_\nu \Sigma \Sigma^\dagger \partial_\rho \Sigma \Sigma^\dagger \partial_\sigma \Sigma \Sigma^\dagger \partial_\lambda \Sigma\right],
\end{aligned}
\tag{25}
$$

where $N_c$ is the number of colours. Expanding in terms of Goldstone fields and using Stokes' theorem as well as $dx^\mu dx^\nu dx^\rho dx^\sigma = dx^4 \epsilon^{\mu\nu\rho\sigma}$ makes the five-point interaction evident in the Lagrangian density,

$$
\begin{aligned}
\mathcal{L}_{\text{WZW}} &= \frac{2N_c}{15\pi^2 f_\pi^5} \epsilon^{\mu\nu\rho\sigma} \text{Tr}\left[\pi\partial_\mu\pi\partial_\nu\pi\partial_\rho\pi\partial_\sigma\pi\right] + \mathcal{O}\left(\pi^6/f_\pi^6\right) \\
&= \frac{2N_c}{15\pi^2 f_\pi^5} \epsilon^{\mu\nu\rho\sigma} \pi_i \partial_\mu\pi_j \partial_\nu\pi_k \partial_\rho\pi_l \partial_\sigma\pi_n \text{Tr}\left[T^i T^j T^k T^l T^n\right] + \mathcal{O}\left(\pi^6/f_\pi^6\right) \\
&= \frac{N_c}{10\sqrt{2}\pi^2 f_\pi^5} \epsilon^{\mu\nu\rho\sigma} \big(\pi_1\partial_\mu\pi_2\partial_\nu\pi_3\partial_\rho\pi_4\partial_\sigma\pi_5 - \pi_2\partial_\mu\pi_1\partial_\nu\pi_3\partial_\rho\pi_4\partial_\sigma\pi_5 \pi_3\partial_\mu\pi_1\partial_\nu\pi_2\partial_\rho\pi_4\partial_\sigma\pi_5 \\
&\quad + \pi_4\partial_\mu\pi_1\partial_\nu\pi_2\partial_\rho\pi_5\partial_\sigma\pi_3 + \pi_5\partial_\mu\pi_1\partial_\nu\pi_2\partial_\rho\pi_3\partial_\sigma\pi_4\big) + \mathcal{O}\left(\pi^6/f_\pi^6\right).
\end{aligned}
\tag{27}
$$

As can be seen in the final form, the WZW term yields purely flavour off-diagonal interactions. Moreover, the WZW conserves both $P$-parity and $D$-parity, since spatial derivatives change sign under these parity transformations.

The full chiral Lagrangian is then given by the sum of kinetic, mass, and WZW terms,

$$
\mathcal{L} = \mathcal{L}_{\text{kin}} + \mathcal{L}_{\text{mass}} + \mathcal{L}_{\text{WZW}}.
\tag{28}
$$

We point out that four-point and five-point interactions between Goldstone bosons remain present in the massless limit $M \to 0$. The Feynman rules are given in the Appendix C. We note that the expressions here are connected to the basis in section 2 labelled by capital Latin letters $A, B, \ldots$, as

$$
\begin{pmatrix} \pi^A \\ \pi^B \\ \pi^C \\ \pi^D \\ \pi^E \end{pmatrix} = \frac{1}{\sqrt{2}} \begin{pmatrix} 1 & i & 0 & 0 & 0 \\ 1 & -i & 0 & 0 & 0 \\ 0 & 0 & \sqrt{2} & 0 & 0 \\ 0 & 0 & 0 & i & 1 \\ 0 & 0 & 0 & -i & 1 \end{pmatrix} \cdot \begin{pmatrix} \pi_1 \\ \pi_2 \\ \pi_3 \\ \pi_4 \\ \pi_5 \end{pmatrix}.
\tag{29}
$$

This defines $\pi^A = \pi^{B\dagger}$ and $\pi^D = \pi^{E\dagger}$, and is useful once U(1) charges are assigned in section 4. The generators of the new basis are given in (A.18).[5] Also, in Sec. 5 we describe the consequences of introducing a mass splitting between the fermions in the UV. It leads to different couplings among the Goldstone bosons and to a splitting of the low-energy mass spectrum.

---

[5]A similar relation is used in QCD with two flavours $u, d$ where it relates the Goldstone bosons defined through the Pauli matrices of the flavour symmetry $\sigma_i$ by $\pi_i = \bar{\Psi}\gamma_5\sigma_i\Psi$ ($i = 1, 2, 3$) with $\Psi = (u, d)^T$ to the commonly used pions $\pi^\pm = \bar{\Psi}\gamma_5\sigma^\pm\Psi$ and $\pi^0 = \bar{\Psi}\gamma_5\sigma^0\Psi$ as

$$
\begin{pmatrix} \pi^+ \\ \pi^- \\ \pi^0 \end{pmatrix} = \frac{1}{\sqrt{2}} \begin{pmatrix} 1 & i & 0 \\ 1 & -i & 0 \\ 0 & 0 & \sqrt{2} \end{pmatrix} \cdot \begin{pmatrix} \pi_1 \\ \pi_2 \\ \pi_3 \end{pmatrix} \quad (\text{QCD}).
\tag{30}
$$

There, we shall see that one Goldstone occupies the lightest mass state, which can be identified with the state $\pi_3 = \pi^C$, making its special role evident. Explicitly, the expanded chiral Lagrangian in the new basis reads

$$
\begin{aligned}
\mathcal{L} =&\, \partial_\mu \pi^A \partial^\mu \pi^B + \frac{1}{2} \partial_\mu \pi^C \partial^\mu \pi^C + \partial_\mu \pi^D \partial^\mu \pi^E - m_\pi^2 \left( \pi^A \pi^B + \frac{1}{2} \pi^C \pi^C + \pi^D \pi^E \right) \\
&+ \frac{m_\pi^2}{12 f_\pi^2} \left( \pi^A \pi^B + \frac{1}{2} \pi^C \pi^C + \pi^D \pi^E \right)^2 \\
&- \frac{1}{12 f_\pi^2} \Bigg[ 4 \left( \pi^A \pi^B + \frac{1}{2} \pi_C^2 + \pi^D \pi^E \right) \left( \partial_\mu \pi^A \partial^\mu \pi^B + \frac{1}{2} \partial_\mu \pi^C \partial^\mu \pi^C + \partial_\mu \pi^D \partial^\mu \pi^E \right) \\
&- \left( \pi^A \partial^\mu \pi^B + \pi^B \partial^\mu \pi^A + \pi^C \partial^\mu \pi^C + \pi^D \partial^\mu \pi^E + \pi^E \partial^\mu \pi^D \right)^2 \Bigg] \\
&+ \frac{16 N_c}{5 \sqrt{2} \pi^2 f_\pi^5} \epsilon^{\mu\nu\rho\sigma} \Bigg[ 2 \left( \pi^A \partial_\mu \pi^B - \pi^B \partial_\mu \pi^A \right) \partial_\nu \pi^C \partial_\rho \pi^D \partial_\sigma \pi^E \\
&+ 2 \left( \pi^D \partial_\mu \pi^E - \pi^E \partial_\mu \pi^D \right) \partial_\nu \pi^C \partial_\rho \pi^A \partial_\sigma \pi^B + \pi^C \partial_\mu \pi^A \partial_\nu \pi^B \partial_\rho \pi^D \partial_\sigma \pi^E \Bigg] + \mathcal{O}\left( \pi^6 \right),
\end{aligned}
\tag{31}
$$

where we used the relations

$$
\frac{1}{2} \sum_{n=1}^5 \pi_n^2 = \pi^A \pi^B + \frac{1}{2} \pi_C^2 + \pi^D \pi^E,
$$

$$
\frac{1}{2} \sum_{n=1}^5 \partial_\mu \pi_n \partial^\mu \pi_n = \partial_\mu \pi^A \partial^\mu \pi^B + \frac{1}{2} \partial_\mu \pi^C \partial^\mu \pi^C + \partial_\mu \pi^D \partial^\mu \pi^E,
$$

$$
\sum_{n=1}^5 \pi_n \partial_\mu \pi_n = \pi^A \partial^\mu \pi^B + \pi^B \partial^\mu \pi^A + \pi^C \partial^\mu \pi^C + \pi^D \partial^\mu \pi^E + \pi^E \partial^\mu \pi^D, \tag{32}
$$

$$
\begin{aligned}
\frac{4\sqrt{2}}{3} \mathrm{Tr}\left[ \pi \partial_\mu \pi \partial_\nu \pi \partial_\rho \pi \partial_\sigma \pi \right] =&\, 2 \left( \pi^A \partial_\mu \pi^B - \pi^B \partial_\mu \pi^A \right) \partial_\nu \pi^C \partial_\rho \pi^D \partial_\sigma \pi^E \\
&+ 2 \left( \pi^D \partial_\mu \pi^E - \pi^E \partial_\mu \pi^D \right) \partial_\nu \pi^C \partial_\rho \pi^A \partial_\sigma \pi^B \\
&+ \pi^C \partial_\mu \pi^A \partial_\nu \pi^B \partial_\rho \pi^D \partial_\sigma \pi^E.
\end{aligned}
$$

## 3.2 The Pseudoscalar Singlet

Our theory, like in QCD, contains an iso-singlet state $\eta'$. The state is associated with the $U(1)_A$ subgroup of an extended flavour symmetry $U(4) = SU(4) \times U(1)_A$. This symmetry is broken anomalously at the quantum level, just as in the SM. However, unlike for the SM $\eta'$, by construction this state receives no contribution from a heavier flavour like the strange quark, and so it can be close in mass to the Goldstone bosons outside the chiral limit. We can compare this state to the $\eta'$ of QCD, with the distinction that there is just one $\eta$ state in this theory, rather than a doublet which mixes and yields the $\eta$ and $\eta'$. This meson is especially interesting as it can mix with the remaining singlet Goldstone boson $\pi_3$ once the flavour symmetry is broken down to $SU(2)_u \times SU(2)_d$. Also, depending on the mass hierarchy of the theory, it can be unstable, but potentially long-lived, which would have interesting implications for phenomenology.

We account for $\eta'$ in the chiral Lagrangian by including the generator $T^0 = \frac{1}{2\sqrt{2}} \mathbb{1}_{4\times4}$. We remark that while this state may be close in mass to the Goldstones, it is not identifiable as a Goldstone and thus can differ in mass from the Goldstones even for degenerate quark masses. We encode this into our low-energy theory by assigning $\eta'$ a different decay constant from that

of the Goldstones and by explicitly breaking the $U(1)_A$ symmetry through an additional mass term $\frac{\Delta m_{\eta'}^2}{2N_c}\left[\frac{f_\pi}{4}\ln\left(\frac{\det\Sigma}{\det\Sigma^\dagger}\right)\right]^2 = -\frac{\Delta m_{\eta'}^2}{2}\eta'^2$. The $\eta'$ is then included in our chiral Lagrangian by performing an extra rotation on the vacuum[6]

$$\Sigma = \exp\left(\frac{2i\eta'T^0}{f_{\eta'}}\right)\exp\left(\frac{2i\pi}{f_\pi}\right)E. \tag{33}$$

The various interactions involving the $\eta'$ can then be calculated from the relevant terms in the chiral Lagrangian. We find,

$$\mathcal{L} = \mathcal{L}_{\text{kin}}^\pi + \mathcal{L}_{\text{mass}}^\pi + \mathcal{L}_{\text{WZW}}^\pi + \frac{1}{2}\partial_\mu\eta'\partial^\mu\eta' - \frac{m_\pi^2 + \Delta m_{\eta'}^2}{2}\eta'^2 + \frac{m_\pi^2}{48f_\pi^2}\eta'^4 + \frac{m_\pi^2}{8f_\pi^2}\eta'^2\sum_{k=1}^5\pi_k^2 + \mathcal{O}\left(\pi^6\right), \tag{34}$$

where $\mathcal{L}_{\text{kin}}^\pi + \mathcal{L}_{\text{mass}}^\pi + \mathcal{L}_{\text{WZW}}^\pi$ is the Lagrangian for the Goldstone bosons $\pi$ given in Eqs. (23), (24) and (26). Here, we used the redefinition $\eta' \to \frac{f_{\eta'}}{f_\pi}\eta'$ in order to ensure that the kinetic term for $\eta'$ is canonically normalized. We hence find, that the decay constants of $\eta'$ and the Goldstone fields are equal in the leading order of the chiral Lagrangian.[7] In the limit $\Delta m_{\eta'} \to 0$, the pseudoscalar singlet $\eta'$ and the Goldstones $\pi_n$ have the same mass $m_\pi = 2\mu^3 m/f_\pi^2$ at the lowest order in the chiral Lagrangian, since $\text{Tr}[\pi^n] = \text{Tr}[\eta'\pi^n] = \text{Tr}[\eta'^2\pi^n] = \text{Tr}[\eta'^3\pi^n] = 0$ for $n$ odd and $n \le 3$. For $\Delta m_{\eta'} \ne 0$, $\eta'$ remains massive even in the chiral limit. Interactions with $\partial_\mu\eta'$ are absent and four-point interactions with $\eta'$ are only generated by the mass term of the chiral Lagrangian.

In the mass degenerate case, the WZW term involving only pions and $\eta'$ vanishes due to anti-symmetry of epsilon tensor and flavour structure of the coset space where the pions live. The phenomenology of such $\eta'$ has been explored in the context of $Sp(4)$ composite Higgs theories [53, 57]. We once again stress that the phenomenology in our case will be different, for example the $\eta'$ in our theory can not decay to SM particles at tree level. Any decay to SM final state will be mediated by the portal between SM and $Sp(4)$ gauge group.

### 3.3 Chiral Lagrangian including spin-1 states

In order to include the lightest vector and axial-vector states of the theory, we can use the concept of hidden local symmetry from QCD [71], as has already been done in [7]. Here, the spin-1 mesons are introduced as dynamical objects by describing them as the gauge bosons of a spontaneously broken local symmetry. We stress that this is an effective treatment and that the spin-1 states are not fundamental gauge fields and that the local symmetry is purely auxiliary.

To this end, we take a second copy of the $SU(4)$ flavour symmetry and "gauge it." This introduces 15 vector bosons, corresponding to the gauged $SU(4)$. The symmetry is broken in the low-energy regime, and in addition to the five PNGBs $\pi^a T^a$, we have fifteen (exact) PNGB $\sigma^b T^b$. The latter are "eaten" by the vector bosons, providing them with the longitudinal degrees of freedom required for the massive vector fields,

$$\rho_\mu = \sum_{a=1}^{15}\rho_\mu^a T^a. \tag{35}$$

---

[6]Under the axial group $U(1)_A$ we have $\psi_{R/L} \to e^{\pm i\alpha}\psi_{R/L}$ and the chiral field transforms as $\Sigma \to e^{-2i\alpha}\Sigma$. Upon confinement, the flavour symmetry $U(4)$ breaks spontaneously to $Sp(4)\times U(1)_A$. The Goldstone bosons including the $\eta'$ live in the coset space $U(4)/(Sp(4)\times U(1)_A)$. The determinant of the chiral field is then $\det\Sigma = \exp\left(\frac{2\sqrt{2}i\eta'}{f_{\eta'}}\right) \ne 1$.

[7]This is not overly surprising, as in the large $N_c$-limit the $U(1)_A$ anomaly vanishes, $f_{\eta'} = f_\pi(1 + O(1/N_c))$ [70], and $\eta'$ can be considered as an additional Goldstone boson in line with the construction of the chiral field in Eq. (33).

The $\rho_\mu^a$ decompose into a lighter ten-plet and a heavier quintuplet, corresponding to the unbroken/broken generators of $SU(4)$, respectively. These can be identified with the ordinary vector and axial-vector multiplets in QCD, see Fig. 2. Explicitly $\rho_\mu$ is given by

$$
\begin{aligned}
\rho_\mu =&\frac{1}{2\sqrt{2}}
\begin{pmatrix}
\rho^3 & \rho^1 - i\rho^2 & 0 & \rho^5 - i\rho^4 \\
\rho^1 + i\rho^2 & -\rho^3 & -\rho^5 + i\rho^4 & 0 \\
0 & -\rho^5 - i\rho^4 & \rho^3 & \rho^1 + i\rho^2 \\
\rho^5 + i\rho^4 & 0 & \rho^1 - i\rho^2 & -\rho^3
\end{pmatrix}_\mu \\
&+\frac{1}{2\sqrt{2}}
\begin{pmatrix}
\rho^{14} + \rho^{15} & \rho^{13} - i\rho^8 & \sqrt{2}\rho^{10} - i(\rho^6 + \rho^9) & \rho^{11} - i\rho^7 \\
\rho^{13} + i\rho^8 & \rho^{15} - \rho^{14} & \rho^{11} - i\rho^7 & \sqrt{2}\rho^{12} - i(\rho^6 - \rho^9) \\
\sqrt{2}\rho^{10} + i(\rho^6 + \rho^9) & \rho^{11} + i\rho^7 & -(\rho^{14} + \rho^{15}) & -(\rho^{13} + i\rho^8) \\
\rho^{11} + i\rho^7 & \sqrt{2}\rho^{12} + i(\rho^6 - \rho^9) & -(\rho^{13} - i\rho^8) & -(\rho^{15} - \rho^{14})
\end{pmatrix}_\mu ,
\end{aligned}
\tag{36}
$$

with the first line defining the matrix describing the lightest axial-vector states under $D$-parity, and the second line being the vector states under $D$. We may then complement our chiral Lagrangian by adding the kinetic and mass terms for the spin-1 states,

$$
\mathcal{L}_\rho = -\frac{1}{2}\mathrm{Tr}\left(\rho_{\mu\nu}\rho^{\mu\nu}\right) + \frac{1}{2}m_\rho^2\,\mathrm{Tr}\left(\rho_\mu\rho^\mu\right),
\tag{37}
$$

with $\rho_{\mu\nu} = \partial_\mu\rho_\nu - \partial_\nu\rho_\mu - ig_\rho\left[\rho_\mu, \rho_\nu\right]$ being the non-Abelian field strength tensor. The $\rho$ mass can be expressed in terms of low-energy constants (LECs) of the full $SU(4)\times SU(4)$ theory [8]. We retain here $m_\rho$ as a free parameter, but it will be ultimately fixed by the ultraviolet theory, see section 6. By inspection of the $\rho$ matrix in the generator basis (36), we can identify the QCD-like spin-1 states. Transformations between these two bases are given in Appendix A, see in particular equation (A.17)

The interactions between the Goldstone bosons and spin-1 states may be obtained by introduction of a "covariant derivative",

$$
D_\mu\Sigma = \partial_\mu\Sigma + ig_\rho\left(\rho_\mu\Sigma + \Sigma\rho_\mu^T\right),
\tag{38}
$$

where $g_\rho$ is the phenomenological coupling between spin-1 meson states and the Goldstones. In practice, coupling the axial-vectors in this way results in non-diagonal kinetic terms when we promote our derivatives to covariant ones in the leading order chiral Lagrangian. Coupling both vectors and axial-vectors in this way ensures that axial vector states are always higher in mass than the vector states. To make this explicit, we follow [72] and expand our kinetic term to quadratic order in the fields. The kinetic terms for Goldstone and axial-vector fields are non-diagonal:

$$
\mathcal{L}_{\mathrm{kin}} = \frac{1}{2}\mathrm{Tr}\left(\partial_\mu\pi - \frac{1}{\sqrt{2}}f_\pi g_\rho a_\mu\right)^2,
\tag{39}
$$

where $a_\mu = \sum_{k=1}^5 \rho_\mu^k T^k$ is the matrix describing the five axial-vector states, defined in (36). The action is diagonalized by the field re-definitions

$$
\begin{aligned}
&a_\mu \rightarrow \tilde{a}_\mu + \frac{g_\rho}{\sqrt{2}m_\rho^2}\tilde{f}_\pi\partial_\mu\tilde{\pi}, \qquad \pi \rightarrow Z^{-1}\tilde{\pi}, \qquad f_\pi \rightarrow Z^{-1}\tilde{f}_\pi, \\
&Z^2 = \left(1 - g_\rho^2\tilde{f}_\pi^2/2m_\rho^2\right) \approx \left(1 + g_\rho^2\tilde{f}_\pi^2/2m_\rho^2\right)^{-1},
\end{aligned}
\tag{40}
$$

where the tilde superscript is reserved for the physical basis, and can be dropped once the action contains only physical states. This splits the vector and axial-vector states in terms of their masses, with the axial-vector mass now related by

$$
m_a^2 = m_\rho^2/Z^2.
\tag{41}
$$

The exact value of $Z$, and therefore the relative mass of the axial-vectors, now depends on the low-energy constants $g_\rho$, $f_\pi$ and $m_\rho$. The value of $g_\rho$ is often estimated by the Kawarabayashi-Suzuki-Riazuddin-Fayyazuddin (KSRF) relation [73, 74] $2g_{\rho\pi\pi}^2 \approx \frac{m_\rho^2}{f_\pi^2}$, which implies that $Z^2 \approx \frac{1}{2}$. This value of $Z$ ensures that axial-vector states are heavier than vector states by a factor of $\sim \sqrt{2}$. The KSRF relation has been tested on the lattice for $Sp(4)_c$ gauge theory with quenched fundamental fermions, with the discrepancy between lattice and theoretical values being $\sim 10\%$ [8]. We can conclude that the axial-vector states should always be significantly heavier than the vectors. In what follows, we will usually neglect these states, assuming them to be decoupled.

We stress, however, that the applicability of KSRF in symplectic theories is not yet a settled issue. Studies on $SU(2)$ gauge theories have found that the relation overestimates the value of the $\rho\pi\pi$ coupling by $> 20\%$ [75]. We reference it here only to provide a rough estimate on mass splitting between vector and axial-vector states.

### 3.4 The WZW action including spin-1 states

In this section we show how to include the spin-1 composite states in the "anomalous" dark sector interactions induced by the WZW action. In the spirit of the construction above, it is clear that the action (25) alone is not "gauge-invariant" and additional terms must be added to restore invariance; they may be obtained iteratively by the Noether method.

To proceed, we use the Hidden Local Symmetry framework, and following [72, 76], we first define the fields[8]

$$L \equiv d\Sigma\Sigma^\dagger, \; R \equiv \Sigma^\dagger d\Sigma, \tag{42}$$

with $d\Sigma = dx^\mu \partial_\mu \Sigma$, in terms of which the WZW action may be rewritten in the compact form,

$$\Gamma_{\text{WZW}} = C \int_{Q_5} \text{Tr}(L)^5 = C \int_{Q_5} \text{Tr}(R)^5, \tag{43}$$

with $C$ the prefactor of the integral in (25). We recall that the chiral field transforms as $\Sigma \to U\Sigma U^T$ with $U \in Sp(4)$. Restricting our focus to the lighter spin-1 ten-plet, infinitesimally, $U$ is given by $\mathbb{1} + iG + \dots$ with $G = G^a T^a$, $a = 6, \dots, 15$ an element of the Lie algebra; the generators are given in App. A. The infinitesimal transformation is then

$$\Sigma \to \Sigma + i\left(G\Sigma + \Sigma G^T\right) + \dots, \tag{44}$$

i.e., the infinitesimal variation in $\Sigma$ under $Sp(4)$ is given by $\delta\Sigma = i\left(G\Sigma + \Sigma G^T\right)$. The ensuing variation of WZW action is then given by

$$\delta\Gamma_{\text{WZW}} = 5Ci\int_{Q_5} \text{Tr}\left(dGL^4 + dG^T R^4\right) = -5Ci\int_{M^4} \text{Tr}\left(dGL^3 - dG^T R^3\right). \tag{45}$$

In the second equality we have used the property of the exact forms $dL - L^2 = dL^3 - L^4 = 0$, $dR + R^2 = dR^3 + R^4 = 0$, and applied Stokes' theorem. Now we introduce the 1-form $\rho = \rho_\mu dx^\mu$, and use the leading order expression for the variation in $\rho$, $\delta\rho \approx dG$, to find the first correction the WZW action,

$$\Gamma_{\text{WZW}}^{(1)} = \Gamma_{\text{WZW}} + 5Cig_\rho \int_{M^4} \text{Tr}\left(\rho L^3 - \rho^T R^3\right). \tag{46}$$

---

[8]Unlike in QCD [72] field $L(R)$ does not transform as a nonet under the left(right)-handed part of the chiral symmetry. Our considered theory possesses an enlarged flavour symmetry, and we consider only the two-flavour case. We use this notation only to save space, and for some consistency with the literature.

This process can be repeated to iteratively build the full expression for the gauged WZW action. We simply find the variation in this first correction under local $Sp(4)$ and determine the appropriate counterterm whose variation in $\rho$ will cancel it. We now present the particular form of the gauged action for $Sp(4)_c$ gauge theory:

$$
\begin{aligned}
\Gamma_{\text{WZW}}(\Sigma, \rho) = \Gamma_{\text{WZW}}^{(0)} &+ 5Cig_\rho \int_{M^4} \text{Tr}\left(\rho L^3 - \rho^T R^3\right) \\
&+ \frac{5}{2}Cg_\rho^2 \int_{M^4} \text{Tr}\left(\rho L \rho L - \rho^T R \rho^T R\right) \\
&+ 5Cg_\rho^2 \int_{M^4} \text{Tr}\left(d\rho \, d\Sigma \rho^T \Sigma^\dagger - d\rho^T d\Sigma^\dagger \rho \Sigma\right) \\
&- 5Cig_\rho^3 \int_{M^4} \text{Tr}\left(\rho^3 L - \left(\rho^T\right)^3 R + \rho \Sigma \rho^T \Sigma^\dagger \rho L - \rho^T \Sigma^\dagger \rho \Sigma \rho^T R\right) \\
&- 5Cg_\rho^4 \int_{M^4} \text{Tr}\left(\rho^3 \Sigma \rho^T \Sigma^\dagger - \left(\rho^T\right)^3 \Sigma^\dagger \rho \Sigma\right),
\end{aligned}
\tag{47}
$$

with $L$ and $R$ given by (42). It should be noted that the previous expression is not just valid for working out interactions involving composite vector states. It is valid for any symmetry with the transformation law (44). This means that gauging the full $SU(4)$ flavour symmetry, rather that just the $Sp(4)$ subgroup, allows us to characterize the interactions involving also the axial-vector quintuplet. We stress that the gauged expression for the WZW action allows studying number changing interactions that include vector states. If the kinematic conditions are favourable, i.e., if vectors and scalars are close in mass, this will affect the SIMP mechanism for DM freeze-out [33]. We also note that the explicit construction of WZW including the rho mesons has allowed us to realize that two of the remaining $\rho$ mesons are unstable due to the AVV anomaly. Thus, not all (off-diagonal) $\rho$ mesons of the theory are stable as claimed in [29]. The generic expressions for the WZW term for $SU(2N_f)/Sp(2N_f)$ breaking have been previously presented in [77] and for general coset space they are presented in [78]. It should be noted that the use of Hidden Local Symmetry (HLS) has allowed us to use gauge invariance, thus fixing all free parameters of the Lagrangian shown in Eq. 47. The HLS formalism in general holds for a complex representation of $SU(N)$ gauge group. We expect it to be valid also in our theory, however no explicit lattice tests exist.

## 4 SIMPs under Abelian gauge symmetries

Eventually, we want to couple the non-Abelian dark sector to the SM. A simple, but by no means the only option is to gauge (part of) the theory under a new Abelian dark gauge group $U(1)'$.[9] The new (massive) vector $V^\mu$ may then kinetically mix with SM hypercharge. Denoting the respective field strengths by $V_{\mu\nu}$ and $B_{\mu\nu}$, the interaction is given by,

$$
\mathcal{L}_{\text{int}} = \frac{\varepsilon}{2\cos\theta_W} B_{\mu\nu} V^{\mu\nu},
\tag{48}
$$

where $\theta_W$ is the SM weak angle. The consequences and phenomenology of this "vector portal" have been studied in great detail; see [80–82] and references therein. For example, after electroweak symmetry breaking and when the $V^\mu$-mass is well below the electroweak scale, (48)

---

[9]See e.g. [79], where a $Sp(2N)$ gauge theory with $2N_f$ Weyl fermions is coupled to the SM through an axion-like mediator. We also note in passing that the vector portal induces at loop-level also a Higgs-portal coupling. Since we only consider leading-order effects, this loop-suppressed contribution is left for future work. Note that any explicit Higgs portal coupling will be at least a dimension-five operator, due to the fermionic nature of our dark quarks, and thus will be suppressed as well, and is hence likewise postponed.

Table 2: Possible charge assignments, their associated symmetry breaking patterns, and ensuing multiplet structure. When $\pi^C$ is a singlet, it is not protected by the flavour symmetry and can in principle decay.

| Charge Assignment $\mathcal{Q}$ | Breaking Pattern | Multiplet Structure |
|---|---|---|
| $\mathrm{diag}(+a,-a,-a,+a)$ | $Sp(4) \to SU(2) \times U(1)$ | $\begin{pmatrix} \pi^C \\ \pi^D \\ \pi^E \end{pmatrix}, \begin{pmatrix} \pi^A \\ \pi^B \end{pmatrix}$ |
| $\mathrm{diag}(+a,+a,-a,-a)$ | $Sp(4) \to SU(2) \times U(1)$ | $\begin{pmatrix} \pi^C \\ \pi^A \\ \pi^B \end{pmatrix}, \begin{pmatrix} \pi^D \\ \pi^E \end{pmatrix}$ |
| $\mathrm{diag}(+a,+b,-a,-b)\,,\, a \neq b$ | $Sp(4) \to U(1) \times U(1)$ | $(\pi^C), \begin{pmatrix} \pi^A \\ \pi^B \end{pmatrix}, \begin{pmatrix} \pi^D \\ \pi^E \end{pmatrix}$ |
| $\begin{pmatrix} 0 & 0 & a & 0 \\ 0 & 0 & 0 & \pm a \\ a & 0 & 0 & 0 \\ 0 & \pm a & 0 & 0 \end{pmatrix}$ | $Sp(4) \to SU(2) \times U(1)$ | $\begin{pmatrix} \pi^C \\ \pi^{A,B} \\ \pi^{E,D} \end{pmatrix}, \begin{pmatrix} \pi^{D,E} \\ \pi^{B,A} \end{pmatrix}$ |
| other off-diagonal assignments | $Sp(4) \to U(1) \times U(1)$ | $(\pi^C), \begin{pmatrix} \pi^A \\ \pi^E \end{pmatrix}, \begin{pmatrix} \pi^B \\ \pi^D \end{pmatrix}$ or similar |

induces "photon-like" interactions with SM fermions $f$, $\epsilon q_f \bar{f} \gamma_\mu f V^\mu$ where $q_f$ is the electric charge of $f$. The new vector is then commonly referred to as "dark photon." For the purpose of this work, our principal interest lies in exploring the various options of coupling $V^\mu$ to the non-Abelian dark sector. The study of its phenomenological consequences is left for future work.

### 4.1 Charge assignment in the UV

In what follows, we explore the possible charge assignments of the Dirac flavours under $U(1)'$. We summarize the findings in Tab. 2. Since the Weyl basis is used to make the global symmetries of the theory manifest, we may also look at the transformation properties of the Weyl spinors under $U(1)'$. The "vector" $\Psi$ which collects the Weyl spinors of our theory is related to the Dirac fields through (6). Fixing the charges of our Dirac fields is therefore sufficient in fixing the charge assignment in the basis of Weyl spinors.

Under a local $U(1)'$ transformation with gauge parameter $\alpha(x)$, the components of $\Psi$ transform as

$$\Psi^{ia} \to \exp\big[i\alpha(x)\mathcal{Q}_{ij}\big]\Psi^{ja} \simeq \Psi^{ia} + i\alpha(x)\mathcal{Q}_{ij}\Psi^{ja}\,, \tag{49}$$

with $i,j$ flavour indices and $a$ denoting the colour index of each component of $\Psi$. The introduction of the covariant derivative then renders the theory invariant under $U(1)'$ gauge transformations,

$$\partial_\mu \Psi^{ia} \to D_\mu \Psi^{ia} = \partial_\mu \Psi^{ia} + i e_D V_\mu \mathcal{Q}_{ij} \Psi^{ja}\,. \tag{50}$$

$\mathcal{Q}_{ij}$ is a symmetric matrix in flavour space containing the $U(1)'$ charges of our Weyl flavours; $e_D$ is the $U(1)'$ gauge coupling and the gauge field transforms as $V^\mu \to V^\mu + \frac{1}{e_D}\alpha(x)V^\mu$.

To begin with, let us restrict ourselves to the particular case of diagonal charge prescriptions. To obtain vector-like couplings, the definition of $\Psi$ in (6) implies that the first (second) and third (fourth) components must have opposite charge. Now for any vector-like charge assignment in Dirac flavour space $(a,b)$, the charge matrix for the Weyl spinors is then given

by

$$\mathcal{Q} = \text{diag}(+a, +b, -a, -b) \quad \text{(vector-like assignment)}. \tag{51}$$

Gauging the theory under $U(1)'$ may provide a source of explicit global symmetry breaking. To understand this, we study how the gauge interaction term in the Lagrangian,

$$\mathcal{L} \supset -e_D V_\mu \left( \left(\Psi^i\right)^\dagger_a \overline{\sigma}^\mu \mathcal{Q}_{ij} \Psi^{ja} \right), \tag{52}$$

transforms under the *remaining* flavour symmetry $Sp(4)$ as

$$\Psi^{ia} \to V_{ij} \Psi^{ja} = \left( 1 + i\theta^N T^N_{ij} + \dots \right) \Psi^{ja}, \tag{53}$$

with $i, j$ flavour indices. Here, $N = F, \dots, O$, correspond to the $Sp(4)$ subgroup of the $SU(4)$ flavour symmetry, as given by (A.18). Only the variation under this subgroup is relevant since the global $SU(4)$ is already broken.

The variation in the Lagrangian density is found to be

$$\delta\mathcal{L} = ie_D V_\mu \left( (\Psi)^\dagger \overline{\sigma}_\mu \theta^N \left[ \mathcal{Q}, T^N \right] \Psi \right), \tag{54}$$

This expression is general and not specific to a particular choice of $U(1)'$ charges. We see that the remaining flavour symmetry is spanned by those generators $T^N$ of $Sp(4)$ that commute with $\mathcal{Q}$.

Making use of this expression allows us to identify the unbroken symmetries for any particular charge assignments. The distinct choices are outlined in table 2. We find that two diagonal assignments preserve an $SU(2) \times U(1)$ subgroup of the flavour symmetry, while all others maximally break it to $U(1) \times U(1)$. In what follows, we show that these flavour-conserving interactions lead to anomaly-free couplings between Goldstone bosons and the dark photon.

We now turn our attention to the possibility of off-diagonal couplings, i.e., allowing charge assignments $\mathcal{Q}_{ij}, i \neq j$ in (50). Thereby, the various Weyl flavours may couple differently to the $U(1)'$ field, allowing, for example, flavour changing processes. Since (54) is valid for any symmetric $\mathcal{Q}$, we can also see how these flavour changing assignments affect the global symmetries of the theory. We see similarities with the diagonal prescriptions; again exactly two assignments preserve a $SU(2) \times U(1)$ symmetry, while all others maximally break the flavour symmetry to $U(1) \times U(1)$. We note that from the definition of $\Psi$ (6), it can be seen that the flavour-conserving assignments are those which couple the left- and right-handed components of the same Dirac spinors $u$ and $d$. While two of the charge assignments shown in Tab. 2 are the same as those identified in [11], we also identify the existence of the flavour changing currents.

## 4.2 Stability of the flavour diagonal Goldstone

In the DM context, the question of the stability of the Goldstone bosons is of course of crucial importance phenomenologically. In particular, the stability of the flavour diagonal Goldstone, $\pi^C$, is important to understand. As is the case for the SM QCD neutral pion, a coupling to an external mediator (dark photon) can destabilize the particle. In order to establish the stability of particles under a particular charge assignment, it is sufficient to determine the representations in which the particles transform under the remaining flavour symmetry. In particular, when all Goldstones transform in a non-trivial representation of the preserved symmetry, they are protected from decay. The multiplet structure of Goldstones under various charge assignments are given in Tab. 2.

For all assignments discussed in the previous section which maximally break $Sp(4)$ flavour to $U(1) \times U(1)$, all off-diagonal Goldstones acquire a net $U(1)'$ charge. Since the diagonal

Goldstone state remains uncharged, it splits from the others and is made a flavour singlet. As a result, it can decay. The symmetry-preserving assignments are a different story, however. When the symmetry is instead broken as $Sp(4) \rightarrow SU(2) \times U(1)$, then only a pair of Goldstones from $\pi^{A,B,D,E}$ acquire equal and opposite $U(1)'$ charges. The remaining PNGBs form an uncharged triplet which transforms in the adjoint of the remaining $SU(2)$ flavour symmetry.

We remind the reader that our discussion of global symmetries thus far has assumed that the theory contains two degenerate Dirac flavours. In the presence of mass-splitting between the dark quarks, flavour symmetry is already broken to $SU(2) \times SU(2)$ and the $\pi^C$ is always a singlet, although accidentally stable in the isolated $Sp(4)_c$ sector. Once we couple it to the dark photon, it is expected to become unstable, as it is not protected by any symmetry. Care then must be taken in ensuring that such theories, when entertained as DM scenarios, can be made consistent with constraints from the astrophysics and cosmology; we leave such exploration for future work.

Finally, we comment on the stability of the iso-singlet pseudoscalar, the $\eta'$. This particle is always a flavour singlet. This ensures no charge assignment protects its decay. As soon as the hidden sector is coupled to $U(1)'$ the $\eta'$ will be destabilized, and decays through tree level processes mediated by pairs of dark photons, like the corresponding $\eta'$ does in QCD.

## 4.3 $U(1)'$ interactions with mesons

We now turn our attention to the interactions in the dark sector with the $U(1)'$ field. To see how the light mesonic states interact with the associated gauge field, we must examine the transformation properties of the vacuum under the $U(1)'$ symmetry. Under $U \in U(1)'$, $\Sigma$ transforms as

$$\Sigma \rightarrow U\Sigma U^T \sim \Sigma + i\alpha'(x)\left(\mathcal{Q}\Sigma + \Sigma\mathcal{Q}^T\right), \tag{55}$$

with $\alpha'$ parameterizing the local $U(1)'$ transformation. From this we can determine that the covariant derivative acting on $\Sigma$,

$$D_\mu\Sigma = \partial_\mu\Sigma + ie_D V_\mu(\mathcal{Q}\Sigma + \Sigma\mathcal{Q}^T), \tag{56}$$

In the following $\mathcal{Q}$ is taken to be diagonal and $U = U^T$. As discussed in the previous section, the presence of this coupling breaks the global flavour symmetry to some subgroup of $Sp(4)$. A subset of the PNGBs become charged under $U(1)'$ and couple directly to the dark photon.

The specifics of the interactions of the dark photon with the pion fields depend on the charge prescription $\mathcal{Q}$. An interesting aspect of this theory is that through different choices of charge assignment, we can selectively couple the vector to different pairs of off-diagonal pions. For example, for the previously mentioned $(+1,-1,-1,+1)$ prescription, we couple only to the $\pi^{A,B}$ states with interactions of the form

$$\mathcal{L}_{V-\pi} = -2ie_D V^\mu\left(\pi^A\partial_\mu\pi^B - \pi^B\partial_\mu\pi^A\right) + e_D^2 V_\mu V^\mu \pi^A\pi^B. \tag{57}$$

For a $(+1,+1,-1,-1)$ charge prescription, we couple only to the other pair of Goldstones $\pi^{D,E}$ with the same type of interaction seen above. This property distinguishes the theory from two-flavour QCD. Here, even nontrivial charge assignments can preserve stability of all Goldstones. A pair of states carry $U(1)'$ charge, while the remaining states form a flavour triplet. The above interaction Lagrangian sources two distinct interactions, a three point interaction from the first term, and four point interaction from the second. Both of these interactions must be taken into account to obtain gauge invariant results. The explicit Feynman rules are provided in Appendix C.1.

In addition to the Goldstone mesons, the spin-1 mesonic states interact with $V^\mu$. Gauge invariance of our action is preserved through the inclusion of the term

$$\mathcal{L}_{V-\rho} = -\frac{e_D}{g_\rho} V_{\mu\nu} \operatorname{Tr}(\mathcal{Q}\rho^{\mu\nu}) . \tag{58}$$

Performing a global flavour transformation of the vector multiplet (transforming in the adjoint representation of the flavour symmetry), we verify that $\mathcal{L}_{V-\rho}$ respects all symmetries not already explicitly broken by gauging the theory.

Again, the question of charge prescription is of central importance. The trace in (58) picks out only the flavour diagonal vector mesons, but, in addition, specific charge prescriptions will induce mixing among *different* flavour-diagonal vector mesons. For $Sp(4)_c$ at hand, there are two flavour diagonal vector mesons, $\rho^{N,O}$. The first (second) charge assignment in Table (2) couples only to $\rho^N$ ($\rho^O$). This implies that one of these states is stable against decays into the SM, while the other is unstable for both of these assignments. This fact also highlights an interesting difference between $Sp(4)$ gauge theory and standard QCD: since the Goldstones of the former theory do not transform in the same representation as the vectors under flavour, they are not protected by the same symmetry. The flavour-diagonal $\rho$ can decay while the flavour-diagonal Goldstone boson remains stable.

Decays of Goldstone bosons occur through processes mediated by pairs of dark photons. The relevant $\pi - V - V$ vertex is sourced by the gauged WZW action. Following the same process described in subsection 3.4, the anomalous interactions involving the dark photon can be computed by gauging the WZW term under the $U(1)'$ symmetry. The complete expression is

$$
\begin{aligned}
\Gamma^V_{\text{WZW}} = {}& 5Ci \int_{M^4} e_D V \operatorname{Tr} \mathcal{Q}\left(L^3 - R^3\right) \\
& -10C \int_{M^4} e_D^2 V dV \operatorname{Tr} \mathcal{Q}^2 (L+R) ,
\end{aligned}
\tag{59}
$$

with $dV = \partial_\mu V dx^\mu$. For example, the term in the second line leads to the following interaction

$$\mathcal{L}_{\text{int}} \supset \frac{40iCe_D^2}{f_\pi^2} \varepsilon_{\mu\nu\alpha\beta} V^\mu \partial^\nu V^\alpha \operatorname{Tr}\left(\mathcal{Q}^2 \partial^\beta \pi\right) . \tag{60}$$

Since all the generators of $SU(4)/Sp(4)$ are traceless, it is clear that the condition $\mathcal{Q}^2 \propto \mathbb{1}$ is sufficient for this vertex to vanish for all Goldstone bosons. The two flavour-conserving charge assignments given in table 2 satisfy precisely this condition. Since these charge assignments ensure that all Goldstones transform nontrivially under flavour, this cancellation holds to all orders, and this anomalous vertex is not generated by higher order chiral terms. This is not generically the case for other gauge theories. For fermions in complex representations, the anomaly cancellation condition $\mathcal{Q}^2 \propto \mathbb{1}$, cancels this vertex only at leading order. Higher order terms facilitate this interaction [35] and so allow Goldstones to decay, highlighting another key difference between pseudo-real and complex representations.

## 4.4 Goldstone mass splitting through radiative corrections

In the presence of explicit symmetry breaking due to $U(1)'$ charges, the masses of our charged Goldstones are renormalized. This can be incorporated into the theory through the inclusion of an explicit term of the form

$$\mathcal{L}_{V\text{-split}} = \kappa \operatorname{Tr}\left(\mathcal{Q}\Sigma\mathcal{Q}\Sigma^\dagger\right) , \tag{61}$$

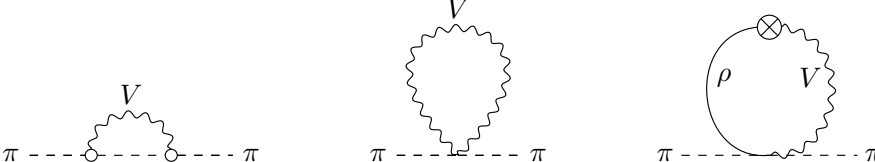

Figure 3: Diagrams contributing to the Goldstone mass-splitting due to $U(1)'$ gauge interactions. Curly lines denote $U(1)'$ gauge propagators, solid lines $\rho$ meson propagators. Empty dots signify vertices which sum over bare couplings of the Goldstones to $V^\mu$, as well as contributions from mixing with the $\rho$ mesons. The crossed dot indicates oscillation between $V^\mu$ and $\rho$.

where $\kappa$ is the associated LEC, describing the induced mass splitting due to the gauge interactions. For all charge prescriptions which preserve $\pi^C$ stability (with $\mathcal{Q}^2 \propto \mathbb{1}$), Goldstones charged under $U(1)'$ acquire a correction to their masses which takes the form

$$\Delta m_\pi^2 = \frac{2\kappa e_D^2}{f_\pi^2} \,. \tag{62}$$

Any overall rescaling of the charge matrix can be absorbed into the definition of $e_D$. For other charge prescriptions, corrections for a given Goldstone can be acquired by multiplying the above by a factor of $Q_\pi^2/(2e_D)^2$, with $Q_\pi$ the $U(1)'$ charge associated with the particle in question. With the full chiral theory, this mass splitting can be estimated through resonance contributions to the self-energy of the Goldstones, under the assumption of vector meson dominance [83].

Assuming that axial vectors are heavy enough as to be decoupled, the three diagrams of figure 3 need to be computed. Using the interactions discussed in previous sections, we may evaluate the mass corrections at leading order in $m_\pi^2$. We compute the diagrams in figure 3 with the external Goldstone lines on-shell and assume *vector meson dominance* [83], wherein the form factors of the Goldstone bosons are dominated by the spin-1 multiplet at low energies. The leading order expression is obtained by neglecting the terms proportional to the on-shell momentum squared. Expressed as a loop integral, the correction takes the form

$$\Delta m_\pi^2 = 6e_D^2 m_\rho^4 \int \frac{d^4q}{(2\pi)^4} \frac{1}{\left(q^2 - m_V^2\right)\left(q^2 - m_\rho^2\right)^2} \,, \tag{63}$$

with the left-hand side of the equation denoting the self-energy of the Goldstone evaluated on-shell. Such a treatment done entirely in terms of the low energy degrees of freedom of the theory hence allows us to estimate the Goldstone mass splitting. Equation (63) can be evaluated to determine the leading correction to the charged Goldstone masses, which takes the compact form

$$\Delta m_\pi^2 = \frac{6e_D^2}{(2\pi)^2} \frac{m_\rho^4}{m_V^2 - m_\rho^2} \log\left(\frac{m_V^2}{m_\rho^2}\right). \tag{64}$$

This result is exact in the chiral limit, and is independent of the hierarchy between $V$ and $\rho$. The expression is continuous even at the crossover point $m_V = m_\rho$. Comparison with (62) lets us finally give an estimate of $\kappa$, expressed as a dimensionless ratio, at leading order

$$\frac{\kappa}{m_\rho^4} \approx \frac{3}{4\pi^2} \frac{f_\pi^2/m_\rho^2}{m_V^2/m_\rho^2 - 1} \log\left(\frac{m_V^2}{m_\rho^2}\right). \tag{65}$$

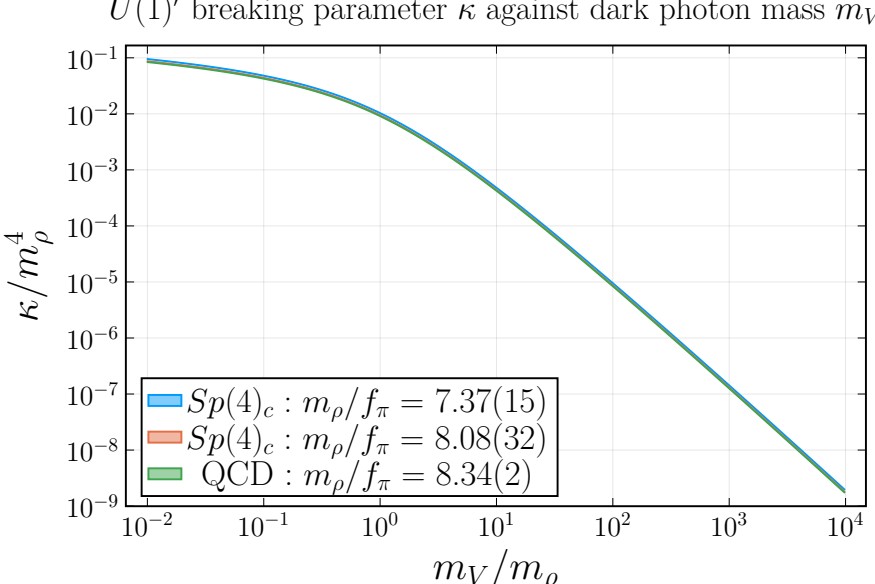

Figure 4: The low energy constant $\kappa$ parameterizing the explicit symmetry breaking due to the $U(1)'$ interaction in units of $m_\rho^4$ as a function of $m_V/m_\rho$ according to (65). The ratio of the dark vector meson mass to the dark Goldstone decay constant $m_\rho/f_\pi$ has been constrained through the lattice results of [6]. We quote the results for two limiting cases: First, for the lattice simulation with the heaviest fermions for which leading-order chiral perturbation theory is expected to hold. This corresponds to $m_\rho/f_\pi = 7.37(15)$. Secondly, for the extrapolation to the chiral limit, where $m_\rho/f_\pi = 8.08(32)$ was found. We conclude that this relation shows only a very weak dependence on the dark quark masses. In addition, we plot (65) for the experimental value of Standard Model QCD.

Here, two different hadronic quantities of the strongly interacting theory enter: the mass of the vector mesons $m_\rho$ and the decay constant of the (pseudo-)Goldstone bosons $f_\pi$. They are, however, not independent LECs and their ratio is constrained from lattice data [6]. Hence, once the underlying theory is fixed, $\kappa$ only depends on $m_V$ and one ratio of dark hadronic observables; see also section 6.1 for a discussion of the free parameters of the UV complete theory.

Close to the chiral limit, i.e., in a regime where $m_\rho/m_\pi > 1.4$, $m_\rho/f_\pi > 7$ always holds [6]. The heaviest fermions for which the theory was sufficiently close to chirality resulted in $m_\rho/f_\pi = 7.37(13)$. An extrapolation to the chiral limit yielded $m_\rho/f_\pi = 8.08(32)$.

In Fig. 4 we plot $\left(\kappa/m_\rho^4\right)$ as a function of $m_V/m_\rho$ for the two aforementioned ratios of $m_\rho/f_\pi$ as well as the experimental value from SM QCD. We see that the underlying gauge theory strongly constrains (65). Even for different gauge groups the results are very similar [84]. Therefore, the quantities $f_\pi$ and $m_\rho$ and even their ratio can, in general, *not* be considered as free, independent parameters. Fig. 4 also illustrates the potential for fine radiative mass splittings, independent of the splitting between Dirac flavours. $m_V$ and $e_D$ remain independently variable parameters, allowing a wide range of mass corrections to be obtained depending on what phenomenology is of interest. We also note that the expected behaviour of $\kappa$ as a splitting between $V$ and $\rho$ occurs is reflected in Fig. 4, with radiative corrections diminishing as $m_V$ becomes large.

# 5 Chiral Lagrangian in presence of small mass splittings in fermion masses

In this section, we study the consequences for *non-degenerate* fermion masses in the UV, $m_u \neq m_d$. Whereas this is the case for any two flavours in QCD, for the $SU(4)_c$ case of interest here, the low-energy effective theory in the presence of a fundamental mass-splitting has not yet been studied. As mentioned in Sec. 2.1, the global flavour symmetry group $SU(4)$ breaks explicitly to $Sp(2)_u \times Sp(2)_d$ for $m_u \neq m_d$ in the two flavours $u = (u_L, \tilde{u}_R), d = (d_L, \tilde{d}_R)$. We may restrict ourselves to positive mass splittings $\Delta m_{du} = m_d - m_u \geq 0$; the opposite case is simply achieved by a relabelling of the fields $u \to d$ and $d \to u$ as nothing else distinguishes them. The remaining symmetry $Sp(2)_u \times Sp(2)_d$ invites us to switch to another basis where the generators are (anti-)block-diagonal given by (A.12). We denote the generators in this basis as $\tilde{T}^a$. The mass matrix can then be expressed as a sum of block-diagonal forms,

$$M = M_u + M_d = m_u \begin{pmatrix} i\sigma_2 & 0 \\ 0 & 0 \end{pmatrix} + m_d \begin{pmatrix} 0 & 0 \\ 0 & i\sigma_2 \end{pmatrix} = 2i \left( m_u \tilde{T}_u^2 + m_d \tilde{T}_d^2 \right), \qquad (66)$$

where $M_u, M_d$ are mass matrices of the respective flavours and $\tilde{T}_u^2, \tilde{T}_d^2$ are generators of $Sp(2)_u \times Sp(2)_d$.

## 5.1 Mass and kinetic terms in the non-degenerate case

When mass splittings are taken into account, the low energy decay constants as well as vacuum condensates will generally be modified, or, better, carry a flavour-dependence. Our choice of decay constants and vacuum condensates that enter the construction of the chiral Lagrangian is motivated by the lattice results of Sec. 6. Concretely, we make the following replacement when considering split $u$ and $d$ quark masses,

$$f_\pi \xrightarrow{m_u \neq m_d} \begin{cases} f_\pi & \text{for } \pi_{1,2,4,5} \\ f_{\pi_3} & \text{for } \pi_3 \end{cases}, \quad \mu^3 \xrightarrow{m_u \neq m_d} \begin{cases} \mu_u^3 = \frac{1}{2} \langle u^T \sigma_2 S E_2 u \rangle \\ \mu_d^3 = \frac{1}{2} \langle d^T \sigma_2 S E_2 d \rangle \end{cases}, \qquad (67)$$

where $E_2 = i\sigma_2$ is a matrix in flavour space. The chiral condensate changes to

$$\langle q_i q_j \rangle = \begin{pmatrix} 0 & \mu_u^3 & 0 & 0 \\ -\mu_u^3 & 0 & 0 & 0 \\ 0 & 0 & 0 & \mu_d^3 \\ 0 & 0 & -\mu_d^3 & 0 \end{pmatrix} = \mu_u^3 \begin{pmatrix} E_2 & 0 \\ 0 & \frac{\mu_d^3}{\mu_u^3} E_2 \end{pmatrix} = \mu_u^3 \left( \Sigma_0^{\text{deg.}} + \Delta\Sigma_0 \right)_{ij} = \mu_u^3 (\Sigma_0)_{ij}, \quad (68)$$

where $\Sigma_0^{\text{deg.}} = \tilde{E}$ is the vacuum alignment for $m_u = m_d$; for concreteness we have chosen to pull out a factor of $\mu_u^3$ which shall serve as overall normalization. Switching to the (anti-)block-diagonal basis given in (A.12) and denoting the change of $\Sigma_0$ in presence of different chiral condensates as $\Delta\Sigma_0$ we arrive at,

$$\Delta\Sigma_0 = \begin{pmatrix} 0 & 0 \\ 0 & \frac{\mu_d^3 - \mu_u^3}{\mu_u^3} E_2 \end{pmatrix}. \qquad (69)$$

For the chiral Lagrangian for non-degenerate fermion masses in a generalization of (22) we may take the ansatz,

$$\mathcal{L} = \mathcal{L}_{\text{kin}} + \mathcal{L}_{\text{mass}} = \frac{f_\pi^2}{4} \text{Tr} \left[ \partial_\mu \Sigma \partial^\mu \Sigma^\dagger \right] - \frac{1}{2} \mu_u^3 \left( \text{Tr}[M\Sigma] + \text{h.c.} \right). \qquad (70)$$

The chiral field is now parameterized as follows,

$$\Sigma = \exp\left[\frac{i}{f_\pi}\sum_{k=1,2,4,5}\pi_k\tilde{T}^k + \frac{i}{f_{\pi_3}}\pi_3\tilde{T}^3\right]\Sigma_0\exp\left[\frac{i}{f_{\pi_3}}\pi_3\tilde{T}^3 + \frac{i}{f_\pi}\sum_{k=1,2,4,5}\pi_k\left(\tilde{T}^k\right)^T\right]. \quad (71)$$

In order to obtain canonical kinetic terms, a rescaling of the fields is necessary,

$$\pi_k \to \frac{2\mu_u^3}{\mu_u^3+\mu_d^3}\pi_k \ (k\neq 3), \quad \pi_3 \to \frac{f_{\pi_3}}{f_\pi}\frac{\sqrt{2}\mu_u^3}{\sqrt{\mu_u^6+\mu_d^6}}\pi_3. \quad (72)$$

In terms of the rescaled fields according to (72) and the chiral field (71) expanding (70) we may write the Lagrangian in terms of its mass-degenerate form $\mathcal{L}_{\text{kin}}^{\text{deg.}}+\mathcal{L}_{\text{mass}}^{\text{deg.}}$ plus additional terms that will vanish once the mass-degenerate limit $m_u = m_d$ is taken,

$$\begin{aligned}
\mathcal{L} = &\ \mathcal{L}_{\text{kin}}^{\text{deg.}} + \mathcal{L}_{\text{mass}}^{\text{deg.}} + \frac{m_\pi^2-m_{\pi_3}^2}{2}\pi_3^2 \\
&+ \frac{1}{12f_\pi^2}\frac{\left(\mu_d^6+2\mu_u^3\mu_d^3-3\mu_d^6\right)}{(\mu_d^3+\mu_u^3)^2}\sum_{k,n=1}^5\left(\pi_k\pi_k\partial_\mu\pi_n\partial^\mu\pi_n - \pi_k\partial_\mu\pi_k\pi_n\partial^\mu\pi_n\right) \\
&+ \frac{1}{8f_\pi^2}\frac{\mu_u^6(\mu_d^3-\mu_u^3)^2}{(\mu_d^3+\mu_u^3)^2(\mu_d^6+\mu_u^6)}\sum_{\substack{k=1\\k\neq3}}^5\left(\partial_\mu\pi_3\pi_k - \pi_3\partial_\mu\pi_k\right)\left(\partial^\mu\pi_3\pi_k - \pi_3\partial^\mu\pi_k\right) \\
&- \frac{1}{6f_\pi^2}\frac{\mu_u^6(\mu_d^3-\mu_u^3)^2}{(\mu_d^3+\mu_u^3)^2(\mu_d^6+\mu_u^6)}\sum_{\substack{k=1\\k\neq3}}^5\pi_3\partial_\mu\pi_k\left(\pi_k\partial^\mu\pi_3 - \pi_3\partial^\mu\pi_k\right) \\
&- \frac{\left(\mu_d^6+2\mu_u^3\mu_d^3-3\mu_d^6\right)m_\pi^2}{48f_\pi^2(\mu_d^3+\mu_u^3)^2}\left(\sum_{k=1}^5\pi_k^2\right)^2 - \frac{\mu_u^6\left((\mu_d^3-\mu_u^3)^2m_\pi^2+2\left(\mu_u^6+\mu_d^6\right)(m_\pi^2-m_{\pi_3}^2)\right)}{24f_\pi^2(\mu_d^3+\mu_u^3)^2\left(\mu_u^6+\mu_d^6\right)}\pi_3^2\left(\sum_{n=1}^5\pi_n^2\right) \\
&+ \frac{\mu_u^6\left(\mu_d^3-\mu_u^3\right)^2\left(m_\pi^2-m_{\pi_3}^2\right)}{24f_\pi^2(\mu_d^3+\mu_u^3)^2\left(\mu_u^6+\mu_d^6\right)}\pi_3^4 + \mathcal{O}\left(\pi^6\right).
\end{aligned} \quad (73)$$

Here, $\mathcal{L}_{\text{kin}}^{\text{deg.}}, \mathcal{L}_{\text{mass}}^{\text{deg.}}$ are found in (23) and (24) with $f_\pi$ as on the R.H.S. of (67) and $m_\pi$ defined through

$$m_\pi^2 \equiv m_{\pi_{1,2,4,5}}^2 = \frac{2\mu_u^6(m_u+m_d)}{f_\pi^2(\mu_u^3+\mu_d^3)}, \quad m_{\pi_3}^2 = \frac{2\mu_u^6(m_u\mu_u^3+m_d\mu_d^3)}{f_\pi^2(\mu_u^6+\mu_d^6)}. \quad (74)$$

As can be seen, whereas $\pi_{1,2,4,5}$ remain degenerate, $\pi_3$ is now split from the other states. In other words, $\pi_{1,2,4,5}$ form a multiplet and $\pi_3$ transforms as a singlet under the symmetry $SU(2)_u \times SU(2)_d$. It should be noted, that such mass-difference only appears for $\mu_d \neq \mu_u$ and the introduction of different chiral condensates was necessary to induce such splitting. Similarly to QCD, in leading order in the chiral expansion, the Goldstone masses do not depend on the difference in decay constants $f_\pi \neq f_{\pi_3}$. Of course, once $m_u = m_d$, the Goldstone spectrum becomes degenerate and the global flavour symmetry $Sp(4)$ is restored.

## 5.2 WZW Lagrangian in the non-degenerate case

Next, we consider the WZW term for the PNGBs for *non-degenerate* fermion masses. Due to the correction factor in $\Sigma_0$ in (68) also the five point interaction is modified. In its compact form, the WZW term may now be written as,

$$\mathcal{L}_{\text{WZW}} = -\frac{N_c}{240\pi^2 f_\pi^5}\varepsilon^{\mu\nu\rho\sigma}\text{Tr}\left[\mathcal{A}_\Sigma\partial_\mu\mathcal{A}_\Sigma\partial_\nu\mathcal{A}_\Sigma\partial_\rho\mathcal{A}_\Sigma\partial_\sigma\mathcal{A}_\Sigma,\right], \quad (75)$$

where $\mathcal{A}_\Sigma := \Sigma_0 \mathcal{A} = (E\mathcal{A} + \Delta\Sigma_0\mathcal{A})$, $\mathcal{A} := \left(\hat{\pi}\Sigma_0 + \Sigma_0\hat{\pi}^T\right)$ and

$$\hat{\pi} := \frac{2\mu_u^3}{\left(\mu_u^3 + \mu_d^3\right)} \sum_{k=1,2,4,5} \pi_k \tilde{T}^k + \frac{\sqrt{2}\mu_u^3}{\sqrt{\left(\mu_u^6 + \mu_d^6\right)}} \pi_3 \tilde{T}^3 \,.$$

Here, $\pi$ has already been used in its rescaled form according to (72). Expanding the Lagrangian as usual, we find that the effects due to non-degeneracy yield a modified pre-factor multiplied onto the Lagrangian that is equivalent to the degenerate case $\mathcal{L}_{\text{WZW}}^{\text{deg.}}$ with $f_\pi$ as on the R.H.S. of (67),

$$\mathcal{L}_{\text{WZW}} = \frac{\mu_d^6 \sqrt{\mu_u^6 + \mu_d^6}}{\sqrt{2}\mu_u^9} \mathcal{L}_{\text{WZW}}^{\text{deg.}} \,. \tag{76}$$

The prefactor is larger (smaller) than unity for $\mu_u < \mu_d$ ($\mu_u > \mu_d$.) The total Lagrangian including the WZW interaction is then given by the sum of (73) and (75).

## 5.3 Goldstone mass spectrum through partially conserved currents

We may also obtain the mass difference of the pseudo-Goldstones from the chiral Lagrangian (70) by means of current algebra instead of direct computation. We switch again to basis of generators given in Eq. (A.12), but all following quantities are also valid in the basis (A.6), replacing $\tilde{T}^a$ by $T^a$. Recall that under a $SU(4)$ transformation, the chiral field infinitesimally transforms as

$$\Sigma \to U\Sigma U^T \simeq \Sigma + i\theta_a \left[\tilde{T}^a\Sigma + \Sigma\left(\tilde{T}^a\right)^T\right] \,. \tag{77}$$

Before we derive the Goldstone masses, for completeness, we begin in the *massless* theory and establish some basic facts. The Noether currents associated with the $SU(4)$ flavour symmetry read

$$J_a^\mu\big|_{M=0} = i\frac{f_\pi^2}{2}\text{Tr}\left[\partial^\mu\Sigma^\dagger\left(\tilde{T}^a\Sigma + \Sigma\left(\tilde{T}^a\right)^T\right)\right] \equiv \begin{cases} i\frac{f_\pi^2}{2}\text{Tr}\left[\tilde{T}^a\left\{\Sigma, \partial^\mu\Sigma^\dagger\right\}\right] & a = 1,3,5 \\ i\frac{f_\pi^2}{2}\text{Tr}\left[\tilde{T}^a\left[\Sigma, \partial^\mu\Sigma^\dagger\right]\right] & a = 2,4 \\ 0 & a = 6,\dots,15 \,. \end{cases} \tag{78}$$

Hence, the currents are non-vanishing for generators associated with the Goldstone bosons. Under ordinary parity $P$ (see Eq. (14)), the currents associated with $a = 1,3,5$ and $a = 2,4$ are axial-vector and vector-currents, respectively: $PJ_{1,3,5}^\mu\big|_{M=0}(\vec{x}, t) = -J_{1,3,5}^\mu\big|_{M=0}(-\vec{x}, t)$ and $PJ_{2,4}^\mu\big|_{M=0}(\vec{x}, t) = J_{2,4}^\mu\big|_{M=0}(-\vec{x}, t)$. However, under the generalized parity $D$, all Goldstone bosons are pseudoscalars and the currents associated with $a = 1,\dots,5$ are indeed axial-vector currents, $DJ_a^\mu\big|_{M=0}(\vec{x}, t) = -J_a^\mu\big|_{M=0}(-\vec{x}, t)$. To leading order we have,

$$J_a^\mu\big|_{M=0} \simeq f_\pi\partial^\mu\pi_a, \quad a = 1,\dots,5 \,. \tag{79}$$

As is evident, in the massless theory, these currents are conserved, $\partial_\mu J_a^\mu\big|_{M=0} = 0$, implying the masslessness of the Goldstone bosons. We now turn to the massive theory, $M \neq 0$, and establish the mass spectrum. To this end, we first note that the change of the Lagrangian upon the $SU(4)$ transformation in (77) reads,

$$\mathcal{L} \to \mathcal{L} - \frac{i\mu_u^3}{2}\theta_a \left\{\text{Tr}\left[\tilde{T}^a\left(\Sigma M - M^\dagger\Sigma^\dagger\right)\right] + \text{Tr}\left[\left(\tilde{T}^a\right)^T\left(M\Sigma - \Sigma^\dagger M^\dagger\right)\right]\right\} \,. \tag{80}$$

From the general definition of the Noether current $J_a^\mu$, one may use the above expression to identify the non-conserved currents for the generators associated with the Goldstone bosons,

$$
\begin{aligned}
\partial_\mu J_a^\mu &= \frac{i\mu_u^3}{2} \left\{ \mathrm{Tr}\left[ \tilde{T}^a \left( \Sigma M - M^\dagger \Sigma^\dagger \right) \right] + \mathrm{Tr}\left[ \left( \tilde{T}^a \right)^T \left( M\Sigma - \Sigma^\dagger M^\dagger \right) \right] \right\} \\
&= \frac{i\mu_u^3}{2} \times \begin{cases} \mathrm{Tr}\left[ \tilde{T}^a \left( \{\Sigma, M\} - \{M^\dagger, \Sigma^\dagger\} \right) \right] & \text{for } \left( \tilde{T}^a \right)^T = \tilde{T}^a, \, a = 1, 3, 5 \\ \mathrm{Tr}\left[ \tilde{T}^a \left( [\Sigma, M] - [M^\dagger, \Sigma^\dagger] \right) \right] & \text{for } \left( \tilde{T}^a \right)^T = -\tilde{T}^a, \, a = 2, 4 \\ 0 & \text{for } a = 6, \dots, 15 . \end{cases}
\end{aligned}
\tag{81}
$$

We may now use the method of *partially conserved axial currents* which posits,

$$
\partial_\mu J_a^\mu \left| 0 \right\rangle \simeq m_{\pi_a}^2 f_{\pi_a} \left| \pi_a(p) \right\rangle ,
\tag{82}
$$

and compare the masses on the right-hand-side with the ones obtained by the direct evaluation of the traces in (81). Accounting for the rescaling in (72), we obtain

$$
\begin{aligned}
\partial_\mu J_k^\mu \left| 0 \right\rangle &\simeq \frac{2\mu_u^6 (m_u + m_d)}{f_\pi (\mu_u^3 + \mu_d^3)} \left| \pi_k(p) \right\rangle , \quad k = 1, 2, 4, 5 , \\
\partial_\mu J_3^\mu \left| 0 \right\rangle &\simeq \frac{2\mu_u^6 (m_u \mu_u^3 + m_d \mu_d^3)}{f_\pi (\mu_u^6 + \mu_d^6)} \left| \pi_3(p) \right\rangle . \\
\partial_\mu J_n^\mu \left| 0 \right\rangle &= 0, \quad n = 6, \dots, 15 ,
\end{aligned}
\tag{83}
$$

The comparison with (82) shows that we recover the masses previously found in (74).

## 5.4 The pseudoscalar singlet in the non-degenerate case

Finally, we work out the role of $\eta'$ in the non-degenerate fermion mass case. For this, we again perform an extra rotation on the chiral field in Eq. (71),

$$
\Sigma \to \exp\left[ \frac{2i}{f_{\eta'}} \eta' T^0 \right] \Sigma .
\tag{84}
$$

Then, the kinetic term in the chiral Lagrangian up to fourth order in the fields is given by

$$
\mathcal{L}_\mathrm{kin} = \frac{1}{2} \sum_{k \neq 3} \partial_\mu \pi_k \partial^\mu \pi_k + \frac{1}{2} \begin{pmatrix} \partial_\mu \pi_3 & \partial_\mu \eta' \end{pmatrix} K \begin{pmatrix} \partial^\mu \pi_3 \\ \partial^\mu \eta' \end{pmatrix} + O(\pi^4) ,
\tag{85}
$$

where we have used the redefinition of the fields following (72) and, additionally rescaled $\eta'$ as,

$$
\eta' \to \frac{f_{\eta'}}{f_\pi} \frac{\sqrt{2}\mu_u^3}{\sqrt{\mu_u^6 + \mu_d^6}} \eta' .
\tag{86}
$$

As can be seen, in the non-degenerate case, a kinetic mixing between $\pi_3$ and $\eta'$ is induced by the off-diagonal elements of $K$,

$$
K = \begin{pmatrix} 1 & \frac{(\mu_u^6 - \mu_d^6)}{2(\mu_u^6 + \mu_d^6)} \\ \frac{(\mu_u^6 - \mu_d^6)}{2(\mu_u^6 + \mu_d^6)} & 1 \end{pmatrix} .
\tag{87}
$$

We put the kinetic term into canonical form in terms of the diagonal fields,

$$
\begin{pmatrix} \hat{\pi}_3 \\ \hat{\eta}' \end{pmatrix} = S^{1/2} O_\mathrm{kin}^T \begin{pmatrix} \pi_3 \\ \eta' \end{pmatrix} ,
\tag{88}
$$

where $O_{\text{kin}}$ is the orthogonal matrix, that diagonalizes $K$ (mixing angle $\pi/4$) and $S$ is composed of the eigenvalues of $K$ in the diagonal, fulfilling $S = O_{\text{kin}}^T K O_{\text{kin}}$. The matrix $S$ is given by

$$S = \frac{1}{2(\mu_d^6 + \mu_u^6)} \begin{pmatrix} 3\mu_d^6 + \mu_u^6 & 0 \\ 0 & \mu_d^6 + 3\mu_u^6 \end{pmatrix}. \tag{89}$$

In the new basis Eq. (88), the kinetic terms are then diagonal and canonically normalized.

We now turn to the mass term in the chiral Lagrangian after rescaling of the fields (72) and (86),

$$\mathcal{L}_{\text{mass}} = 2(m_u\mu_u^3 + m_d\mu_d^3) - \frac{\mu_u^6(m_u + m_d)}{f_\pi^2(\mu_u^3 + \mu_d^3)} \sum_{k\neq3} \pi_k^2 - \frac{1}{2}\begin{pmatrix}\pi_3 & \eta'\end{pmatrix} M_{\pi_3\eta'}^2 \begin{pmatrix}\pi_3 \\ \eta'\end{pmatrix}, \tag{90}$$

where the squared mass matrix of $\pi_3$ and $\eta'$ takes the form

$$M_{\pi_3\eta'}^2 = \frac{\mu_u^6}{f_\pi^2(\mu_u^6 + \mu_d^6)} \begin{pmatrix} 2(m_u\mu_u^3 + m_d\mu_d^3) & 2(m_u\mu_u^3 - m_d\mu_d^3) \\ 2(m_u\mu_u^3 - m_d\mu_d^3) & 2(m_u\mu_u^3 + m_d\mu_d^3) + 2\frac{\mu_d^6}{\mu_u^6}\Delta m_{\eta'}^2 f_{\eta'}^2 \end{pmatrix}. \tag{91}$$

We see that in addition to kinetic mixing there is also a mass mixing. We recall that $\Delta m_{\eta'}^2$ was introduced as an explicit breaking of the $U(1)_A$ symmetry. Switching to the new basis (88), the mass term is then diagonalized by the transformation,

$$\begin{pmatrix}\tilde{\pi}_3 \\ \tilde{\eta}'\end{pmatrix} = O_{\text{mass}}^T \begin{pmatrix}\hat{\pi}_3 \\ \hat{\eta}'\end{pmatrix}, \quad O_{\text{mass}} = \begin{pmatrix} \cos\theta_{\text{mass}} & \sin\theta_{\text{mass}} \\ -\sin\theta_{\text{mass}} & \cos\theta_{\text{mass}} \end{pmatrix}, \tag{92}$$

with the mixing angle $\theta_{\text{mass}}$ given by

$$\tan 2\theta_{\text{mass}} = \frac{f_{\eta'}^2 \Delta m_{\eta'}^2 \sqrt{(\mu_d^6 + 3\mu_u^6)(3\mu_d^6 + \mu_u^6)}}{f_{\eta'}^2 \Delta m_{\eta'}^2(\mu_d^6 - \mu_u^6) - 2x_-}, \tag{93}$$

where $x_\pm := \frac{\mu_u^6}{\mu_d^6}\left((m_d\mu_d^3(\mu_d^6 + 3\mu_u^6) \pm m_u\mu_u^3(3\mu_d^6 + \mu_u^6)\right)$. For $\Delta m_{\eta'} = 0$, the squared mass matrix is automatically diagonal ($\theta_{\text{mass}} = 0$) after diagonalization of the kinetic term. Likewise, in the mass-degenerate case, no kinetic- and mass-mixing is induced even for $\Delta m_{\eta'} \neq 0$. The total Lagrangian in the new basis is given by,

$$\begin{aligned}\mathcal{L} = \mathcal{L}_{\text{kin}} + \mathcal{L}_{\text{mass}} &= \frac{1}{2}\sum_{k\neq3}\partial_\mu\pi_k\partial^\mu\pi_k + \frac{1}{2}\partial_\mu\tilde{\pi}_3\partial^\mu\tilde{\pi}_3 + \frac{1}{2}\partial_\mu\tilde{\eta}'\partial^\mu\tilde{\eta}' \\ &+ 2(m_u\mu_u^3 + m_d\mu_d^3) - \frac{m_\pi^2}{2}\sum_{k\neq3}\pi_k^2 - \frac{m_{\pi_3}^2}{2}\tilde{\pi}_3^2 - \frac{m_{\eta'}^2}{2}\tilde{\eta}'^2 + O(\tilde{\pi}^4),\end{aligned} \tag{94}$$

with masses

$$m_\pi^2 \equiv m_{\pi_{1,2,4,5}}^2 = \frac{2\mu_u^6(m_u + m_d)}{f_\pi^2(\mu_u^3 + \mu_d^3)},$$

$$m_{\pi_3}^2 = \frac{4\mu_d^6}{f_\pi^2}\left(\frac{f_{\eta'}^2\Delta m_{\eta'}^2(\mu_d^6 + \mu_u^6) + x_+}{(3\mu_d^6 + \mu_u^6)(\mu_d^6 + 3\mu_u^6)} - \frac{\sqrt{f_{\eta'}^4\Delta m_{\eta'}^4(\mu_d^6 + \mu_u^6)^2 + x_-^2 - f_{\eta'}^2\Delta m_{\eta'}^2(\mu_d^6 - \mu_u^6)x_-}}{(3\mu_d^6 + \mu_u^6)(\mu_d^6 + 3\mu_u^6)}\right), \tag{95}$$

$$m_{\eta'}^2 = \frac{4\mu_d^6}{f_\pi^2}\left(\frac{f_{\eta'}^2\Delta m_{\eta'}^2(\mu_d^6 + \mu_u^6) + x_+}{(3\mu_d^6 + \mu_u^6)(\mu_d^6 + 3\mu_u^6)} + \frac{\sqrt{f_{\eta'}^4\Delta m_{\eta'}^4(\mu_d^6 + \mu_u^6)^2 + x_-^2 - f_{\eta'}^2\Delta m_{\eta'}^2(\mu_d^6 - \mu_u^6)x_-}}{(3\mu_d^6 + \mu_u^6)(\mu_d^6 + 3\mu_u^6)}\right).$$

As expected, the pseudoscalar $\eta'$ has a larger mass than $\pi_3$ and remains massive even in the chiral limit for $\Delta m_{\eta'} \neq 0$. The Goldstone and $\eta'$ spectrum become degenerate for $\Delta m_{\eta'} = 0$, once $m_u = m_d, \mu_u = \mu_d$. For $\Delta m_{\eta'} = 0$ the masses reduce to

$$m_{\pi_3}^2 = \frac{8\mu_u^9 m_u}{f_\pi^2 \left(\mu_d^6 + 3\mu_u^6\right)}, \quad m_{\eta'}^2 = \frac{8\mu_u^6 \mu_d^3 m_d}{f_\pi^2 \left(3\mu_d^6 + \mu_u^6\right)}. \tag{96}$$

In Sec. 3.2 we showed that at leading order $\eta'$ does not appear in the WZW term in the case of degenerate fermion masses. In case of *non-degenerate* fermion masses, the iso-singlet $\eta'$ enters the WZW interaction through the mixing effects between $\eta'$ and $\pi_3$. In the basis in which kinetic and mass terms are diagonal, we obtain,

$$\mathcal{L}_{\text{WZW}} = \frac{\mu_d^6 \left(\mu_u^6 + \mu_d^6\right)}{\sqrt{2}\mu_u^9} \left[ \left( \frac{\cos\theta_{\text{mass}}}{\sqrt{3\mu_d^6 + \mu_u^6}} - \frac{\sin\theta_{\text{mass}}}{\sqrt{\mu_d^6 + 3\mu_u^6}} \right) \mathcal{L}_{\text{WZW}}^{\text{deg.},\tilde{\pi}} + \left( \frac{\cos\theta_{\text{mass}}}{\sqrt{\mu_d^6 + 3\mu_u^6}} - \frac{\sin\theta_{\text{mass}}}{\sqrt{3\mu_d^6 + \mu_u^6}} \right) \mathcal{L}_{\text{WZW}}^{\text{deg.},\tilde{\eta}'} \right], \tag{97}$$

Here, $\mathcal{L}_{\text{WZW}}^{\text{deg.},\tilde{\pi}}$ is given in (76) where $\pi_3$ is replaced by $\tilde{\pi}_3$ and $\mathcal{L}_{\text{WZW}}^{\text{deg.},\tilde{\eta}'}$ takes the same form as $\mathcal{L}_{\text{WZW}}^{\text{deg.},\pi}$ in (76) with $\pi_3$ being replaced by $\tilde{\eta}'$.

# 6 Lattice results on $Sp(4)_c$ with $N_f = 1 + 1$

## 6.1 Free parameters of the microscopic Lagrangian

The chiral Lagrangian has several low-energy constants that cannot be obtained from the effective theory itself. These can be determined from the underlying UV-complete theory. As the UV theory is strongly interacting, this requires a non-perturbative method, for which we choose lattice here. This allows us to follow standard procedure for the task at hand [4]. Any potential influence of gauging under $U(1)'$ as discussed above is expected to be small, and can thus be accounted for by using perturbation theory *a posteriori*. A similar situation arises in the SM weak interactions of mesons, where the same approximation also holds very well [85].

In the following, we thus consider the strongly interacting $Sp(4)_c$ gauge theory with two fundamental Dirac fermions in isolation. The mass degenerate case was already studied in [6] and pioneering studies in $Sp(4)_c$ Yang-Mills theory and the quenched theory have been carried out in [7, 86, 87]. Since we use the $Sp(4)_c$ branch of the HiRep code [88] developed in [6, 7] for the simulations, we use the same technical framework. We briefly repeat the pertinent details in App. D.

The theory has three free parameters, the gauge coupling $g$ and the two bare fermion masses $m_u$ and $m_d$. In the context of lattice calculations, it is convenient to express the gauge coupling as $\beta = 8/g^2$. Note that both the coupling and the fermion masses are the unrenormalized bare parameters and thus unphysical.

In the continuum theory, the overall scale would be set by one of the dimensionful parameters, but in a lattice calculation it is convenient to use instead the finite lattice spacing $a$. Masses are then measured as a multiple of the inverse lattice spacing $a^{-1}$ and we report here the dimensionless products $am$ in the following. Only once some dimensionful quantity is fixed, e.g., by experimental input, explicit units become possible. Fixing the scale, and thus the lattice spacing, implies that also one of the bare lattice parameters is fixed. It is convenient to choose the gauge coupling for this fixing of the scale, leaving two dimensionless quark masses to uniquely characterize the physics. These two free parameters can be used to fix two observable quantities, e. g. properties of the dark hadrons such as masses or scattering cross-sections. All other results are then fixed.

Since a tractable way of deriving these quantities from observations requires a treatment using the effective field theory, we can only see *a posteriori* which input parameters of the microscopic theory (if any) provide viable Dark Matter candidates. In addition, we need to ensure that the effective theory is a sufficiently controlled approximation to the underlying UV theory. This will be done by comparing predictions of the EFT to first-principles results from the lattice.

Remaining agnostic about the values of the two fundamental dark quark masses, we study different combinations of them. We always start from degenerate dark quark masses, and we then incrementally increase one of them, breaking the flavour symmetry from $Sp(4)$ down to $SU(2)_u \times SU(2)_d$ explicitly. For small breaking and sufficiently light quarks, $Sp(4)$ should still be an approximate symmetry and we expect to see 5 relatively (compared to the other meson masses) light pseudo-Goldstone states of which 4 will remain degenerate. For larger breaking this will at some point no longer be the case and the system is expected to resemble a heavy-light system. For one extremely heavy dark quark we expect to see the pattern [34, 60] of a corresponding one-flavour theory for the lightest states. Since we vary the value of one of the bare quark masses to study a one-dimensional subspace of the three-dimensional parameter space and we already leave an overall scale undetermined we only have to fix one remaining bare quark mass through a suitable observable. Once this is done all other lattice observables are predictions

In Fig. 5 we depict the previously outlined workflow. We emphasize that all bare input parameters are unrenormalized and thus unphysical (this includes the bare quark masses).[10]

We fix the remaining bare quark mass (which remains unchanged when we break the flavour symmetry) through the ratio of the vector meson masses to (one of) the pseudo-Goldstones $m_\rho/m_\pi$ at degeneracy. In the chiral limit, the Goldstone modes become massless and the ratio diverges, $m_\rho/m_\pi \to \infty$. In the limit of extremely heavy quarks all meson masses are dominated by the mass of the valence quarks and the ratio approaches unity, $m_\rho/m_\pi \to 1$. Thus, the larger this ratio becomes the closer we are to the chiral limit. In the Standard Model the experimental value of this is $m_\rho/m_\pi \approx 5.5 - 5.7$ and for the pseudo-Goldstone bosons of a three-flavour theory the $K$ and $\eta$ mesons, they are $m_\rho/m_K \approx 1.5$ and $m_\rho/m_\eta \approx 1.4$.

In this work we have studied ensembles where this ratio in the mass degenerate limits[11] are $m_\rho/m_\pi \approx 1.15, 1.25$ and $1.4$. These values are slightly smaller than those suggested by existing phenomenological investigations of such theories as dark matter candidates [35]. Eventually, we also want to study ensembles with $m_\rho/m_\pi > 1.4$. However, they come at a significantly increased computational cost and we defer this to future work. We conclude, that our quark masses relate to the intrinsic scale of our theory similarly as the QCD strange quark mass relates to the QCD scale as the mass ratio is similar. Note that in all cases the aforementioned ratio is smaller than 2 and the $\rho$ at rest cannot decay into two Goldstone bosons.

## 6.2 Quark masses and partially conserved axial current

In the previous section we discussed how to choose the unrenormalized bare quark masses for our lattice simulations. We stressed that these input masses are unrenormalized and thus regulator-dependent and unphysical. An obvious question is, therefore, if there is a way of calculating a "physical" quark mass. This is however *not* possible. Due to confinement there is no notion of a physical quark mass since no physical quark has ever been observed in experiment. Any definition of a quark mass is scheme-dependent and necessarily not unique. In

---

[10]More precisely, the lattice spacing acts as an ultraviolet cutoff $\Lambda = a^{-1}$, and the bare parameters are the ones at the cutoff. Thus, in the formal continuum limit of $a \to 0$, they will be either zero or infinite.

[11]For the degenerate case, if possible, we give results from [6], since these have been obtained on larger lattices with better statistics in comparison to ours. This will be indicated where necessary.

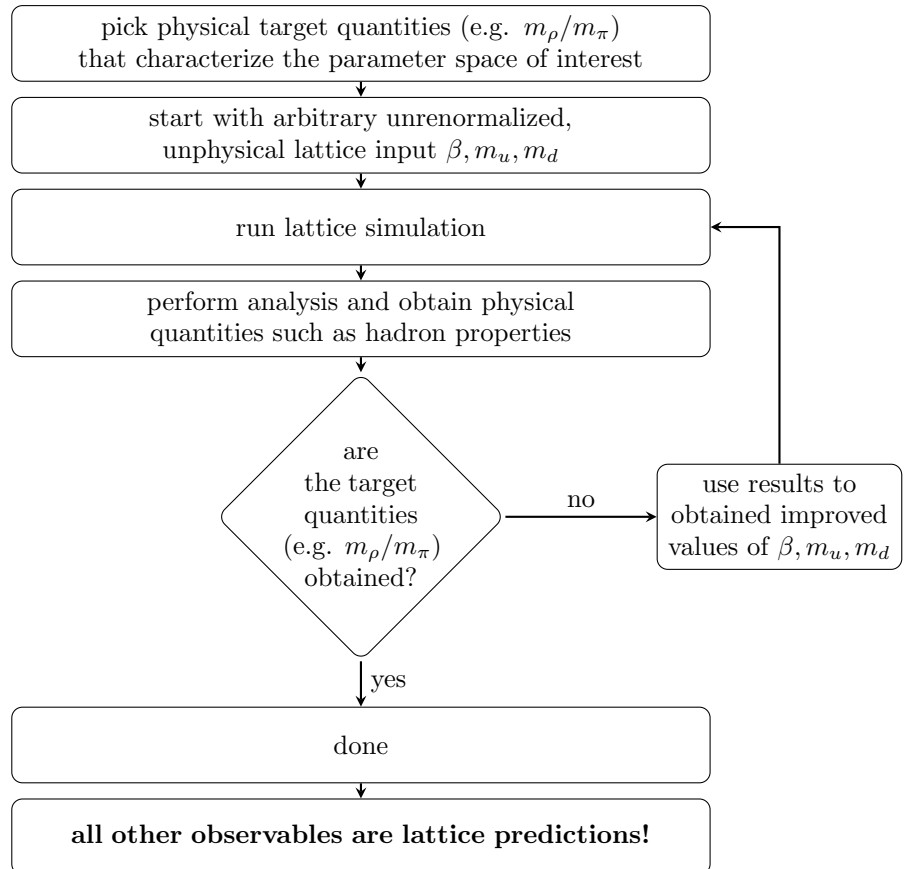

Figure 5: Workflow for choosing suitable input parameters $m_u, m_d$ and $\beta$. These input parameters are unrenormalized and thus unphysical. They are chosen such that a set of observables has a prescribed value (in our case the ratio $m_\rho/m_\pi$). All other observables are then predictions from the lattice. Note that we do not fix the overall scale of our observables in our analysis.

the context of Standard Model QCD several ways of a defining a quark mass are being actively used. See for example the current PDG review [89] for a detailed discussion of quark masses in the SM.[12]

The scheme-dependence of any quark mass implies that quark masses are comparable only in the same scheme at the same scale $\mu$. This implies that we need to determine the renormalization constants of the bare quark masses $m_u$ and $m_d$ in order to obtain the renormalized quark masses $m_u^{(r)}$ and $m_d^{(r)}$. Our discretization of the fermion fields on the lattice makes this even more challenging: The Wilson discretization of fermions breaks chiral symmetry explicitly on the lattice and results in both a multiplicative *and* additive renormalization of the bare quark mass as long as the lattice is finite, i.e. $a > 0$ [4].

We can, however, instead define a quark mass based on the *partially conserved axial current* (PCAC) equation from (82) which relates the axial current to the pion field and the *axial Ward identity* (AWI) which connects the axial current to the renormalized quark mass $m^{(r)}$. We can restrict ourselves to the AWI for the $\pi^A$ pseudo-Goldstone which reads $\partial_\mu J_A^\mu = (m_u^{(r)} + m_d^{(r)})\bar{d}\gamma_5 u$. This entails that we can also define an unrenormalized quark mass through correlation functions $C_\Gamma(t)$ of the unrenormalized axial currents $\bar{u}\gamma_0\gamma_5 d$ and $\bar{u}\gamma_5 d$

---

[12]In fact, beyond perturbation theory the situation is even worse, and it is not settled if a concept like a quark mass can be defined in a confining theory at all. Especially, the quark propagator has not necessarily a suitable pole structure, see e. g. [90] for a discussion.

(see e.g. [91] for a detailed discussion and other equivalent definitions),

$$m^{\text{PCAC}} = \lim_{t\to\infty} \frac{1}{2} \frac{\partial_t C_{\gamma_0\gamma_5,\gamma_5}(t)}{C_{\gamma_5}(t)} = \lim_{t\to\infty} \frac{1}{2} \frac{\partial_t \int d^3\vec{x} \langle (\bar{u}(\vec{x},t)\gamma_0\gamma_5 d(\vec{x},t))^\dagger \bar{u}(0)\gamma_5 d(0)\rangle}{\int d^3\vec{x} \langle (\bar{u}(\vec{x},t)\gamma_5 d(\vec{x},t))^\dagger \bar{u}(0)\gamma_5 d(0)\rangle}. \tag{98}$$

At large times $t$ the ratio of the two correlation functions in eq. (98) tends to a constant which we identify as the PCAC-mass. In the mass-non-degenerate case this expression gives the average mass of up-type quarks and down-type quarks. In our setup we try to keep one of the quark masses fixed. This then allows us to determine the fixed mass directly in the degenerate calculations and deduce the other mass in the non-degenerate case.[13] The PCAC-mass is related to the renormalized (average) mass by a multiplicative factor

$$m^{(r)} = \frac{Z_A}{Z_P} m^{\text{PCAC}}, \tag{99}$$

and is therefore an unrenormalized quantity. In order to obtain the renormalized mass the factor $Z_A/Z_P$ needs to be determined and a scheme to be chosen. This is however quite involved and in addition a matching to other commonly used renormalization scheme is needed in order for this mass to be used in perturbative calculations (see e.g. [85]). We therefore skip this calculation in this work and point out that the renormalization factors cancel if we consider ratios of PCAC-masses such as $m_u^{\text{PCAC}}/m_d^{\text{PCAC}}$ since only multiplicative renormalization occurs.

Note that the PCAC relation is closely linked to chiral perturbation theory and the Gell-Mann-Oakes-Renner (GMOR) relation in particular - see section 5.3. Calculating the PCAC masses and the chiral condensate and comparing this to the GMOR relation cannot be considered to be a truly independent and quantitative test of chiral perturbation theory. We can still use it, however, as a qualitative test by examining the dependence of square of the Goldstone masses on the PCAC-masses and comparing the results to the expected linear behaviour. This can be found (although with a differently defined unrenormalized quark mass) in [6]. In this work we are primarily interested in the effects of strong isospin breaking. We will take the relation at mass-degeneracy for granted and using (74) we determine the (unrenormalized) chiral condensates. We will see that at a sufficiently large strong isospin breaking this will no longer be possible which might suggest that at this amount of strong isospin breaking the chiral Lagrangian at this order is not an adequate description of the Goldstone dynamics of the underlying theory. This is done in section 6.4.

## 6.3 Results: Masses, decay constants and quark masses

We have calculated the masses and decay constants of both the Goldstones and the vector mesons as well as the previously outlined unrenormalized PCAC masses. The lattice setup and the lattice action as well as the techniques used for extracting the masses and decay constants from the lattice can be found in appendix D. A detailed study of lattice systematics is given in appendix D.4. Here we present the results which we expect to give a suitable approximation of the continuum theory.

The masses for the ensembles with varying $m_\rho/m_\pi$ at degeneracy are shown in Fig. 6. We see that, both for the Goldstone and vector mesons, the flavour-neutral states are the lighter states once strong isospin breaking is introduced. This makes the $\pi^C$ the lightest state in the theory. The remaining 4 Goldstones are heavier and remain degenerate. At some point the 6 lighter vector mesons—among them the $\rho^N$—become even lighter than the heavier pseudoscalars.

---

[13]In principle, there is the possibility that changing one parameter can affect the other. We do not see any sign of this, but it would require further investigations to settle beyond doubt.

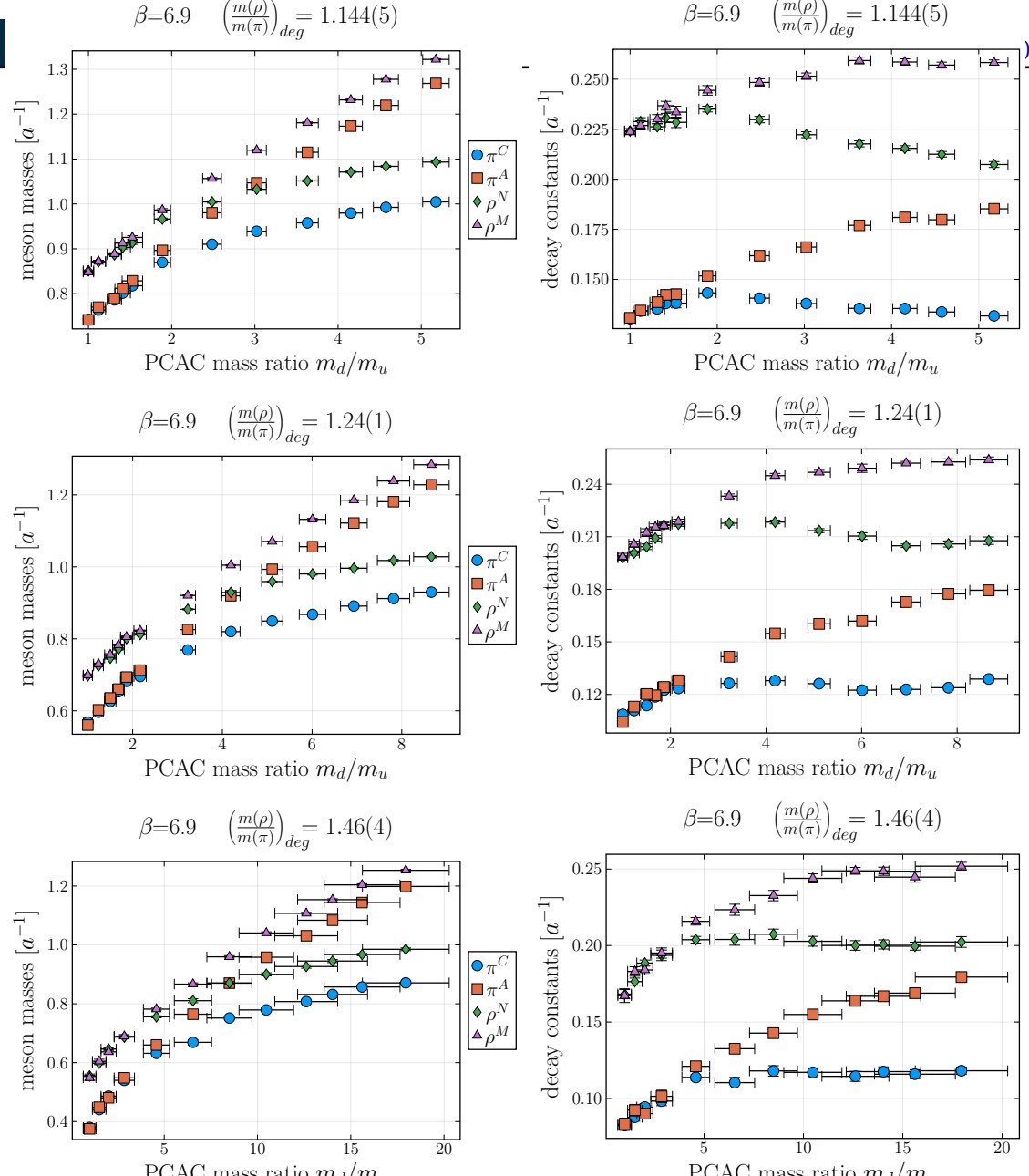

Figure 6: Masses and decay constants of the pseudo-Goldstone mesons and the vector mesons for different non-degenerate fermion masses against the ratio of the unrenormalized PCAC masses. One fermion mass is kept fixed while the other is incrementally increased. Therefore, all meson masses increase with larger fermionic mass difference. The unflavoured pseudo-Goldstone is the lightest particle in the spectrum of the isolated theory. For a larger mass difference between the fermions the unflavoured vector mesons get lighter than the flavoured pseudo-Goldstones. At this point the EFT based solely on would-be-Goldstones is certainly no longer adequate.

A similar pattern is observed for the decay constants of those dark mesons shown. The unflavoured decay constants are smaller than their flavoured counterparts. This is seen for both the Goldstone and the vector mesons. Furthermore, the unflavoured vector decay constant shows a maximum at intermediate values of the dark quark mass splitting.

Note that we observe more effects of the finite lattice extent for the lightest ensembles with $m_\rho/m_\pi \approx 1.25$ at degeneracy and in addition we see finite lattice spacing effects for heavy mesons for both $m_\rho/m_\pi \approx 1.4$ and $1.15$ at degeneracy. We again refer to appendix D.4 for a discussion of lattice systematics.

We conclude that for large mass splittings the system more closely resembles a heavy-light system where the unflavoured mesons are the lightest hadronic states. At some point the mass of the second dark quark is so heavy that it decouples and the low-energy part of this theory is effectively a $N_f = 1$ theory which contains three vector mesons and one (massive) pseudoscalar [34]. In this case the theory develops a hierarchy of scales.

## 6.4 Validity of the Chiral Lagrangian

The chiral theory of Secs. 3 to 5 is based on the dynamics of the lightest hadronic states, which in this case are the pseudo-Goldstone bosons $\pi$. In the chiral limit, i.e., in the limit of massless dark fermions, the flavour symmetry is broken only spontaneously and the $\pi$'s become massless themselves. Close to the chiral limit for degenerate quarks the GMOR relation gives the dependence of the product of Goldstone masses on the renormalized quark masses $m_q^{(r)}$ and the chiral condensate $(\mu^3)^{(r)}$

$$(f_\pi m_\pi)^2 = 2m^{(r)}(\mu^3)^{(r)}. \tag{100}$$

It can be seen that the square of the pseudo-Goldstone mass depends linearly on the average renormalized quark mass and in the chiral limit the equation is trivially fulfilled. As pointed out in section 6.2 the renormalized quark mass and the condensate are scheme-dependent. However, their product is scheme-independent since the involved renormalization constants cancel. Since we forwent a determination of the renormalization constants due to the technically involved nature of such a calculation we use the unrenormalized PCAC mass $m_{\mathrm{PCAC}}$ and define the chiral condensate through GMOR. This entails that also the condensate is then unrenormalized and regulator-dependent. We therefore drop the superscript $^{(r)}$ in the following. For non-degenerate quarks GMOR is given by equation (74). For convenience, we rewrite is as

$$\mathrm{GMOR}_{\pi^A}(m_u, m_d, \mu_u, \mu_d) = (m_{\pi^A} f_{\pi^A})^2 - \frac{2\mu_u^6(m_u + m_d)}{\mu_u^3 + \mu_d^3} = 0,$$

$$\mathrm{GMOR}_{\pi^C}(m_u, m_d, \mu_u, \mu_d) = (m_{\pi^C} f_{\pi^C})^2 - \frac{2\mu_u^6(m_u\mu_u^3 + m_d\mu_d^3)}{\mu_u^6 + \mu_d^6} = 0. \tag{101}$$

We have seen that for sufficiently large UV mass difference in the dark fermions, the mass hierarchy in the mesonic spectrum changes qualitatively: the multiplet of vector mesons containing the $\rho^N$ becomes lighter than the multiplet of flavoured pseudoscalars containing the $\pi^A$. The set of $\pi$'s are then no longer the lightest hadronic states and an inclusion of the relevant vector states becomes necessary. This provides an upper limit on the amount of strong isospin breaking.

Using the aforementioned GMOR relations we can set an even stronger bound on the amount of strong isospin breaking. The validity of the GMOR relation was already studied for degenerate fermions in [6]. It was found that dependence of the square of the pseudo-Goldstone mass on a (differently defined) unrenormalized quark mass is linear for ensembles with $m_\rho/m_\pi > 1.4$, and thus chiral effective theory can be expected to work adequately latest from there on. Due to the increased computational cost of non-degenerate fermions we do not have results on ensembles that fulfil $m_\rho/m_\pi \gg 1.4$. Nevertheless, we can perform consistency tests on the GMOR relations in (101) around the threshold $m_\rho/m_\pi \approx 1.4$, see table 3 below. We use the fact that in our simulations one bare quark mass has been kept fixed and proceed as follows:

At degenerate fermion masses we take the degenerate GMOR relation (100) for granted

Table 3: The point at which a change in the meson mass hierarchy occurs and the point at which the non-degenerate GMOR relation breaks down at one-sigma significance. In general, this breakdown sets a stronger bound. Note that with increased statistics the breakdown of the non-degenerate GMOR can occur at even smaller PCAC-mass-ratios.

| $\beta$ | $\left(\frac{m_\rho}{m_\pi}\right)_{\text{deg}}$ | $\left(\frac{m_u}{m_d}\right)$ where $m_{\rho^N} = m_{\pi^A}$ | $\left(\frac{m_u}{m_d}\right)$ where non-deg. GMOR breaks down |
|---|---|---|---|
| 6.9 | 1.144(5) | 2.8(3) | 3.0(2) |
| 6.9 | 1.25(1) | 4.4(4) | 1.5(1) |
| 6.9 | 1.46(4) | 8(1) | 1.8(5) |
| 7.05 | 1.16(1) | 2.7(3) | 2.2(2) |
| 7.05 | 1.29(2) | 4.7(5) | 1.7(2) |
| 7.05 | 1.46(4) | 6.8(8) | 1.7(2) |
| 7.2 | 1.17(1) | 2.7(4) | 4(1) |
| 7.2 | 1.26(2) | 4.4(7) | 1.7(4) |
| 7.2 | 1.37(4) | 6(2) | 4(2) |

and use it to determine the fixed chiral condensate $(\mu_u^3)^{\text{PCAC}}$ from the fixed quark mass $m_u^{\text{PCAC}}$.[14] Since these quantities are regulator-dependent we only compare results at the same value of the (bare) inverse gauge coupling $\beta$.[15] This determines three out of the four quantities that enter the non-degenerate GMOR relations in (101). We can then use both equations in (101) to determine $(\mu_d^3)^{\text{PCAC}}$. If the non-degenerate GMOR relation holds we expect the functions $\text{GMOR}_{\pi^{A,C}}(m_u, m_d, \mu_u, x)$ to have common roots at $x = \mu_d^{\text{PCAC}}$. Using our lattice data we find for ratios starting at $m_d/m_u \gtrsim 1.5$ that $\text{GMOR}_{\pi^A}$ and $\text{GMOR}_{\pi^C}$ cease to have a common root. We exemplify this in Fig. 7 for specific ensembles $\text{GMOR}_{\pi^{A,C}}(m_u, m_d, \mu_u, x)$ as a function of $x$. It can be seen that at larger quark-mass-ratios the functions do not share a common root. In particular $\text{GMOR}_{\pi^C}$ no longer has root in this region of $x$.

We interpret this as a sign that at this point the description provided by the leading order chiral Lagrangian that lead to (101) no longer captures the underlying theory. This sets another upper bound on the validity of the non-degenerate GMOR relation. We might therefore hope that as long as these functions have a common root, all pseudo-Goldstone are still the lightest mesons in the spectrum and the degenerate GMOR relation holds true and the leading order Lagrangian might be an adequate description of strong isospin breaking in this theory at fixed $m_\rho/m_\pi \gtrsim 1.4$ at degeneracy. The tabulated upper limits can be found in table 3. Note that this is only an *upper bound*.

There are two reasons why the non-degenerate GMOR relation can break down: 1) The quark-mass-difference is too large in order for the system to be treated at leading order. In this case the next step would be to investigate this chiral Lagrangian at next-to-leading order in strong isospin breaking. 2) The average pseudo-Goldstone masses are in general too large to reliably use GMOR at leading order even close to the mass-degenerate limit. If this is the case we can either go to next-to-leading order in chiral perturbation theory or study the system for even lighter pseudo-Goldstones (at significantly increased computational cost).

In any case, we have shown that for small isospin breaking the non-degenerate GMOR relations do not break down immediately even for significantly heavier quarks than those used in the chiral extrapolation of [6] and also in the case of the $(m_\rho/m_\pi)_{\text{deg}} \approx 1.4$ ensemble. It will

---

[14]From [6] we know that the dependence of the squared Goldstone mass on the quark mass only becomes linear for $m_\rho/m_\pi \gtrsim 1.4$. However, this approach allows us to test whether the explicit introduction of isospin breaking effects cause a breakdown.

[15]In principle the regulator depends also on the bare quark masses. Experience has shown that these effects are subleading compared to the effect of $\beta$.



Figure 7: GMOR$_{\pi^{A,C}}(m_u, m_d, \mu_u, x)$ as a function of $x$ for an inverse gauge coupling of $\beta = 6.9$ and values of $m_\rho/m_\pi \approx 1.24$ and $m_\rho/m_\pi \approx 1.46$ at degeneracy. If the non-degenerate GMOR relation holds the condensate $\mu_d$ is given by the common root of the two functions. For small isospin breaking a common root exists. Starting at around $m_d/m_u \approx 1.5$ in both cases a tension develops. At around $m_d/m_u \approx 1.7$ for the heavier ensemble and at around $m_d/m_u \approx 2$ for the lighter ensemble the non-existence of a common root is more than one-sigma significant.

be therefore already worthwhile to study this UV complete theory and the chiral Lagrangian at leading order in this region of parameter space.

## 7 Conclusions

Summarizing, within this work we have laid the foundation for understanding strongly- interacting dark matter based on QCD-like symplectic gauge theories. To this end, we have

constructed all possible symmetries for the non-degenerate underlying theory, including the case when gauged under a new $U(1)'$ symmetry. This yields symmetry-based constraints on absolutely stable and unstable light states.

We have constructed the chiral low-energy effective theory of the bound states, the dark hadrons, of this theory. In this process we extended previous results, see e. g. [1, 7, 11, 29, 33, 48, 60, 77, 78, 92], to include simultaneously processes mediated by the WZW term, mass-split spectra, and couplings to an Abelian gauge messenger. We fully elaborated the leading-order structure, including the Feynman rules. This comprehensive construction integrates many elements, which have been available separately in the literature, in one common framework. It therefore provides a ready-to-use effective theory to search for this type of DM at colliders and in direct detection experiments, and understand how it can fulfil astrophysical constraints. Especially, this theory can be straightforwardly extended to any QCD-like theory with a pseudo-real gauge group with two fundamental dark quarks. It will therefore serve as a versatile framework for future investigations.

The particular strength of our formulation is that it has been developed in parallel with a lattice implementation of the same theory, ensuring a coherent language and conventions. This is decisive, as effective low-energy theories require limits of validity. In the degenerate case, previous lattice results [6, 7] gave already the condition $m_\rho/m_\pi \gtrsim 1.4$. We extended this to the mass non-degenerate case, yielding the additional requirement that the PCAC dark quark masses, as defined in section 6.2, may not be split by more than a factor of 1.5 at the validity edge. We also showed how the theory at larger mass splittings is crossing over into a heavy-light system. While this requires a new low-energy effective theory, these results open interesting possibilities for strongly-interacting dark matter with split hierarchies and stable dark matter particles at different scales.

Finally, in section 4.4 we used the lattice results to determine the relevant LEC when coupling to the SM through a vector mediator. This shows how non-perturbative results can be used to reduce the number of free parameters when searching for DM by implementing constraints from the ultraviolet completion. This integration between effective theory and lattice simulations reduces substantially the number of free parameters in a pure effective theory approach, which is our ultimate aim. As figure 4 shows, the systematic comparison between different ultraviolet theories in this way yields very general constraints, making our approach predictive.

Summing up, we have created both a more comprehensive blueprint for how to construct viable low-energy theories for strongly-interacting dark matter and we have provided a particular implementation for an ultraviolet completion with a $Sp(4)$ QCD-like theory, in the spirit of e.g. [22, 30, 31, 93]. This can now be exploited to perform phenomenological calculations with a much higher degree of systematic control than previously possible. The latter can furthermore be systematically extended by going to higher orders with simultaneous lattice calculations of further LECs, just like in ordinary QCD.

# Acknowledgements

This work was supported by the FWF Austrian Science Fund research teams grant STRONG-DM (FG1). SK is supported by the FWF Austrian Science Fund Elise-Richter grant project number V592-N27. MN is supported by the Austrian Science Fund FWF under the Doctoral Program W1252-N27 Particles and Interactions. We thank E. Bennett, B. Lucini, and J.-W. Lee for helpful discussions and them and the other authors of [6–8] for access to the $Sp(2N)$ HiRep code prior to publication. We are grateful to B. Lucini and M. Piai for a critical reading of the draft and helpful comments. We thank J. Pomper for his thorough proofreading and

clarification on the draft and H. Kolesova for discussions on the WZW term. The lattice results presented have been obtained using the Vienna Scientific Cluster (VSC).

# A $Sp(2N)$ groups: Defining properties and generators

In this section, we provide a general description of the symplectic group $Sp(2N)$. We suppress the colour and flavour indices, since all properties of the group are equivalent in the colour and flavour space and label the generators as $T^a$. We note, however, that all relations quoted here also apply to the colour group $Sp(4)_c$ and its generators $\tau^a$. The fundamental representation of any $Sp(2N)$ group is pseudo-real. The representation is isomorphic to its complex conjugate representation. This can be seen from the defining property of the $Sp(2N)$ group: it is the subgroup of $SU(2N)$ that leaves

$$E = \begin{pmatrix} 0 & \mathbb{1}_N \\ -\mathbb{1}_N & 0 \end{pmatrix}, \tag{A.1}$$

invariant. It consists of all $SU(2N)$ transformations that fulfil $U^* = EUE^\dagger$. In the context of colour groups we will denote the invariant tensor as $S$ to make the context unambiguous. For the invariant tensor $E$ the following relations hold:

$$E^\dagger = E^{-1} = E^T = -E, \quad E^2 = -\mathbb{1}_{2N}. \tag{A.2}$$

On the level of the generators $T^a$ using $U = \exp(i\alpha^a T^a)$ this is equivalent to

$$T^{a*} = -ET^a E^\dagger. \tag{A.3}$$

For completeness, we point out that not every representation of a $Sp(2N)$ group is pseudo-real but also real representations exist, such as the adjoint and antisymmetric representations, see e.g. [8]. However, complex representations of $Sp(2N)$ do not exist. The group $SU(4)$ has 15 generators and the subgroup $Sp(4)$ has 10 generators. From (A.3) follows the relation

$$ET^a + (T^a)^T E = 0 \quad \text{for } a = 6, \ldots, 15, \tag{A.4}$$

for the generators of the $Sp(4)$ subgroup while the other 5 generators satisfy

$$ET^a - (T^a)^T E = 0 \quad \text{for } a = 1, \ldots, 5. \tag{A.5}$$

The generators with normalization $\text{Tr}\{T^a T^b\} = \frac{1}{2}\delta^{ab}$ are given by [94]

$$
T^1 = \frac{1}{2\sqrt{2}}\begin{pmatrix} 0 & 1 & 0 & 0 \\ 1 & 0 & 0 & 0 \\ 0 & 0 & 0 & 1 \\ 0 & 0 & 1 & 0 \end{pmatrix}, \quad
T^2 = \frac{1}{2\sqrt{2}}\begin{pmatrix} 0 & -i & 0 & 0 \\ i & 0 & 0 & 0 \\ 0 & 0 & 0 & i \\ 0 & 0 & -i & 0 \end{pmatrix}, \quad
T^3 = \frac{1}{2\sqrt{2}}\begin{pmatrix} 1 & 0 & 0 & 0 \\ 0 & -1 & 0 & 0 \\ 0 & 0 & 1 & 0 \\ 0 & 0 & 0 & -1 \end{pmatrix},
$$

$$
T^4 = \frac{1}{2\sqrt{2}}\begin{pmatrix} 0 & 0 & 0 & -i \\ 0 & 0 & i & 0 \\ 0 & -i & 0 & 0 \\ i & 0 & 0 & 0 \end{pmatrix}, \quad
T^5 = \frac{1}{2\sqrt{2}}\begin{pmatrix} 0 & 0 & 0 & 1 \\ 0 & 0 & -1 & 0 \\ 0 & -1 & 0 & 0 \\ 1 & 0 & 0 & 0 \end{pmatrix}, \quad
T^6 = \frac{1}{2\sqrt{2}}\begin{pmatrix} 0 & 0 & -i & 0 \\ 0 & 0 & 0 & -i \\ i & 0 & 0 & 0 \\ 0 & i & 0 & 0 \end{pmatrix},
$$

$$
T^7 = \frac{1}{2\sqrt{2}}\begin{pmatrix} 0 & 0 & 0 & -i \\ 0 & 0 & -i & 0 \\ 0 & i & 0 & 0 \\ i & 0 & 0 & 0 \end{pmatrix}, \quad
T^8 = \frac{1}{2\sqrt{2}}\begin{pmatrix} 0 & -i & 0 & 0 \\ i & 0 & 0 & 0 \\ 0 & 0 & 0 & -i \\ 0 & 0 & i & 0 \end{pmatrix}, \quad
T^9 = \frac{1}{2\sqrt{2}}\begin{pmatrix} 0 & 0 & -i & 0 \\ 0 & 0 & 0 & i \\ i & 0 & 0 & 0 \\ 0 & -i & 0 & 0 \end{pmatrix}, \tag{A.6}
$$

$$
T^{10} = \frac{1}{2}\begin{pmatrix} 0 & 0 & 1 & 0 \\ 0 & 0 & 0 & 0 \\ 1 & 0 & 0 & 0 \\ 0 & 0 & 0 & 0 \end{pmatrix}, \quad
T^{11} = \frac{1}{2\sqrt{2}}\begin{pmatrix} 0 & 0 & 0 & 1 \\ 0 & 0 & 1 & 0 \\ 0 & 1 & 0 & 0 \\ 1 & 0 & 0 & 0 \end{pmatrix}, \quad
T^{12} = \frac{1}{2}\begin{pmatrix} 0 & 0 & 0 & 0 \\ 0 & 0 & 0 & 1 \\ 0 & 0 & 0 & 0 \\ 0 & 1 & 0 & 0 \end{pmatrix},
$$

$$
T^{13} = \frac{1}{2\sqrt{2}}\begin{pmatrix} 0 & 1 & 0 & 0 \\ 1 & 0 & 0 & 0 \\ 0 & 0 & 0 & -1 \\ 0 & 0 & -1 & 0 \end{pmatrix}, \quad
T^{14} = \frac{1}{2\sqrt{2}}\begin{pmatrix} 1 & 0 & 0 & 0 \\ 0 & -1 & 0 & 0 \\ 0 & 0 & -1 & 0 \\ 0 & 0 & 0 & 1 \end{pmatrix}, \quad
T^{15} = \frac{1}{2\sqrt{2}}\begin{pmatrix} 1 & 0 & 0 & 0 \\ 0 & 1 & 0 & 0 \\ 0 & 0 & -1 & 0 \\ 0 & 0 & 0 & -1 \end{pmatrix}.
$$

The totally antisymmetric structure constants $f^{a,b,c}$ of the $SU(4)$ group are defined as

$$[T^a, T^b] = i f^{abc} T^c, \tag{A.7}$$

and all non-zero structure constants are given by

$$f^{1,2,14} = f^{2,3,13} = f^{2,4,6} = f^{3,5,7} = -f^{1,3,8} = -f^{1,5,9} = -f^{3,4,11} = -f^{4,5,15} = \frac{1}{\sqrt{2}}, \tag{A.8}$$

$$f^{1,4,10} = f^{2,5,10} = f^{2,5,12} = -f^{1,4,12} = \frac{1}{2}, \tag{A.9}$$

In section 2 we found that two non-degenerate fermion masses break the global symmetry down to $SU(2)_u \times SU(2)_d$. This group has 6 generators, that are combinations of the generators of $Sp(4)$

$$T_u^1 = T^{10}, \quad T_u^2 = \frac{1}{\sqrt{2}}(T^6 + T^9), \quad T_u^3 = \frac{1}{\sqrt{2}}(T^{14} + T^{15}), \tag{A.10}$$

$$T_d^1 = T^{12}, \quad T_d^2 = \frac{1}{\sqrt{2}}(T^6 - T^9), \quad T_d^3 = \frac{1}{\sqrt{2}}(T^{15} - T^{14}). \tag{A.11}$$

Occasionally it is convenient to use another basis of the generators than the one in (A.6). A different basis of $SU(4)$ and $Sp(4)$ algebras is obtained by exchanging the second row with the third row and the second column with the third column. In this form, all generators are either block-diagonal or anti-block-diagonal where the different blocks are either the identity matrix $\mathbb{1}$ or one of the Pauli matrices $\sigma_i$,

$$
\begin{aligned}
\tilde{T}^1 &= \frac{1}{2\sqrt{2}}\begin{pmatrix} 0 & \mathbb{1}_2 \\ \mathbb{1}_2 & 0 \end{pmatrix}, &
\tilde{T}^2 &= \frac{1}{2\sqrt{2}}\begin{pmatrix} 0 & -i\sigma_3 \\ i\sigma_3 & 0 \end{pmatrix}, &
\tilde{T}^3 &= \frac{1}{2\sqrt{2}}\begin{pmatrix} \mathbb{1}_2 & 0 \\ 0 & -\mathbb{1}_2 \end{pmatrix}, \\
\tilde{T}^4 &= \frac{1}{2\sqrt{2}}\begin{pmatrix} 0 & -i\sigma_1 \\ i\sigma_1 & 0 \end{pmatrix}, &
\tilde{T}^5 &= \frac{1}{2\sqrt{2}}\begin{pmatrix} 0 & i\sigma_2 \\ -i\sigma_2 & 0 \end{pmatrix}, &
\tilde{T}^6 &= \frac{1}{2\sqrt{2}}\begin{pmatrix} \sigma_2 & 0 \\ 0 & \sigma_2 \end{pmatrix}, \\
\tilde{T}^7 &= \frac{1}{2\sqrt{2}}\begin{pmatrix} 0 & \sigma_2 \\ \sigma_2 & 0 \end{pmatrix}, &
\tilde{T}^8 &= \frac{1}{2\sqrt{2}}\begin{pmatrix} 0 & -i\mathbb{1}_2 \\ i\mathbb{1}_2 & 0 \end{pmatrix}, &
\tilde{T}^9 &= \frac{1}{2\sqrt{2}}\begin{pmatrix} \sigma_2 & 0 \\ 0 & -\sigma_2 \end{pmatrix}, \\
\tilde{T}^{10} &= \frac{1}{2}\begin{pmatrix} \sigma_1 & 0 \\ 0 & 0 \end{pmatrix}, &
\tilde{T}^{11} &= \frac{1}{2\sqrt{2}}\begin{pmatrix} 0 & \sigma_1 \\ \sigma_1 & 0 \end{pmatrix}, &
\tilde{T}^{12} &= \frac{1}{2}\begin{pmatrix} 0 & 0 \\ 0 & \sigma_1 \end{pmatrix}, \\
\tilde{T}^{13} &= \frac{1}{2\sqrt{2}}\begin{pmatrix} 0 & \sigma_3 \\ \sigma_3 & 0 \end{pmatrix}, &
\tilde{T}^{14} &= \frac{1}{2\sqrt{2}}\begin{pmatrix} \sigma_3 & 0 \\ 0 & -\sigma_3 \end{pmatrix}, &
\tilde{T}^{15} &= \frac{1}{2\sqrt{2}}\begin{pmatrix} \sigma_3 & 0 \\ 0 & \sigma_3 \end{pmatrix}.
\end{aligned} \tag{A.12}
$$

Then, the symplectic matrix $E$ is written as

$$\tilde{E} = \begin{pmatrix} i\sigma_2 & 0 \\ 0 & i\sigma_2 \end{pmatrix}. \tag{A.13}$$

In this basis, all $SU(2)_u \times SU(2)_d$ generators and thus transformations are fully block-diagonal:

$$
\begin{aligned}
\tilde{T}_u^1 &= \tilde{T}^{10} = \frac{1}{2}\begin{pmatrix} \sigma_1 & 0 \\ 0 & 0 \end{pmatrix}, &
\tilde{T}_d^1 &= \tilde{T}^{12} = \frac{1}{2}\begin{pmatrix} 0 & 0 \\ 0 & \sigma_1 \end{pmatrix}, \\
\tilde{T}_u^2 &= \frac{(\tilde{T}^6 + \tilde{T}^9)}{\sqrt{2}} = \frac{1}{2}\begin{pmatrix} \sigma_2 & 0 \\ 0 & 0 \end{pmatrix}, &
\tilde{T}_d^2 &= \frac{(\tilde{T}^6 - \tilde{T}^9)}{\sqrt{2}} = \frac{1}{2}\begin{pmatrix} 0 & 0 \\ 0 & \sigma_2 \end{pmatrix}, \\
\tilde{T}_u^3 &= \frac{(\tilde{T}^{14} + \tilde{T}^{15})}{\sqrt{2}} = \frac{1}{2}\begin{pmatrix} \sigma_3 & 0 \\ 0 & 0 \end{pmatrix}, &
\tilde{T}_d^3 &= \frac{(\tilde{T}^{15} - \tilde{T}^{14})}{\sqrt{2}} = \frac{1}{2}\begin{pmatrix} 0 & 0 \\ 0 & \sigma_3 \end{pmatrix}.
\end{aligned} \tag{A.14}
$$

Another useful basis can be obtained by considering the matrix of the Goldstone bosons in (21) as well as the matrices of the spin-1 states in (36). In (29) we have rewritten the Goldstone bosons such that in every matrix element only one Goldstone field appears. Then, the Goldstone matrix takes the form

$$\pi = \sum_{i=1,\dots,5} \pi_a T^a = \sum_{N=A,\dots,E} \pi_N T^N = \frac{1}{2} \begin{pmatrix} \pi^C & \pi^B & 0 & \pi^E \\ \pi^A & -\pi^C & -\pi^E & 0 \\ 0 & -\pi^D & \pi^C & \pi^A \\ \pi^D & 0 & \pi^B & -\pi^C \end{pmatrix}, \qquad (A.15)$$

which implicitly defines the generators $T^N$ with $N = A, B, C, D, E$. In QCD this leads to the usual pion charge eigenstates $\pi^\pm$ and $\pi^0$ as shown in (30). We can extend that to the spin-1 states and by that define a new basis for the generators. It is was shown that the spin-1 states $\rho_{14}$ and $\rho_{15}$ correspond to the $\rho^0$ and $\omega$ meson of QCD [8]. Therefore, we define the other $T^N$ by specifying the $J^D = 1^-$ matrix

$$\rho^{(J^D=1^-)} = \sum_{a=6,\dots,15} \rho_a T^a = \sum_{N=F,\dots,O} \rho_N T^N \qquad (A.16)$$

$$= \frac{1}{2} \begin{pmatrix} \frac{1}{\sqrt{2}}\left(\rho^O + \rho^N\right) & \rho^M & -\rho^J & -\rho^K \\ \rho^H & \frac{1}{\sqrt{2}}\left(\rho^O - \rho^N\right) & -\rho^K & -\rho^L \\ \rho^F & \rho^G & -\frac{1}{\sqrt{2}}\left(\rho^O + \rho^N\right) & -\rho^H \\ \rho^G & \rho^I & -\rho^M & -\frac{1}{\sqrt{2}}\left(\rho^O - \rho^N\right) \end{pmatrix}. \qquad (A.17)$$

All off-diagonal elements contain only one vector field and the states corresponding to the $\rho^0$ and $\omega$ of QCD appear on the diagonal. The generators in this basis are given by

$$T^A = \frac{1}{\sqrt{2}}\left(T^1 - iT^2\right), \qquad\qquad T^B = \frac{1}{\sqrt{2}}\left(T^1 + iT^2\right), \quad T^C = T^3 \qquad (A.18)$$

$$T^D = \frac{1}{\sqrt{2}}\left(T^5 - iT^4\right), \qquad\qquad T^E = \frac{1}{\sqrt{2}}\left(T^5 + iT^4\right), \quad T^F = \frac{1}{\sqrt{2}}\left(\sqrt{2}T^{10} - i\left(T^6 + T^9\right)\right),$$

$$T^G = \frac{1}{\sqrt{2}}\left(T^{11} - iT^7\right), \qquad\qquad T^H = \frac{1}{\sqrt{2}}\left(T^{13} - iT^8\right), \quad T^I = \frac{1}{\sqrt{2}}\left(\sqrt{2}T^{12} - i\left(T^6 - T^9\right)\right),$$

$$T^J = \frac{1}{\sqrt{2}}\left(-\sqrt{2}T^{10} - i\left(T^6 + T^9\right)\right), \quad T^K = \frac{-1}{\sqrt{2}}\left(T^{11} + iT^7\right), \quad T^L = \frac{1}{\sqrt{2}}\left(-\sqrt{2}T^{12} - i\left(T^6 - T^9\right)\right),$$

$$T^M = \frac{1}{\sqrt{2}}\left(T^{13} + iT^8\right), \qquad\qquad T^N = T^{14}, \qquad\qquad T^O = T^{15}.$$

Analogously, we can define the axial-vector matrix, i.e., the $J^D = 1^+$ mesons, as

$$\rho^{(J^D=1^+)} = \sum_{a=1,\dots,5} \rho_a T^a = \sum_{N=A,\dots,E} a_N T^N = \frac{1}{2} \begin{pmatrix} a^C & a^B & 0 & a^E \\ a^A & -a^C & -a^E & 0 \\ 0 & -a^D & a^C & a^A \\ a^D & 0 & a^B & -a^C \end{pmatrix}, \qquad (A.19)$$

where we have given them a different name $a_N$ in the $T^N$ basis in order to emphasize the different parity compared to the $\rho_N$ states. Similar relations like (29) can then be obtained from (A.18) for the vector and axial-vector matrix.

# B Mesonic states and multiplets

## B.1 States and $Sp(4)$ multiplets

The meson spin-0 and spin-1 fermion bilinears have already been constructed in [8]. They have the same structure as those appearing in a $N_f = 1$ theory [34]. For completeness we give

in table 4 here the operators that source the $J^D = 0^-, 1^+$ and $1^-$ multiplets of $Sp(4)$, i.e. the pseudoscalars, axial-vectors and vectors constructed from the generators in the basis of (A.6) and (A.18), respectively. The other spin-0 and spin-1 states are the scalar 5-plet as well as the scalar flavour singlet.

## B.2 Multiplet structure under $SU(2)_u \times SU(2)_d$

Because $SU(2)$ exponentials are easily calculated analytically, we may provide a general expression for $SU(2)_d \times SU(2)_u$ transformations. In the basis given by (A.12), the transformation is in block diagonal form

$$V = \begin{pmatrix} a & -b^* & 0 & 0 \\ b & a^* & 0 & 0 \\ 0 & 0 & e & -c^* \\ 0 & 0 & c & e^* \end{pmatrix}, \tag{B.1}$$

where the diagonal matrix blocks are elements of SU(2). We denote the complex coefficients of the individual $SU(2)$ by $(a, b)$ and $(e, c)$; they fulfil $|a|^2 + |b|^2 = 1$ and $|c|^2 + |e|^2 = 1$, respectively. It is convenient to rewrite the Dirac spinors in terms of their left- and right-handed projections

$$P_{R/L} u = u_{R/L}, \tag{B.2}$$
$$P_{R/L} d = d_{R/L}, \tag{B.3}$$
$$P_{R/L} = (1 \pm \gamma_5)/2. \tag{B.4}$$

Under a $SU(2)_d \times SU(2)_u$ transformation the components of $\Psi$ transform as $\Psi \to V\Psi$, or, explicitly,

$$\begin{pmatrix} u_L \\ \tilde{u}_R \\ d_L \\ \tilde{d}_R \end{pmatrix} = \begin{pmatrix} u_L \\ -SC\bar{u}_R^T \\ d_L \\ -SC\bar{d}_R^T \end{pmatrix} \to \begin{pmatrix} au_L - b^*\tilde{u}_R \\ bu_L + a^*\tilde{u}_R \\ ed_L - c^*\tilde{d}_R \\ cd_L + e^*\tilde{d}_R \end{pmatrix}. \tag{B.5}$$

For completeness we give the following useful relations for the colour matrix $S$ as well as the charge conjugation operator in Minkowski space and the spinors $\tilde{u}_R$ and $\tilde{d}_R$:

$$C^\dagger = C^{-1} = C^T = -C, \quad C^2 = -\mathbb{1}, \quad C\gamma_\mu C^{-1} = -\gamma_\mu^T, \tag{B.6}$$
$$(SC)^\dagger = (SC)^{-1} = (SC)^T = SC, \quad (SC)^2 = \mathbb{1}, \quad SC\gamma_\mu SC = -\gamma_\mu^T, \tag{B.7}$$

$$q_R = SC\bar{\tilde{q}}_R^T, \qquad\qquad q_R^T = \bar{\tilde{q}}_R SC, \tag{B.8}$$
$$\bar{q}_R = -\tilde{q}_R^T SC, \qquad\qquad \bar{q}_R^T = -SC\tilde{q}_R. \tag{B.9}$$

The scalar $\bar{u}d$ and vectors $\bar{u}\gamma_\mu d$ transform under $SU(2)_d \times SU(2)_u$ as

$$\begin{aligned}
\bar{u}d = \bar{u}_L d_R + \bar{u}_R d_L \to \quad & a^*c^* \left( \bar{u}_L SC\bar{d}_L^T + \bar{u}_R SC\bar{d}_R^T \right) - be \left( u_L^T SC d_L + u_R^T SC d_R \right) \\
& + a^*e \left( \bar{u}_L d_R + \bar{u}_R d_L \right) - bc^* \left( u_L^T \bar{d}_R^T + u_R^T \bar{d}_L^T \right) \\
& = a^*c^* \left( \bar{u} SC\bar{d}^T \right) - be \left( u^T SC d \right) + a^*e \left( \bar{u}d \right) + bc^* \left( \bar{d}u \right), \tag{B.10}
\end{aligned}$$

$$\begin{aligned}
\bar{u}\gamma_\mu d = \bar{u}_L \gamma_\mu d_L + \bar{u}_R \gamma_\mu d_R \to \quad & a^*e \left( \bar{u}_L \gamma_\mu d_L + \bar{u}_R \gamma_\mu d_R \right) + bc^* \left( u_L^T \gamma_\mu^T \bar{d}_L^T + u_R^T \gamma_\mu^T \bar{d}_R^T \right) \\
& + a^*c^* \left( \bar{u}_L \gamma_\mu SC\bar{d}_R^T + \bar{u}_R \gamma_\mu SC\bar{d}_L^T \right) + be \left( u_L^T \gamma_\mu^T SC d_R + u_R^T \gamma_\mu^T SC d_L \right) \\
& = a^*e \left( \bar{u}\gamma_\mu d \right) - bc^* \left( \bar{d}\gamma_\mu u \right) + a^*c^* \left( \bar{u}\gamma_\mu SC\bar{d}^T \right) - be \left( u^T SC\gamma_\mu d \right).
\end{aligned}$$

Table 4: Fermion bilinears of the $J^D = 0^-, 1^\pm$ meson multiplets constructed from the generators in the basis of (A.6) (left) and (A.18) (right) respectively. In addition we give the $J^P$ quantum numbers.

| | $\Psi^T SC\gamma_5 T^n E\Psi + \bar{\Psi}ET^n SC\gamma_5\bar{\Psi}^T$ | | $\Psi^T SCT^N\gamma_5 E\Psi + \bar{\Psi}ET^N SC\gamma_5\bar{\Psi}^T$ | $J^P$ | $J^D$ |
|---|---|---|---|---|---|
| $\pi_1$ | $\frac{1}{\sqrt{2}}\left(\bar{u}\gamma_5 d + \bar{d}\gamma_5 u\right)$ | $\pi^A$ | $\bar{u}\gamma_5 d$ | $0^-$ | $0^-$ |
| $\pi_2$ | $\frac{i}{\sqrt{2}}\left(\bar{d}\gamma_5 u - \bar{u}\gamma_5 d\right)$ | $\pi^B$ | $\bar{d}\gamma_5 u$ | $0^-$ | $0^-$ |
| $\pi_3$ | $\frac{1}{\sqrt{2}}\left(\bar{u}\gamma_5 u - \bar{d}\gamma_5 d\right)$ | $\pi^C$ | $\frac{1}{\sqrt{2}}\left(\bar{u}\gamma_5 u - \bar{d}\gamma_5 d\right)$ | $0^-$ | $0^-$ |
| $\pi_4$ | $\frac{i}{\sqrt{2}}\left(\bar{d}\gamma_5 SC\bar{u}^T - d^T SC\gamma_5 u\right)$ | $\pi^D$ | $\bar{d}\gamma_5 SC\bar{u}^T$ | $0^+$ | $0^-$ |
| $\pi_5$ | $\frac{1}{\sqrt{2}}\left(\bar{d}\gamma_5 SC\bar{u}^T + d^T SC\gamma_5 u\right)$ | $\pi^E$ | $d^T SC\gamma_5 u$ | $0^+$ | $0^-$ |

| | $\Psi^T SC\gamma_5 T^0 E\Psi + \bar{\Psi}ET^0 SC\gamma_5\bar{\Psi}^T$ | | $\Psi^T SC\gamma_5 T^0 E\Psi + \bar{\Psi}ET^0 SC\gamma_5\bar{\Psi}^T$ | $J^P$ | $J^D$ |
|---|---|---|---|---|---|
| $\eta'$ | $\frac{1}{\sqrt{2}}\left(\bar{u}\gamma_5 u + \bar{d}\gamma_5 d\right)$ | $\eta'$ | $\frac{1}{\sqrt{2}}\left(\bar{u}\gamma_5 u + \bar{d}\gamma_5 d\right)$ | $0^-$ | $0^-$ |

| | $2\,\bar{\Psi}T^n\gamma_\mu\gamma_5\Psi$ | | $2\bar{\Psi}T^N\gamma_\mu\gamma_5\Psi$ | $J^P$ | $J^D$ |
|---|---|---|---|---|---|
| $\rho_1$ | $\frac{1}{\sqrt{2}}\left(\bar{u}\gamma_\mu\gamma_5 d + \bar{d}\gamma_\mu\gamma_5 u\right)$ | $a^A$ | $\bar{d}\gamma_\mu\gamma_5 u$ | $1^+$ | $1^+$ |
| $\rho_2$ | $\frac{i}{\sqrt{2}}\left(\bar{d}\gamma_\mu\gamma_5 u - \bar{u}\gamma_\mu\gamma_5 d\right)$ | $a^B$ | $\bar{u}\gamma_\mu\gamma_5 d$ | $1^+$ | $1^+$ |
| $\rho_3$ | $\frac{1}{\sqrt{2}}\left(\bar{u}\gamma_\mu\gamma_5 u - \bar{d}\gamma_\mu\gamma_5 d\right)$ | $a^C$ | $\frac{1}{\sqrt{2}}\left(\bar{u}\gamma_\mu\gamma_5 u - \bar{d}\gamma_\mu\gamma_5 d\right)$ | $1^+$ | $1^+$ |
| $\rho_4$ | $\frac{i}{\sqrt{2}}\left(\bar{d}SC\gamma_\mu\gamma_5\bar{u}^T - d^T SC\gamma_\mu\gamma_5 u\right)$ | $a^D$ | $d^T SC\gamma_\mu\gamma_5 u$ | $1^-$ | $1^+$ |
| $\rho_5$ | $\frac{1}{\sqrt{2}}\left(\bar{d}SC\gamma_\mu\gamma_5\bar{u}^T + d^T SC\gamma_\mu\gamma_5 u\right)$ | $a^E$ | $\bar{d}\gamma_\mu\gamma_5 SC\bar{u}^T$ | $1^-$ | $1^+$ |

| | $2\bar{\Psi}T^n\gamma_\mu\Psi$ | | $2\bar{\Psi}T^N\gamma_\mu\Psi$ | $J^P$ | $J^D$ |
|---|---|---|---|---|---|
| $\rho_6$ | $\frac{i}{\sqrt{2}}\left(\bar{u}\gamma_\mu SCP_L\bar{u}^T + u^T SC\gamma_\mu P_L u + \bar{d}\gamma_\mu SCP_L\bar{d}^T + d^T SC\gamma_\mu P_L d\right)$ | $\rho^F$ | $u^T SC\gamma_\mu P_L u$ | $1^+$ | $1^-$ |
| $\rho_7$ | $\frac{i}{\sqrt{2}}\left(\bar{u}\gamma_\mu SC\bar{d}^T + u^T SC\gamma_\mu d\right)$ | $\rho^G$ | $u^T SC\gamma_\mu d$ | $1^+$ | $1^-$ |
| $\rho_8$ | $\frac{i}{\sqrt{2}}\left(-\bar{u}\gamma_\mu d + \bar{d}\gamma_\mu u\right)$ | $\rho^H$ | $\bar{d}\gamma_\mu u$ | $1^-$ | $1^-$ |
| $\rho_9$ | $\frac{i}{\sqrt{2}}\left(\bar{u}\gamma_\mu SCP_L\bar{u}^T + u^T SC\gamma_\mu P_L u - \bar{d}\gamma_\mu SCP_L\bar{d}^T - d^T SC\gamma_\mu P_L d\right)$ | $\rho^I$ | $d^T SC\gamma_\mu P_L d$ | $1^+$ | $1^-$ |
| $\rho_{10}$ | $u^T SC\gamma_\mu P_L u - \bar{u}\gamma_\mu SCP_L\bar{u}^T$ | $\rho^J$ | $\bar{u}\gamma_\mu SCP_L\bar{u}^T$ | $1^+$ | $1^-$ |
| $\rho_{11}$ | $\frac{1}{\sqrt{2}}\left(-\bar{u}\gamma_\mu SC\bar{d}^T + u^T SC\gamma_\mu d\right)$ | $\rho^K$ | $\bar{u}\gamma_\mu SC\bar{d}^T$ | $1^+$ | $1^-$ |
| $\rho_{12}$ | $d^T SC\gamma_\mu P_L d - \bar{d}\gamma_\mu SCP_L\bar{d}^T$ | $\rho^L$ | $\bar{d}\gamma_\mu SCP_L\bar{d}^T$ | $1^+$ | $1^-$ |
| $\rho_{13}$ | $\frac{1}{\sqrt{2}}\left(\bar{u}\gamma_\mu d + \bar{d}\gamma_\mu u\right)$ | $\rho^M$ | $\bar{u}\gamma_\mu d$ | $1^-$ | $1^-$ |
| $\rho_{14}$ | $\frac{1}{\sqrt{2}}\left(\bar{u}\gamma_\mu u - \bar{d}\gamma_\mu d\right)$ | $\rho^N$ | $\frac{1}{\sqrt{2}}\left(\bar{u}\gamma_\mu u - \bar{d}\gamma_\mu d\right)$ | $1^-$ | $1^-$ |
| $\rho_{15}$ | $\frac{1}{\sqrt{2}}\left(\bar{u}\gamma_\mu u + \bar{d}\gamma_\mu u\right)$ | $\rho^O$ | $\frac{1}{\sqrt{2}}\left(\bar{u}\gamma_\mu u + \bar{d}\gamma_\mu d\right)$ | $1^-$ | $1^-$ |

Here, we use the property $\left(\phi^T\Gamma\chi\right)^T = -\chi^T\Gamma^T\phi$ for Grassmann variables $\phi,\chi$. The pseudo-scalars and axial-vectors transform similarly since the only difference is the gamma matrix $\gamma_5$.

$$\bar{u}\gamma_5 d = \bar{u}_L d_R - \bar{u}_R d_L$$
$$\rightarrow a^* c^*\left(\bar{u}SC\gamma_5\bar{d}^T\right) - be\left(u^T SC\gamma_5 d\right) + a^* e\left(\bar{u}\gamma_5 d\right) + bc^*\left(\bar{d}\gamma_5 u\right), \tag{B.11}$$
$$\bar{u}\gamma_\mu\gamma_5 d = \bar{u}_R\gamma_\mu d_R - \bar{u}_L\gamma_\mu d_L$$
$$\rightarrow a^* e\left(\bar{u}\gamma_\mu\gamma_5 d\right) - bc^*\left(\bar{d}\gamma_\mu\gamma_5 u\right) + a^* c^*\left(\bar{u}\gamma_\mu\gamma_5 SC\bar{d}^T\right) - be\left(u^T SC\gamma_\mu\gamma_5 d\right), \tag{B.12}$$

From this we conclude that these states form a quadruplet under $SU(2)_d \times SU(2)_u$. The remaining pseudo-scalars and scalars transform as singlets. The associated operators have the

form $\bar{u}\Gamma u \pm \bar{d}\Gamma d$. The individual $SU(2)_{d,u}$ only change one of the terms, so it is sufficient to look at the transformation property of, say, $\bar{u}\Gamma u$,

$$
\begin{aligned}
\bar{u}u &= \bar{u}_L u_R + \bar{u}_R u_L \to \bar{u}_L u_R + \bar{u}_R u_L = \bar{u}u\,, \\
\bar{u}\gamma_5 u &= \bar{u}_L u_R - \bar{u}_R u_L \to \bar{u}_L u_R + \bar{u}_R u_L = \bar{u}\gamma_5 u\,, \\
\bar{u}\gamma_\mu u &= \bar{u}_L \gamma_\mu u_L + \bar{u}_R \gamma_\mu u_R \to \left(|a|^2 - |b|^2\right)\left(\bar{u}\gamma_\mu u\right) + 2a^* b^* \left(\bar{u}\gamma_\mu S C P_L \bar{u}^T\right) - 2ab\left(u^T S C \gamma_\mu P_L u\right)\,, \\
\bar{u}\gamma_\mu \gamma_5 u &= \bar{u}_R \gamma_\mu u_R - \bar{u}_L \gamma_\mu u_L \to \bar{u}\gamma_\mu \gamma_5 u\,.
\end{aligned}
\tag{B.13}
$$

Similar expressions for $\bar{d}\Gamma d$ are obtained by replacing $u \to d, a \to e$ and $b \to c$.

## C  Feynman rules

In section C.1 we provide a subset of Feynman rules which are used in this work for Goldstone bosons and the iso-singlet state $\eta'$ in presence and isolation from additional, external interactions for degenerate fermions $m_u = m_d$. Additionally, we show some Feynman rules for axial- and vector states $\rho_\mu$ coupled to Goldstone bosons. In section C.2 the Feynman rules are given for non-degenerate fermions $m_u \neq m_d$. All momenta are taken as *ingoing*, with the Mandelstam variables being defined as

$$
s = (p_1 + p_2)^2 = (k_1 + k_2)^2\,, \tag{C.1}
$$

$$
t = (p_1 + k_1)^2 = (p_2 + k_2)^2\,, \tag{C.2}
$$

$$
u = (p_1 + k_2)^2 = (p_2 + k_1)^2\,, \tag{C.3}
$$

where $p_1 + p_2 + k_1 + k_2 = 0$.

### C.1  Feynman rules for degenerate fermions $m_u = m_d$

$$
\underset{p_{\pi_a}}{\bullet\text{-----}\bullet} = \frac{1}{p_{\pi_a}^2 - m_\pi^2}\,, \tag{C.4}
$$

$$
\underset{p_{\eta'}}{\bullet\text{-----}\bullet} = \frac{1}{p_{\eta'}^2 - m_\pi^2 - \Delta m_{\eta'}^2}\,, \tag{C.5}
$$

$$
= \frac{1}{2f_\pi^2}\left[\delta^{ab}\delta^{cd}(s - m_\pi^2) + \delta^{ac}\delta^{bd}(t - m_\pi^2) + \delta^{ad}\delta^{cb}(u - m_\pi^2)\right], \tag{C.6}
$$

$$
= \frac{N_c}{10\sqrt{2}\pi^2 f_\pi^5}\varepsilon^{\mu\nu\rho\sigma}\left[(p_2 - p_1)_\mu p_{3\nu} k_{1\rho} k_{2\sigma} + \right.
$$
$$
\left. + p_{1\mu} p_{2\nu}\left(k_{1\rho} k_{2\sigma} + k_{2\rho} p_{3\sigma} + p_{3\rho} k_{1\sigma}\right)\right], \tag{C.7}
$$
$$
(a \neq b \neq c \neq d \neq e)
$$

$$\text{(diagram)} = \frac{m_\pi^2}{2f_\pi^2}, \tag{C.8}$$

$$\text{(diagram)} = \frac{m_\pi^2}{2f_\pi^2}, \tag{C.9}$$

$$\text{(diagram)} \quad V^\mu = 2ie_D \operatorname{Tr}\left(\mathcal{Q}\left[T^a, T^b\right]\right)(p_1 - p_2)^\mu, \tag{C.10}$$

$$\text{(diagram)} = 4e_D^2 \operatorname{Tr}\left(\mathcal{Q}^2\left(T^a T^b + T^b T^a\right) - 2T^a \mathcal{Q} T^b \mathcal{Q}\right)\eta^{\mu\nu}, \tag{C.11}$$

$$\text{(diagram)} \quad \rho_\mu^c = ig_\rho f^{abc}(p_1 - p_2)^\mu, \tag{C.12}$$

$$\text{(diagram)} = \begin{aligned}&2g_{\rho\pi\pi}^2 \operatorname{Tr}(T^a T^b T^c T^d + T^b T^a T^c T^d + T^a T^b T^d T^c \\ &\quad + T^b T^a T^d T^c - 2T^a T^c T^b T^d \\ &\quad - 2T^a T^d T^b T^c)\eta^{\mu\nu}.\end{aligned} \tag{C.13}$$

## C.2 Feynman rules for non-degenerate fermions $m_u \neq m_d$

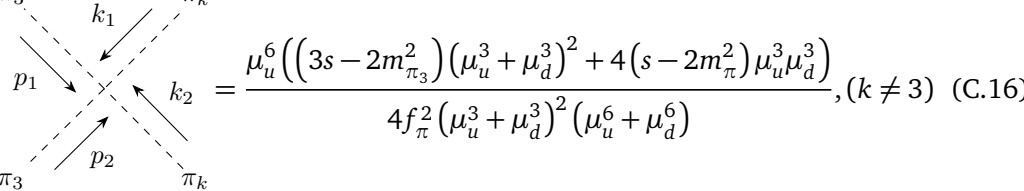

$$\text{(diagram)} = \frac{1}{p_{\pi_a}^2 - m_\pi^2}, \tag{C.14}$$

$$\text{(diagram)} = \frac{2\mu_u^6}{f_\pi^2\left(\mu_u^3 + \mu_d^3\right)^2}\Big[\delta^{ab}\delta^{cd}(s - m_\pi^2) + \delta^{ac}\delta^{bd}(t - m_\pi^2) \\ + \delta^{ad}\delta^{cb}(u - m_\pi^2)\Big], \qquad a,b,c,d \neq 3 \tag{C.15}$$

$$\text{(diagram)} = \frac{\mu_u^6\left(\left(3s - 2m_{\pi_3}^2\right)\left(\mu_u^3 + \mu_d^3\right)^2 + 4\left(s - 2m_\pi^2\right)\mu_u^3\mu_d^3\right)}{4f_\pi^2\left(\mu_u^3 + \mu_d^3\right)^2\left(\mu_u^6 + \mu_d^6\right)}, (k \neq 3) \tag{C.16}$$

$$\frac{\mu_u^6\left(\left(3t-2m_{\pi_3}^2\right)\left(\mu_u^3+\mu_d^3\right)^2+4\left(t-2m_\pi^2\right)\mu_u^3\mu_d^3\right)}{4f_\pi^2\left(\mu_u^3+\mu_d^3\right)^2\left(\mu_u^6+\mu_d^6\right)},\,(k\neq3) \quad\text{(C.17)}$$

$$\frac{\mu_u^6\left(\left(3u-2m_{\pi_3}^2\right)\left(\mu_u^3+\mu_d^3\right)^2+4\left(u-2m_\pi^2\right)\mu_u^3\mu_d^3\right)}{4f_\pi^2\left(\mu_u^3+\mu_d^3\right)^2\left(\mu_u^6+\mu_d^6\right)},\,(k\neq3)$$

(C.18)

$$\frac{\mu_u^6 m_{\pi_3}^2}{f_\pi^2\left(\mu_u^6+\mu_d^6\right)}, \quad\text{(C.19)}$$

$$\begin{aligned}&\frac{N_c\mu_d^6\sqrt{\mu_u^6+\mu_d^6}}{20\pi^2 f_\pi^5\mu_u^9}\varepsilon^{\mu\nu\rho\sigma}\times\\&\times\Big[\,(p_2-p_1)_\mu p_{3\nu}k_{1\rho}k_{2\sigma}+p_{1\mu}p_{2\nu}\big(k_{1\rho}k_{2\sigma}\\&\quad+k_{2\rho}p_{3\sigma}+p_{3\rho}k_{1\sigma}\big)\Big],\qquad(a\neq b\neq c\neq d\neq e)\,.\end{aligned}$$

(C.20)

## C.3 Feynman rules for non-degenerate fermions $m_u \neq m_d$ with $\eta'$

$$\xrightarrow{\quad p_{\pi_a}\quad} = \frac{1}{p_{\pi_3}^2-m_{\pi_3}^2}\,, \quad\text{(C.21)}$$

$$\xrightarrow{\quad p_{\eta'}\quad} = \frac{1}{p_{\eta'}^2-m_{\eta'}^2}\,, \quad\text{(C.22)}$$

$$\begin{aligned}&\frac{N_c\mu_d^6\sqrt{\mu_u^6+\mu_d^6}}{20\pi^2 f_\pi^5\mu_u^9}\left(\frac{\cos\theta_{\text{mass}}}{\sqrt{3\mu_d^6+\mu_u^6}}-\frac{\sin\theta_{\text{mass}}}{\sqrt{\mu_d^6+3\mu_u^6}}\right)\varepsilon^{\mu\nu\rho\sigma}\times\\&\times\Big[\,(p_2-p_1)_\mu p_{3\nu}k_{1\rho}k_{2\sigma}+p_{1\mu}p_{2\nu}\big(k_{1\rho}k_{2\sigma}+k_{2\rho}p_{3\sigma}+p_{3\rho}k_{1\sigma}\big)\Big]\\&(a\neq b\neq c\neq d\neq e)\,,\end{aligned}$$

(C.23)

$$\begin{aligned}
&= \frac{N_c \mu_d^6 \sqrt{\mu_u^6 + \mu_d^6}}{20\pi^2 f_\pi^5 \mu_u^9} \left( \frac{\cos\theta_{\text{mass}}}{\sqrt{\mu_d^6 + 3\mu_u^6}} - \frac{\sin\theta_{\text{mass}}}{\sqrt{3\mu_d^6 + \mu_u^6}} \right) \varepsilon^{\mu\nu\rho\sigma} \times \\
&\quad \times \Big[ (p_2 - p_1)_\mu\, p_{3\nu} k_{1\rho} k_{2\sigma} + p_{1\mu} p_{2\nu} \big( k_{1\rho} k_{2\sigma} + k_{2\rho} p_{3\sigma} + p_{3\rho} k_{1\sigma} \big) \Big] \\
&\qquad (a \neq b \neq c \neq d \neq 3)\,.
\end{aligned}$$

(C.24)

## D  Lattice setup

### D.1  Action

The HiRep code [88] is based on the Wilson gauge action

$$S_g[U] = \beta \sum_x \sum_{\mu<\nu} \left[ \mathbb{1} - \frac{1}{4} \text{Re tr } U_{\mu\nu}(x) \right], \tag{D.1}$$

for the gauge sector, where $U_{\mu\nu}$ is the plaquette at lattice site $x$. For the fermionic part it uses unimproved Wilson fermions for both types of dark quarks,

$$S_F[U, \psi_u, \psi_d, \bar\psi_u, \bar\psi_d] = \sum_{f=u,d} a^4 \sum_{x,y} \bar\psi_f(x) D_f(x|y) \psi_f(m), \tag{D.2}$$

$$D_f(x|y) = \left( m_f + \frac{4}{a} \right) - \frac{1}{2a} \sum_\mu \Big( (1-\gamma_\mu) U_\mu(x) \delta_{y,x+\hat\mu} + (1+\gamma_\mu) U_\mu^\dagger(x-\hat\mu) \delta_{y,x-\hat\mu} \Big). \tag{D.3}$$

This implies that chiral symmetry is explicitly broken at any finite lattice spacing, and is only recovered in the continuum limit. Furthermore, this entails an additive quark mass renormalization. Both effects are well-known, and can be controlled by analyzing the systematic effects, as is done below in appendix D.4. We simulated three different values of the inverse coupling $\beta \geq 6.9$, each of which avoids the unphysical Aoki phase that was found to exist for this theory [7]. Since we are using two distinct fermions the Wilson-Dirac determinant is not necessarily positive definite and in principle a sign problem can arise. Within our parameter range, we did not encounter any indications of a non-positive definite determinant.

### D.2  Masses

We extract the masses of the mesons by determining the exponential decay of the correlation function of a suitable interpolator $O_\Gamma^f(x, y)$ where $x$ and $y$ are sites on the lattice. For the flavoured and unflavoured mesons we choose

$$O_\Gamma^{\bar u d}(x, y) = \bar u(x) \Gamma d(y), \tag{D.4}$$

$$O_\Gamma^0(x, y) = \bar u(x) \Gamma u(y) - \bar d(x) \Gamma d(y). \tag{D.5}$$

We only study the masses and decay constants of the mesons since they are identical to the corresponding diquark states [22]. We denote $x = (\vec x, t)$ and define a general correlator on a lattice of spatial extent $L$ as

$$C_{\Gamma_1 \Gamma_2}(t_x - t_y, \vec p) = \frac{1}{L^3} \sum_{\vec x \vec y} e^{-i\vec p(\vec x - \vec y)} \langle O_{\Gamma_1}(\vec x, t_x) O_{\Gamma_2}^\dagger(\vec y, t_y) \rangle. \tag{D.6}$$

From this the meson masses can be obtained by first setting $\Gamma_1 = \Gamma_2 = \Gamma$ and projecting to zero momentum and studying the limit at large time separation.

$$\lim_{t \to \infty} C_{\Gamma\Gamma}(t, \vec{p} = 0) = \lim_{t \to \infty} \sum_n \frac{1}{2m_n} \langle 0|O_{\Gamma\Gamma}|n\rangle \langle n|O_{\Gamma\Gamma}^{\dagger}|0\rangle e^{-im_n \cdot t} \tag{D.7}$$

$$= \frac{1}{2m_{\Gamma}} |\langle 0|O_{\Gamma\Gamma}|\Gamma_{GS}\rangle|^2 e^{-im_{\Gamma} \cdot t}. \tag{D.8}$$

At large $t$ only the contribution of the lightest state for the quantum numbers of $O_{\Gamma}^f$, $|\Gamma_{GS}\rangle$ survives. The mass of this state can be then extracted, by performing a fit of the lattice data. In the present case the signal-to-noise ratio is very good, and the effective mass shows well-defined plateaus, thus providing for reliable fit results.

The unflavoured mesons sourced by operators of the form (D.4) obtain so-called disconnected contributions which usually suffer from bad statistics. They are however suppressed by both the fermion masses and vanish in the limit of $m_u - m_d \to 0$ and we therefore neglect the disconnected contributions. For operators of the form

$$O_{\Gamma}(x, y) = \bar{u}(x)\Gamma u(y) + \bar{d}(x)\Gamma d(y), \tag{D.9}$$

which source e.g. the $\eta'$ and $\omega$ mesons in QCD this is no longer the case. We do not consider operators of this form in this work.

### D.3 Decay constants

We define the decay constant as in [6] by the matrix elements involving the pseudoscalar $|PS\rangle$ and vector meson $|V\rangle$ ground states

$$\langle 0|O_{\gamma_5\gamma_\mu}^f|PS\rangle = f_{PS}^f p_\mu, \tag{D.10}$$

$$\langle 0|O_{\gamma_\mu}^f|V\rangle = f_V^f m_V \epsilon_\mu. \tag{D.11}$$

In this convention the decay constant of the $\pi$ in QCD is approximately 93 MeV.

Here $\epsilon_\mu$ is the polarisation vector for which $\epsilon_\mu p^\mu = 0$ and $\epsilon_\mu^* \epsilon^\mu = 1$ hold. In the rest frame the pseudoscalar matrix element is then

$$\langle 0|O_{\gamma_5\gamma_0}^f|PS\rangle = f_{PS}^f m_{PS}. \tag{D.12}$$

Inserting this expression into the corresponding correlators at large times gives

$$C_{\Gamma_1=\Gamma_2=\gamma_5\gamma_0}^f(t) = \frac{m_{PS}}{2}\left(f_{PS}^f\right)^2 \exp(-m_{PS} \cdot t), \tag{D.13}$$

$$C_{\Gamma_1=\Gamma_2=\gamma_\mu}^f(t) = \frac{m_V}{2}\left(f_V^f\right)^2 \exp(-m_V \cdot t). \tag{D.14}$$

Alternatively, the mixed correlator with $\Gamma_1 = \gamma_5\gamma_0$ and $\Gamma_2 = \gamma_5$ can be used for the pseudoscalars [6]. From this we can extract the bare decay constants which need to be renormalized. For that we follow the prescription from [6] for determining the renormalization constants defined as

$$f_{PS}^{ren,f} = Z_A f_{PS}^f, \tag{D.15}$$

$$f_V^{ren,f} = Z_V f_V^f. \tag{D.16}$$

### D.4 Systematics

#### D.4.1 Finite volume effects

The finite volume of the 4-dimensional Euclidean space introduces artefacts in the lattice results. We can obtain an estimate of the infinite volume masses by fitting the results obtain at different spatial extents $L$ of the lattice to the known analytic form of the $L$-dependence of the masses. For that we use the same approach as in [6] and fit the function

$$m_{\text{meson}}(L) = m_{\text{meson}}^{\text{inf.}} \left( 1 + A \frac{\exp(-m_{\text{Goldstone}}^{\text{inf}} \cdot L)}{(m_{\text{Goldstone}}^{\text{inf}} \cdot L)^{(3/2)}} \right), \tag{D.17}$$

in the almost mass-degenerate limit, since there the quark masses are the lightest in our setup and therefore the finite volume effects are the largest. We consider $A$ as a free fitting parameter. In figures 8 and 9 we show the infinite volume extrapolation for nearly-degenerate, moderately light ensembles at $\beta = 6.9$ and 7.2 and for a nearly-degenerate ensemble with light fermions at $\beta = 7.2$. Only in the latter ensembles finite volume effects become apparent and generally the deviation from the infinite volume extrapolation stays below 10%.

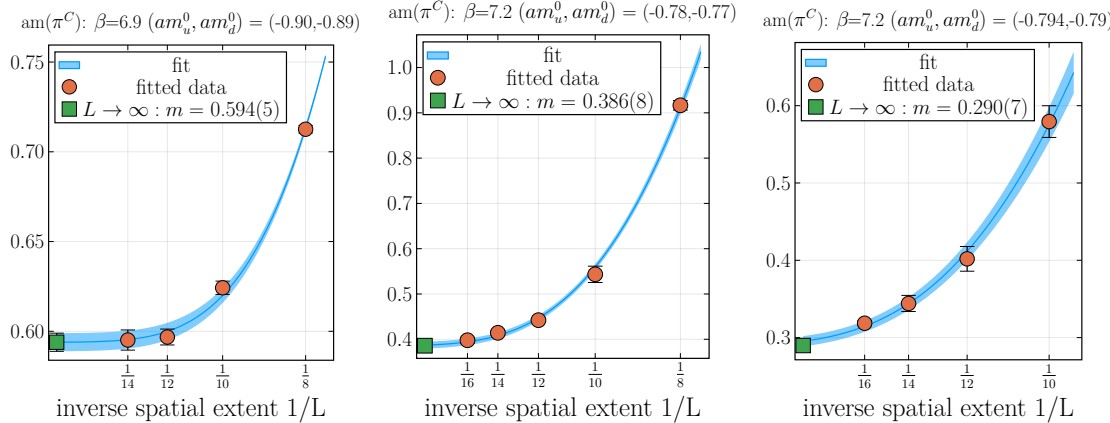

Figure 8: Finite volume extrapolation for the Goldstone mass in the almost mass-degenerate limit. It can be seen that the infinite volume mass and the mass on the largest lattices considered agree within errors. Only in the lightest ensembles on the finest lattices studied here the finite volume effects become apparent. Even in this case the deviation from the infinite volume extrapolation stays below 10%.

#### D.4.2 Finite spacing effects

Apart from the possible systematic errors introduced by the finite volume of the lattice, the finite distance $a$ between two adjacent lattice points is another source of systematic errors. The physical distance $a$ is, however, not an input parameter of simulation. In order to study the systematic effects of $a$ we need to choose input parameters so that the results represent the same physics at different values of $a$. Finding these lines of constant physics in our three-dimensional parameter space is in general not straightforward.

In $SU(3)_c$ gauge theory with fermions it has been found that $a$ depends strongly on the value of the inverse coupling $\beta$ and only weakly on the masses of the fermions $m_f$ [5]. As an first test we have chosen values of the dark fermion masses at the point of degeneracy such that at different values of $\beta$ the ration of the pseudo-Goldstones and the vector mesons is reproduced. We then again increment one of the fermion masses. If the assumption of negligible effects of the fermion masses on $a$ is correct, then the corresponding ratios of Goldstone

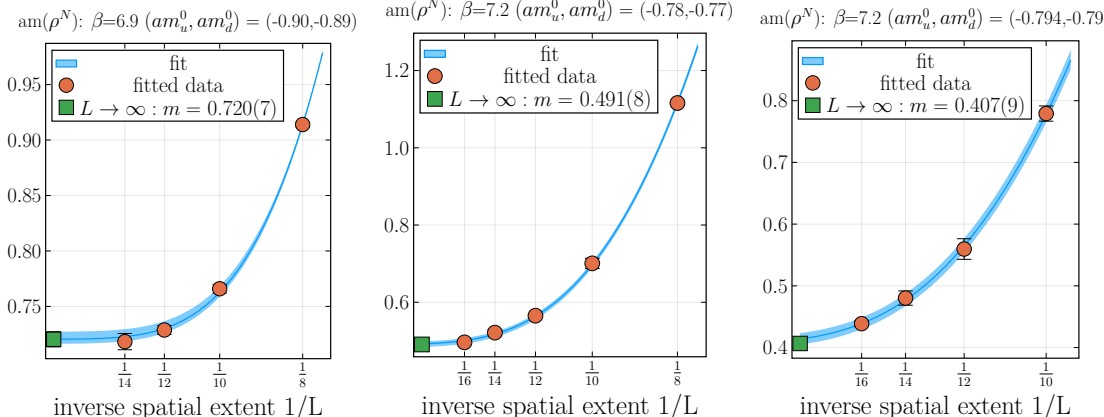

Figure 9: Finite volume extrapolation for the vector meson mass in the almost mass-degenerate limit. The vector meson masses show the same behaviour as the Goldstone bosons.

to vector masses should be the same for all values of $\beta$ since we move then along lines of constant physics in parameter space. Along these lines of constant physics we can then study finite spacing effects of the other observables that do not depend on the lattice spacing, e.g. ratios.

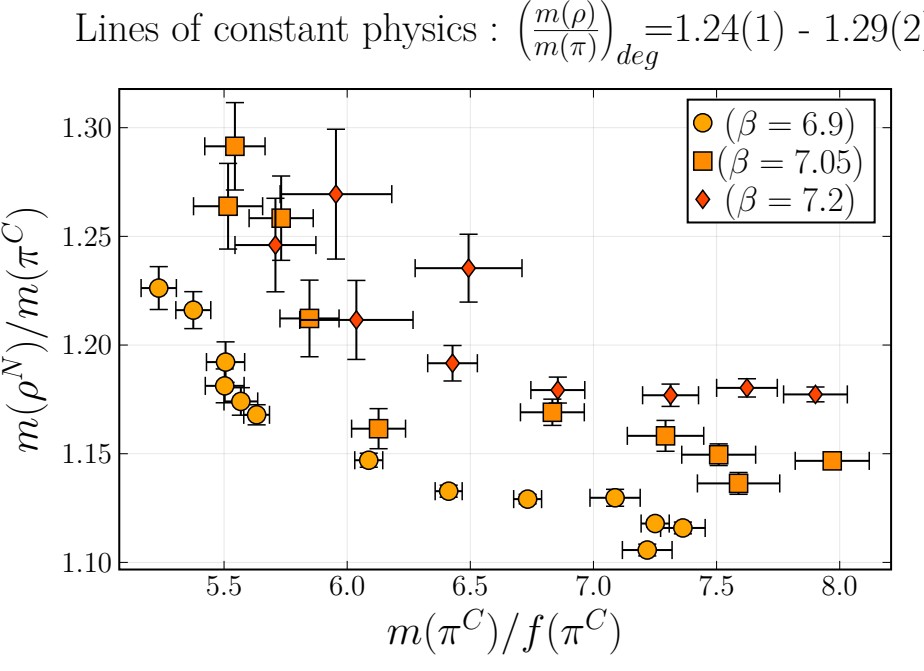

Figure 10: In order to study the effects of the finite lattice spacing we define the lines through parameter space with $m(\rho^N)/m(\pi^C)$ as the lines of constant physics. Here we plot this ratio against the mass of the lightest pseudo-Goldstone in units of its decay constant. We see that the curves at different $\beta$ do not coincide. However, the deviations are only at around 10% and for the two finer lattices they almost coincide.

In figure 10 we plot the ratio of the lightest vector meson's mass to the lightest pseudo-Goldstone's mass $m(\rho^N)/m(\pi^C)$ against the lightest pseudo-Goldstone mass in units of its decay constant. We are not moving exactly along a line of constant physics. However, we

find that the deviations do not significantly exceed 10%. Since this plot involves both the masses and decay constants we plot the masses and decay constants separately in figures 11 and 12 against the masses of the flavoured vector mesons — both in units of the pseudo-Goldstone mass at degeneracy. We find that at different $\beta$ the masses agree within errors and we only see deviations from that behaviour most notably at $\beta = 7.2$ for large flavoured fermion masses of $m(\rho^M)/m_\pi^{\mathrm{deg}} > 2$. For the decay constants in Fig. 12 the deviations are significantly larger and already pronounced at lower masses. For the pseudo-Goldstones the deviations are approximately 10% whereas for the vector mesons they are at around 20%.

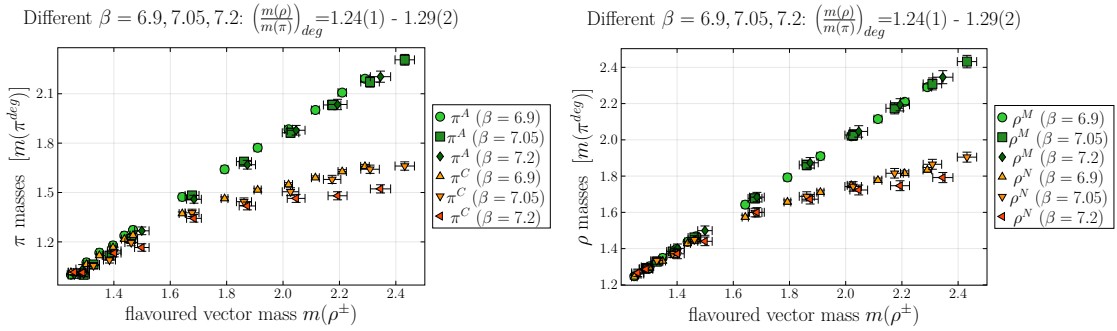

Figure 11: Masses of the pseudo-Goldstones (upper) and vector mesons (lower) for different values of the inverse coupling $\beta$ in units of the pseudo-Goldstone mass degeneracy. The results at different $\beta$ agree within errors except for the $\beta = 7.2$ ensembles with one relatively heavy fermion. From this we conclude that there the finite spacing effects for the meson masses are small.

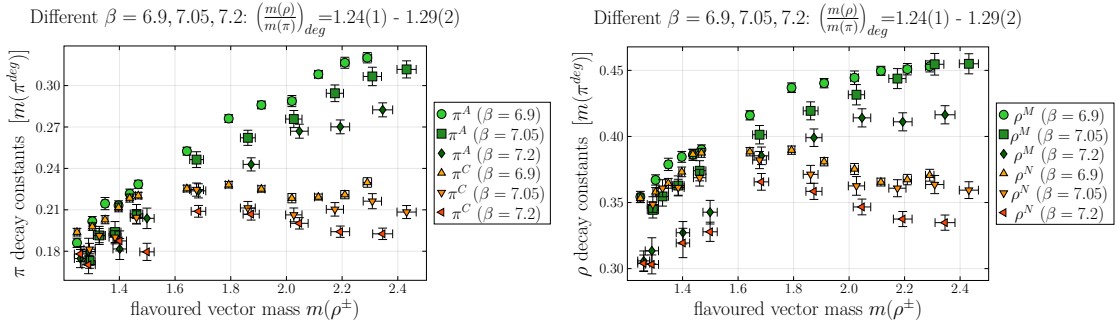

Figure 12: Decay constants of the pseudo-Goldstones (upper) and vector mesons (lower) for different values of the inverse coupling $\beta$ in units of the pseudo-Goldstone mass degeneracy. The results at different $\beta$ show stronger deviations than the meson masses even at very light fermion masses. For the pseudo-Goldstones the deviations are approximately 10% or smaller whereas for the vector mesons they can be as large as 20%.

### D.4.3 Lattice asymmetry

In our analysis we extract the masses and decay constants from the behaviour of correlation functions at large Euclidean time $t$. For convenience we therefore perform simulations on lattices with larger temporal extent than spatial extent, i.e. a lattice of dimensions $L^3 \times T$ with $T > L$. This allows us to calculate the correlation functions at large $t$ while avoiding significantly increased computation time. We have studied the effects of finite $L$ in section D.4.1.

It is, however, still possible, that this introduces systematic error. Here we try to assess whether this has an effect on our results. For this we compare meson masses on symmetric lattices of size $L^4$ to the masses we have obtained for asymmetric lattices. On the symmetric lattices our time extent is substantially smaller and we expect even at the largest Euclidean times to see effects of contamination by states with higher energies. We therefore take the second lightest state into account and extract the mass from the effective mass of the correlator defined as

$$m_{\text{eff}}\left(t + \frac{1}{2}\right) = \frac{C_{O_M}(t)}{C_{O_M}(t+1)}.$$  (D.18)

This allows us to extract the masses from a three-parameter fit while simultaneously minimizing the effects of contamination of higher states. This comes at the cost of larger errors, since the effective mass defined this way discards some information because it as a local quantity around Euclidean time $\left(t + \frac{1}{2}\right)$. As can be seen in table 5 we don't observe any systematic effects from the use of asymmetric lattice within the errors reported.

Table 5: Comparison of meson masses extracted from both symmetric and asymmetric lattices for an inverse coupling of $\beta = 6.9$. The other two input parameters are $m_1^0$ and $m_2^0$ - the unrenormalized bare quark masses. They agree very well within errors and we conclude that there are no relevant systematic effects due to the asymmetric lattices used throughout this work.

| $(-am_1^0, -am_2^0)$ | $am(\pi^A)(24 \times 12^3)$ | $am(\pi^A)(12^4)$ | $am(\pi^C)(24 \times 12^3)$ | $am(\pi^C)(12^4)$ |
|---|---|---|---|---|
| (0.90,0.70) | 0.993(7) | 0.994(15) | 0.83(1) | 0.85(4) |
| (0.90,0.75) | 0.917(14) | 0.92(2) | 0.80(2) | 0.81(5) |
| (0.90,0.80) | 0.826(11) | 0.83(5) | 0.76(2) | 0.77(7) |
| (0.90,0.85) | 0.712(15) | 0.7(2) | 0.69(2) | 0.7(2) |
| (0.90,0.90) | 0.56(4) | 0.58(9) | 0.56(4) | 0.58(9) |
| $(-am_1^0, -am_2^0)$ | $am(\rho^M)(24 \times 12^3)$ | $am(\rho^M)(12^4)$ | $am(\rho^N)(24 \times 12^3)$ | $am(\rho^N)(12^4)$ |
| (0.90,0.70) | 1.070(9) | 1.071(15) | 0.951(14) | 0.97(3) |
| (0.90,0.75) | 1.001(14) | 1.00(2) | 0.92(2) | 0.94(3) |
| (0.90,0.80) | 0.921(15) | 0.93(3) | 0.88(2) | 0.90(3) |
| (0.90,0.85) | 0.82(2) | 0.82(4) | 0.81(2) | 0.82(4) |
| (0.90,0.90) | 0.70(2) | 0.72(4) | 0.70(2) | 0.72(4 |

## D.5 Tabulated lattice results

Table 6: Tabulated results for the ensembles with $m_\rho/m_\pi \approx 1.25$. We report the input parameters the bare inverse gauge coupling $\beta$, the bare fermion masses $am_u^0$ and $am_d^0$ — $a$ is the lattice spacing — as well as the lattice dimensions $T \times L^3$. The reported decay constants have been renormalized and in addition we give the average plaquette $\langle P \rangle$ that is used in the renormalization. In the last column we give the average PCAC-mass.

| $\beta$ | $T$ | $L$ | $am_u^0$ | $am_d^0$ | $\langle P \rangle$ | $am(\pi^A)$ | $am(\pi^C)$ | $af(\pi^A)$ | $af(\pi^C)$ | $am(\rho^M)$ | $am(\rho^N)$ | $af(\rho^M)$ | $af(\rho^N)$ | $a\bar{m}_{\mathrm{PCAC}}$ |
|---|---|---|---|---|---|---|---|---|---|---|---|---|---|---|
| 6.9 | 24 | 12 | -0.9 | -0.5 | 0.5333(2) | 1.228(3) | 0.930(3) | 0.179(2) | 0.129(2) | 1.284(2) | 1.028(2) | 0.254(2) | 0.208(2) | 0.372(7) |
| 6.9 | 24 | 12 | -0.9 | -0.55 | 0.5345(2) | 1.181(2) | 0.912(3) | 0.177(2) | 0.124(2) | 1.239(2) | 1.018(2) | 0.253(2) | 0.206(2) | 0.340(6) |
| 6.9 | 24 | 12 | -0.9 | -0.6 | 0.53655(8) | 1.122(1) | 0.891(2) | 0.173(1) | 0.1229(9) | 1.185(1) | 0.996(2) | 0.252(1) | 0.205(1) | 0.306(3) |
| 6.9 | 24 | 12 | -0.9 | -0.65 | 0.5386(1) | 1.056(3) | 0.868(3) | 0.162(2) | 0.122(2) | 1.132(3) | 0.980(3) | 0.249(2) | 0.210(2) | 0.270(6) |
| 6.9 | 24 | 14 | -0.9 | -0.7 | 0.54098(8) | 0.993(2) | 0.849(2) | 0.160(1) | 0.126(1) | 1.070(1) | 0.959(2) | 0.247(1) | 0.213(1) | 0.236(3) |
| 6.9 | 24 | 14 | -0.9 | -0.75 | 0.54393(8) | 0.920(2) | 0.820(2) | 0.155(1) | 0.128(1) | 1.005(2) | 0.929(2) | 0.245(1) | 0.218(1) | 0.200(3) |
| 6.9 | 24 | 14 | -0.9 | -0.8 | 0.54736(8) | 0.826(2) | 0.769(2) | 0.142(1) | 0.126(1) | 0.921(2) | 0.882(2) | 0.233(1) | 0.218(1) | 0.163(3) |
| 6.9 | 24 | 14 | -0.9 | -0.85 | 0.55163(7) | 0.713(2) | 0.695(2) | 0.128(1) | 0.123(1) | 0.823(2) | 0.812(2) | 0.219(1) | 0.217(1) | 0.122(3) |
| 6.9 | 24 | 14 | -0.9 | -0.86 | 0.5523(1) | 0.694(3) | 0.682(3) | 0.124(2) | 0.122(1) | 0.805(2) | 0.801(2) | 0.216(2) | 0.217(1) | 0.110(3) |
| 6.9 | 24 | 14 | -0.9 | -0.87 | 0.5538(1) | 0.660(4) | 0.653(3) | 0.119(2) | 0.119(2) | 0.783(3) | 0.771(3) | 0.216(2) | 0.209(2) | 0.103(3) |
| 6.9 | 24 | 14 | -0.9 | -0.88 | 0.5547(1) | 0.636(4) | 0.626(3) | 0.120(2) | 0.114(2) | 0.756(3) | 0.746(3) | 0.212(2) | 0.204(2) | 0.096(3) |
| 6.9 | 24 | 14 | -0.9 | -0.89 | 0.55582(9) | 0.603(4) | 0.596(3) | 0.113(2) | 0.111(1) | 0.730(2) | 0.725(2) | 0.206(2) | 0.201(1) | 0.086(3) |
| 6.9 | 24 | 14 | -0.9 | -0.9 | 0.5569(2) | 0.560(4) | 0.569(3) | 0.104(2) | 0.109(1) | 0.699(3) | 0.697(2) | 0.199(2) | 0.198(1) | 0.077(3) |
| 7.05 | 24 | 12 | -0.835 | -0.5 | 0.5603(1) | 1.025(3) | 0.738(5) | 0.139(2) | 0.093(2) | 1.081(3) | 0.847(3) | 0.202(2) | 0.160(2) | 0.274(7) |
| 7.05 | 24 | 12 | -0.835 | -0.55 | 0.5618(2) | 0.965(4) | 0.730(5) | 0.136(2) | 0.096(2) | 1.026(3) | 0.829(4) | 0.202(2) | 0.162(2) | 0.248(6) |
| 7.05 | 24 | 12 | -0.835 | -0.6 | 0.5636(1) | 0.903(3) | 0.702(5) | 0.131(2) | 0.094(2) | 0.967(3) | 0.807(3) | 0.197(2) | 0.160(2) | 0.216(6) |
| 7.05 | 24 | 12 | -0.835 | -0.65 | 0.5654(1) | 0.828(4) | 0.669(5) | 0.123(2) | 0.092(2) | 0.901(4) | 0.775(4) | 0.192(2) | 0.161(2) | 0.182(6) |
| 7.05 | 24 | 14 | -0.835 | -0.7 | 0.5678(1) | 0.750(3) | 0.642(5) | 0.117(2) | 0.094(2) | 0.828(3) | 0.751(4) | 0.186(2) | 0.165(2) | 0.149(4) |
| 7.05 | 24 | 14 | -0.835 | -0.75 | 0.5700(1) | 0.659(4) | 0.612(5) | 0.110(2) | 0.100(2) | 0.746(4) | 0.711(4) | 0.178(2) | 0.170(2) | 0.118(4) |
| 7.05 | 24 | 14 | -0.835 | -0.8 | 0.5731(1) | 0.542(7) | 0.532(4) | 0.092(3) | 0.091(2) | 0.650(4) | 0.644(4) | 0.166(2) | 0.164(2) | 0.079(4) |
| 7.05 | 24 | 14 | -0.835 | -0.82 | 0.5742(1) | 0.496(7) | 0.485(4) | 0.086(3) | 0.085(2) | 0.615(4) | 0.610(4) | 0.161(2) | 0.161(2) | 0.064(4) |
| 7.05 | 24 | 14 | -0.835 | -0.83 | 0.5749(1) | 0.472(6) | 0.469(5) | 0.085(3) | 0.085(2) | 0.590(4) | 0.593(4) | 0.158(2) | 0.160(2) | 0.061(4) |
| 7.05 | 24 | 14 | -0.835 | -0.835 | 0.5755(2) | 0.445(6) | 0.447(5) | 0.077(2) | 0.081(2) | 0.575(4) | 0.578(3) | 0.153(2) | 0.155(2) | 0.051(3) |
| 7.2 | 24 | 14 | -0.78 | -0.5 | 0.58108(7) | 0.842(2) | 0.582(4) | 0.108(1) | 0.074(1) | 0.897(2) | 0.685(2) | 0.159(1) | 0.128(1) | 0.211(4) |
| 7.2 | 24 | 14 | -0.78 | -0.55 | 0.58222(7) | 0.778(2) | 0.566(4) | 0.103(1) | 0.074(1) | 0.838(2) | 0.668(2) | 0.157(1) | 0.129(1) | 0.181(4) |
| 7.2 | 24 | 14 | -0.78 | -0.6 | 0.58342(7) | 0.718(3) | 0.560(4) | 0.102(1) | 0.077(1) | 0.782(2) | 0.659(3) | 0.158(1) | 0.133(1) | 0.153(4) |
| 7.2 | 24 | 14 | -0.78 | -0.65 | 0.58484(6) | 0.638(3) | 0.542(4) | 0.093(1) | 0.079(1) | 0.716(2) | 0.639(3) | 0.153(1) | 0.137(1) | 0.127(3) |
| 7.2 | 24 | 14 | -0.78 | -0.7 | 0.58634(6) | 0.558(3) | 0.513(4) | 0.086(2) | 0.080(1) | 0.644(2) | 0.612(3) | 0.147(1) | 0.140(1) | 0.097(3) |
| 7.2 | 24 | 14 | -0.78 | -0.73 | 0.58743(6) | 0.484(5) | 0.446(5) | 0.078(3) | 0.069(2) | 0.573(5) | 0.551(4) | 0.131(3) | 0.125(2) | 0.078(4) |
| 7.2 | 32 | 16 | -0.78 | -0.75 | 0.58818(6) | 0.438(5) | 0.433(6) | 0.069(3) | 0.072(3) | 0.536(4) | 0.524(7) | 0.125(3) | 0.122(4) | 0.064(4) |
| 7.2 | 32 | 16 | -0.78 | -0.77 | 0.58892(6) | 0.396(8) | 0.387(6) | 0.069(4) | 0.065(2) | 0.493(5) | 0.492(4) | 0.120(3) | 0.116(2) | 0.054(4) |
| 7.2 | 32 | 16 | -0.78 | -0.78 | 0.58923(4) | 0.382(6) | 0.389(5) | 0.067(2) | 0.068(2) | 0.482(4) | 0.484(3) | 0.117(2) | 0.116(2) | 0.049(3) |

Table 7: Tabulated results for the ensembles with $m_\rho/m_\pi \approx 1.4$. We report the input parameters the bare inverse gauge coupling $\beta$, the bare fermion masses $am_u^0$ and $am_d^0$ and $am_u^0$ — $a$ is the lattice spacing — as well as the lattice dimensions $T \times L^3$. The reported decay constants have been renormalized and in addition we give the average plaquette $\langle P \rangle$ that is used in the renormalization. In the last column we give the average PCAC-mass.

| $\beta$ | $T$ | $L$ | $am_u^0$ | $am_d^0$ | $\langle P \rangle$ | $am(\pi^A)$ | $am(\pi^C)$ | $af(\pi^A)$ | $af(\pi^C)$ | $am(\rho^M)$ | $am(\rho^N)$ | $af(\rho^M)$ | $af(\rho^N)$ | $a\bar{m}_{PCAC}$ |
|---|---|---|---|---|---|---|---|---|---|---|---|---|---|---|
| 6.9 | 24 | 12 | -0.92 | -0.5 | 0.5349(2) | 1.198(3) | 0.871(5) | 0.179(2) | 0.118(2) | 1.253(3) | 0.985(4) | 0.252(2) | 0.202(4) | 0.358(9) |
| 6.9 | 24 | 12 | -0.92 | -0.55 | 0.5364(2) | 1.144(4) | 0.857(5) | 0.169(3) | 0.116(3) | 1.204(4) | 0.967(4) | 0.245(3) | 0.200(3) | 0.3147(7) |
| 6.9 | 24 | 12 | -0.92 | -0.6 | 0.5385(2) | 1.084(4) | 0.832(6) | 0.167(3) | 0.118(3) | 1.154(3) | 0.945(5) | 0.249(3) | 0.201(3) | 0.28(1) |
| 6.9 | 24 | 12 | -0.92 | -0.65 | 0.5399(2) | 1.031(4) | 0.807(7) | 0.164(2) | 0.115(3) | 1.107(3) | 0.927(6) | 0.249(2) | 0.200(3) | 0.2577(8) |
| 6.9 | 24 | 12 | -0.92 | -0.7 | 0.5427(2) | 0.958(5) | 0.779(7) | 0.155(3) | 0.117(3) | 1.040(4) | 0.900(5) | 0.244(3) | 0.203(3) | 0.2179(9) |
| 6.9 | 24 | 12 | -0.92 | -0.75 | 0.5459(2) | 0.870(5) | 0.752(7) | 0.143(3) | 0.118(3) | 0.959(4) | 0.870(5) | 0.233(3) | 0.207(3) | 0.180(8) |
| 6.9 | 24 | 12 | -0.92 | -0.8 | 0.5498(2) | 0.765(6) | 0.669(9) | 0.133(3) | 0.110(3) | 0.867(4) | 0.810(7) | 0.223(4) | 0.204(4) | 0.143(8) |
| 6.9 | 24 | 14 | -0.92 | -0.85 | 0.5538(1) | 0.660(5) | 0.632(5) | 0.121(2) | 0.114(2) | 0.782(4) | 0.756(4) | 0.216(2) | 0.204(2) | 0.106(4) |
| 6.9 | 24 | 14 | -0.92 | -0.88 | 0.5572(3) | 0.55(1) | 0.540(7) | 0.101(4) | 0.098(3) | 0.690(6) | 0.687(6) | 0.195(3) | 0.194(3) | 0.073(5) |
| 6.9 | 24 | 14 | -0.92 | -0.9 | 0.5594(1) | 0.480(9) | 0.486(6) | 0.090(3) | 0.094(2) | 0.637(6) | 0.646(4) | 0.184(3) | 0.189(2) | 0.057(3) |
| 6.9 | 24 | 14 | -0.92 | -0.91 | 0.5607(1) | 0.45(1) | 0.441(6) | 0.092(4) | 0.088(3) | 0.605(7) | 0.598(5) | 0.183(3) | 0.176(2) | 0.048(4) |
| 6.9 | 24 | 14 | -0.92 | -0.92 | 0.5621(2) | 0.38(1) | 0.380(8) | 0.083(4) | 0.082(3) | 0.548(9) | 0.555(6) | 0.167(4) | 0.168(3) | 0.038(4) |
| 7.05 | 24 | 14 | -0.85 | -0.5 | 0.5614(1) | 0.999(4) | 0.690(4) | 0.135(2) | 0.086(2) | 1.059(3) | 0.805(4) | 0.201(2) | 0.155(2) | 0.261(8) |
| 7.05 | 24 | 14 | -0.85 | -0.55 | 0.5628(1) | 0.943(3) | 0.669(6) | 0.130(2) | 0.086(2) | 1.001(3) | 0.789(4) | 0.194(2) | 0.156(2) | 0.233(7) |
| 7.05 | 24 | 14 | -0.85 | -0.6 | 0.5648(1) | 0.875(4) | 0.662(7) | 0.128(2) | 0.091(2) | 0.943(3) | 0.772(5) | 0.195(3) | 0.158(2) | 0.204(7) |
| 7.05 | 24 | 14 | -0.85 | -0.65 | 0.5666(1) | 0.791(5) | 0.615(5) | 0.118(2) | 0.089(2) | 0.872(4) | 0.736(4) | 0.188(3) | 0.159(3) | 0.173(6) |
| 7.05 | 24 | 14 | -0.85 | -0.7 | 0.5685(1) | 0.730(5) | 0.602(5) | 0.116(3) | 0.088(2) | 0.805(4) | 0.715(4) | 0.185(2) | 0.160(2) | 0.138(6) |
| 7.05 | 24 | 14 | -0.85 | -0.75 | 0.5712(2) | 0.620(5) | 0.554(7) | 0.101(2) | 0.088(2) | 0.717(4) | 0.676(4) | 0.172(2) | 0.163(2) | 0.106(5) |
| 7.05 | 24 | 14 | -0.85 | -0.8 | 0.5739(1) | 0.501(5) | 0.481(5) | 0.089(2) | 0.083(2) | 0.621(3) | 0.606(4) | 0.163(2) | 0.157(2) | 0.072(3) |
| 7.05 | 24 | 14 | -0.85 | -0.83 | 0.57590(8) | 0.427(8) | 0.422(6) | 0.078(2) | 0.080(2) | 0.563(5) | 0.556(4) | 0.155(2) | 0.153(2) | 0.047(3) |
| 7.05 | 24 | 14 | -0.85 | -0.85 | 0.577736(7) | 0.350(8) | 0.348(6) | 0.067(2) | 0.066(2) | 0.510(4) | 0.518(3) | 0.143(2) | 0.145(1) | 0.035(2) |
| 7.2 | 24 | 14 | -0.794 | -0.45 | 0.5808(1) | 0.874(4) | 0.550(7) | 0.108(2) | 0.070(3) | 0.925(4) | 0.654(5) | 0.158(2) | 0.122(2) | 0.227(7) |
| 7.2 | 24 | 14 | -0.794 | -0.5 | 0.5816(1) | 0.823(5) | 0.535(8) | 0.107(2) | 0.071(2) | 0.879(4) | 0.647(5) | 0.158(2) | 0.125(2) | 0.201(7) |
| 7.2 | 24 | 14 | -0.794 | -0.55 | 0.5829(1) | 0.755(4) | 0.506(8) | 0.100(2) | 0.066(3) | 0.819(3) | 0.633(5) | 0.155(2) | 0.128(2) | 0.172(7) |
| 7.2 | 24 | 14 | -0.794 | -0.6 | 0.58408(5) | 0.686(2) | 0.510(4) | 0.095(1) | 0.070(1) | 0.755(2) | 0.618(3) | 0.151(1) | 0.126(1) | 0.144(3) |
| 7.2 | 24 | 14 | -0.794 | -0.65 | 0.585543(6) | 0.613(3) | 0.488(4) | 0.090(1) | 0.071(1) | 0.693(2) | 0.604(3) | 0.150(1) | 0.130(1) | 0.117(3) |
| 7.2 | 24 | 14 | -0.794 | -0.7 | 0.58696(6) | 0.529(3) | 0.460(5) | 0.081(1) | 0.072(1) | 0.620(2) | 0.577(3) | 0.142(1) | 0.135(1) | 0.086(3) |
| 7.2 | 32 | 16 | -0.794 | -0.75 | 0.58878(5) | 0.420(4) | 0.396(5) | 0.071(2) | 0.066(2) | 0.522(9) | 0.51(1) | 0.124(6) | 0.119(7) | 0.059(2) |
| 7.2 | 32 | 16 | -0.794 | -0.77 | 0.5895(1) | 0.370(7) | 0.345(6) | 0.064(3) | 0.058(2) | 0.477(4) | 0.470(4) | 0.117(2) | 0.117(2) | 0.043(3) |
| 7.2 | 32 | 16 | -0.794 | -0.78 | 0.58988(6) | 0.324(8) | 0.332(7) | 0.059(3) | 0.062(2) | 0.433(6) | 0.44(1) | 0.107(3) | 0.110(6) | 0.039(3) |
| 7.2 | 32 | 16 | -0.794 | -0.794 | 0.590332(6) | 0.315(8) | 0.317(5) | 0.059(3) | 0.059(2) | 0.430(5) | 0.426(4) | 0.110(3) | 0.106(2) | 0.032(2) |

Table 8: Tabulated results for the ensembles with $m_\rho/m_\pi \approx 1.15$. We report the input parameters the bare inverse gauge coupling $\beta$, the bare fermion masses $am_u^0$ and $am_u^0$ — $a$ is the lattice spacing — as well as the lattice dimensions $T \times L^3$. The reported decay constants have been renormalized and in addition we give the average plaquette $\langle P \rangle$ that is used in the renormalization. In the last column we give the average PCAC-mass.

| $\beta$ | $T$ | $L$ | $am_u^0$ | $am_d^0$ | $\langle P \rangle$ | $am(\pi^A)$ | $am(\pi^C)$ | $af(\pi^A)$ | $af(\pi^C)$ | $am(\rho^M)$ | $am(\rho^N)$ | $af(\rho^M)$ | $af(\rho^N)$ | $a\bar{m}_{PCAC}$ |
|---|---|---|---|---|---|---|---|---|---|---|---|---|---|---|
| 6.9 | 24 | 12 | -0.87 | -0.5 | 0.5309(1) | 1.268(2) | 1.004(3) | 0.185(1) | 0.132(1) | 1.322(2) | 1.093(2) | 0.258(1) | 0.207(2) | 0.400(5) |
| 6.9 | 24 | 12 | -0.87 | -0.55 | 0.5326(1) | 1.219(2) | 0.992(2) | 0.180(2) | 0.134(1) | 1.278(2) | 1.084(2) | 0.257(2) | 0.213(1) | 0.361(6) |
| 6.9 | 24 | 12 | -0.87 | -0.6 | 0.5341(1) | 1.173(2) | 0.980(2) | 0.181(2) | 0.135(1) | 1.232(2) | 1.071(2) | 0.259(2) | 0.215(1) | 0.334(5) |
| 6.9 | 24 | 12 | -0.87 | -0.65 | 0.5361(1) | 1.115(2) | 0.958(3) | 0.177(2) | 0.136(1) | 1.181(2) | 1.051(2) | 0.259(2) | 0.218(2) | 0.300(5) |
| 6.9 | 24 | 12 | -0.87 | -0.7 | 0.5384(1) | 1.047(2) | 0.939(3) | 0.166(2) | 0.138(2) | 1.120(2) | 1.033(2) | 0.251(2) | 0.222(2) | 0.260(5) |
| 6.9 | 24 | 12 | -0.87 | -0.75 | 0.5411(1) | 0.980(3) | 0.910(3) | 0.162(2) | 0.141(2) | 1.057(2) | 1.004(2) | 0.248(2) | 0.230(2) | 0.226(5) |
| 6.9 | 24 | 12 | -0.87 | -0.8 | 0.5444(1) | 0.896(3) | 0.870(3) | 0.152(2) | 0.143(2) | 0.986(3) | 0.966(2) | 0.244(2) | 0.235(1) | 0.187(5) |
| 6.9 | 24 | 12 | -0.87 | -0.83 | 0.5470(2) | 0.829(5) | 0.818(4) | 0.143(3) | 0.138(2) | 0.925(4) | 0.913(3) | 0.234(3) | 0.228(3) | 0.164(7) |
| 6.9 | 24 | 12 | -0.87 | -0.84 | 0.5478(2) | 0.812(3) | 0.801(4) | 0.142(2) | 0.138(2) | 0.913(2) | 0.903(3) | 0.237(2) | 0.231(2) | 0.156(5) |
| 6.9 | 24 | 12 | -0.87 | -0.85 | 0.5485(1) | 0.790(4) | 0.786(3) | 0.139(2) | 0.135(2) | 0.889(3) | 0.887(2) | 0.230(2) | 0.226(2) | 0.150(4) |
| 6.9 | 24 | 12 | -0.87 | -0.86 | 0.5497(1) | 0.770(4) | 0.764(3) | 0.134(2) | 0.134(2) | 0.872(3) | 0.871(2) | 0.227(2) | 0.229(2) | 0.137(5) |
| 6.9 | 24 | 12 | -0.87 | -0.87 | 0.5504(2) | 0.742(2) | 0.743(2) | 0.131(2) | 0.13303(9) | 0.848(2) | 0.851(2) | 0.224(2) | 0.224(1) | 0.129(3) |
| 7.05 | 24 | 12 | -0.8 | -0.45 | 0.5566(1) | 1.137(4) | 0.862(4) | 0.151(3) | 0.108(2) | 1.184(3) | 0.949(3) | 0.211(2) | 0.171(2) | 0.340(9) |
| 7.05 | 24 | 12 | -0.8 | -0.5 | 0.5581(2) | 1.080(4) | 0.841(4) | 0.143(2) | 0.102(2) | 1.132(3) | 0.930(4) | 0.206(2) | 0.167(2) | 0.304(9) |
| 7.05 | 24 | 12 | -0.8 | -0.55 | 0.55938(9) | 1.029(2) | 0.833(2) | 0.144(1) | 0.106(1) | 1.084(2) | 0.918(2) | 0.209(1) | 0.170(1) | 0.277(4) |
| 7.05 | 24 | 12 | -0.8 | -0.6 | 0.5611(1) | 0.968(3) | 0.816(3) | 0.139(2) | 0.105(1) | 1.031(2) | 0.906(2) | 0.207(2) | 0.174(1) | 0.244(5) |
| 7.05 | 24 | 12 | -0.8 | -0.65 | 0.56321(9) | 0.895(2) | 0.799(2) | 0.133(1) | 0.112(1) | 0.963(2) | 0.882(2) | 0.204(1) | 0.180(1) | 0.213(4) |
| 7.05 | 24 | 12 | -0.8 | -0.7 | 0.56537(9) | 0.819(3) | 0.769(3) | 0.127(1) | 0.114(1) | 0.893(2) | 0.852(2) | 0.197(1) | 0.185(1) | 0.178(4) |
| 7.05 | 24 | 12 | -0.8 | -0.75 | 0.56774(9) | 0.733(3) | 0.723(3) | 0.117(2) | 0.115(1) | 0.818(2) | 0.809(2) | 0.190(2) | 0.188(1) | 0.146(4) |
| 7.05 | 24 | 12 | -0.8 | -0.78 | 0.5695(1) | 0.670(3) | 0.672(2) | 0.111(2) | 0.109(1) | 0.762(3) | 0.764(2) | 0.184(2) | 0.182(1) | 0.123(4) |
| 7.05 | 24 | 12 | -0.8 | -0.8 | 0.5708(1) | 0.626(6) | 0.632(3) | 0.103(2) | 0.106(1) | 0.724(4) | 0.732(3) | 0.176(2) | 0.183(1) | 0.105(4) |
| 7.2 | 24 | 12 | -0.75 | -0.45 | 0.5787(1) | 0.950(4) | 0.696(5) | 0.117(2) | 0.083(2) | 1.002(3) | 0.780(5) | 0.170(2) | 0.134(2) | 0.262(8) |
| 7.2 | 24 | 12 | -0.75 | -0.5 | 0.5797(2) | 0.903(4) | 0.681(7) | 0.120(2) | 0.083(2) | 0.950(3) | 0.763(5) | 0.172(2) | 0.134(2) | 0.236(7) |
| 7.2 | 24 | 12 | -0.75 | -0.55 | 0.58086(7) | 0.835(2) | 0.670(3) | 0.112(1) | 0.0846(9) | 0.891(2) | 0.755(2) | 0.166(1) | 0.1398(9) | 0.203(4) |
| 7.2 | 24 | 12 | -0.75 | -0.6 | 0.58208(6) | 0.767(2) | 0.653(3) | 0.105(1) | 0.0869(9) | 0.829(2) | 0.736(2) | 0.1619(9) | 0.1421(9) | 0.174(3) |
| 7.2 | 24 | 12 | -0.75 | -0.65 | 0.58353(6) | 0.695(3) | 0.638(3) | 0.100(1) | 0.0895(9) | 0.766(2) | 0.720(2) | 0.159(1) | 0.148(1) | 0.145(3) |
| 7.2 | 24 | 12 | -0.75 | -0.7 | 0.58512(7) | 0.616(4) | 0.597(3) | 0.094(1) | 0.090(1) | 0.697(3) | 0.691(2) | 0.155(1) | 0.154(1) | 0.117(3) |
| 7.2 | 24 | 12 | -0.75 | -0.72 | 0.58586(7) | 0.582(3) | 0.580(3) | 0.089(1) | 0.088(1) | 0.674(2) | 0.667(2) | 0.153(1) | 0.1511(9) | 0.103(3) |
| 7.2 | 24 | 12 | -0.75 | -0.74 | 0.58661(7) | 0.547(4) | 0.546(3) | 0.086(1) | 0.084(1) | 0.640(2) | 0.644(2) | 0.150(1) | 0.150(1) | 0.092(3) |
| 7.2 | 24 | 12 | -0.75 | -0.75 | 0.58690(6) | 0.532(4) | 0.530(3) | 0.084(2) | 0.0832(8) | 0.624(2) | 0.636(2) | 0.147(1) | 0.1505(8) | 0.086(3) |

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
