# Peer review of "Low-energy effective description of dark $Sp(4)$ theories"

_SciPost Physics, doi:SciPost Phys. 14, 044 (2023)_

## Round 1 · Referee Report · Anonymous (Referee 1) · 2022-5-24

Strengths

1.In the paper, a variant of the SM extension with the use of the symplectic symmetry group, Sp(4), is formulated in sufficient detail and mathematically accurately. 2. Some estimates of the mass spectrum of the model obtained in lattice calculations are presented. 3. In the work, dark quarks are not representations of the EW group, but are connected to the SM through a new Abelian vector field that mixes with ordinary gauge fields (so-called “vector portal”). An explicit splitting of the quark masses is introduced, and the phenomenology of dark mesons is proposed to be considered within the framework of an analogue of the chiral perturbation theory of QCD, i.e. nonlinear implementation of the global symmetry of the model.

Weaknesses

1.At the same time, the “vector portal” phenomenology is long and well studied in a bulk of papers. 2.The effective theory, which is discussed in the first five sections, is standard for this science. Apart from taking into account the effects of explicit symmetry breaking by the splitting of quark masses, nothing new has been done here. Exactly the same nonlinear chiral effective theory was built before. (Pseudo)vector mesons were taken into account in the same way in the framework of this scheme; pseudo-goldstones and eta meson - even more so. The classification of states is common for any implementation of symmetry. Accounting for the anomaly through the WZW interaction was done. The new vector field also does not represent a big innovation, since technically, for the effective Lagrangian there is no difference what kind of abelian field is there - new or the same as that already exists in the SM. 3. Although the authors refer to work [7], but also in [11] the WZW-term was already considered, as in other papers, so it should be noted the obvious negligence of the authors in providing links and discussing results in similar scenarios obtained by other authors. 4.Similarly, the lack of completeness of the reference list is also evident when reading peppy, but somewhat premature statements about future research on phenomenology in scenarios of this type, despite the fact that specific predictions of dark matter effects have long been studied in the Sp(4) extension of the SM. However, there are no references to these works (except for review [32]) in the article.

Report

I hope that the authors will be able to conduct a more detailed and accurate analysis of the results obtained by other authors and expand the list of references. They will also be able to indicate, at least in advance, which physical effects and manifestations of the dark matter candidates they propose can be observed in direct experiments or indirect data in various energy regions.
It would also be interesting to see the effects of mixing new composite scalars with the Higgs taken into account within this scheme.
The article as a whole satisfies the criteria for publication in this journal and can be published after the specified additions and revision of the text.

Requested changes

As already noted, it is recommended to carry out a more complete analysis of the results obtained earlier 1. both in the study of the mathematical structure of symplectic extensions with a vector portal 2. and in the discussion of the phenomenological manifestations of dark matter and the possibility of their direct or indirect detection. So, the list of references should be extended. 3. Some preliminary estimations of possible phenomenological manifestations of the DM candidates are also desirable, as is the analysis of scalar-Higgs boson mixing.

---

## Round 1 · Referee Report · Ennio Salvioni (Referee 2) · 2022-5-29

Report

First of all, I wish to apologize to the authors for the delay in providing this report.

In this manuscript the authors study a dark $Sp(4)_c$ gauge theory with $N_f = 2$ fermions in the fundamental representation, which is a promising and fairly minimal candidate to accommodate Strongly Interacting Massive Particle (SIMP) dark matter. They discuss in detail the effective chiral Lagrangian for the pseudo-Goldstone bosons (where the pattern of spontaneous symmetry breaking is $SU(4)/Sp(4)$), including the anomalous Wess-Zumino/Witten terms as well as the lightest spin-1 resonances. They do so both for mass degenerate and mass non-degenerate ($m_u \neq m_d$) quarks. They also analyze the gauging of a dark $U(1)'$ symmetry, which often plays an important role in SIMP models where the associated gauge boson (dark photon) acts as mediator transferring entropy from the dark to the visible sector. Finally, in the last section they provide results of a lattice study of this theory in the case $m_u \neq m_d$ (the degenerate case had been studied before in Ref. [6]), which allows them to determine the low-energy constants and the range of validity of the chiral effective theory from first-principles calculations in the underlying $Sp(4)_c$ gauge/fermion theory. A long series of technical appendices completes the paper; Appendix D is especially useful, providing details on the lattice setup and the analysis of systematic effects.

SIMPs are compelling candidates for particle dark matter, and the fact that they find 'natural' embedding into dark QCD-like theories implies the need to confront the challenges inherent to strong dynamics in order to assess the viability of concrete models. In my opinion, the present manuscript does achieve the goal stated by the authors: by performing a thorough and self-consistent study of the $Sp(4)_c$ model, including genuinely novel aspects such as non-degenerate dark quark masses, it potentially opens the way for future studies in dark matter model building and phenomenology. In addition, the text is clearly written. As a result, I believe the paper deserves to be accepted for publication in SciPost Physics. However, there are a number of questions or comments that I ask the authors to consider in a revised version, with the goal of improving the quality of presentation:

Requested changes

1] One question with an eye to SIMP phenomenology. One of the main results of this work is the detailed study of the $m_u \neq m_d$ scenario, where the flavor symmetry is explicitly broken to $SU(2)_u \times SU(2)_d$, under which the diagonal Goldstone $\pi^C$ transforms as a singlet. The $\pi^C$ is also the lightest meson, as shown in section 6. From the discussion in Sec. 4 (page 17) I understand that $\pi^C$ is always unstable once a dark $U(1)'$ is gauged, regardless of the choice of the charge matrix $\mathcal{Q}$. If this is correct, then rendering such scenario compatible with cosmological/astrophysical constraints would seem to be an important challenge, deserving to be at least mentioned here. Do the authors agree with my reading?

2] A few comments regarding references. a) On the last line of page 1, [22] is cited for 'including dark vector resonances into the SIMP paradigm', however it seems to me that [33] deserves to be included here as well. b) On page 9, when introducing the $\pi^5$ interaction only the Witten paper [39] is cited, but I think it would be fair to also include the classic work by Wess and Zumino. c) While the authors have in mind the application to (SIMP) dark matter, in the broader context of the Introduction it may be worthwhile to reference also previous papers that studied the $SU(4)/Sp(4)$ coset in the context of electroweak symmetry breaking, such as 0902.1483 by Gripaios et al. and 1001.1361 by Galloway et al. This last point is merely a suggestion, which the authors are free to accept or decline.

3] I was somewhat confused by the treatment of the pseudoscalar singlet $\eta$ in section 3.2. In the QCD case one can add the $\eta'$ along similar lines, by extending ChPT to $U(3)_L \times U(3)_R$, however a 'hard breaking' mass term $\delta \mathcal{L} = - m_0^2 \eta^{\prime\,2} / 2$ is also included, to account for the effect of the anomaly of $U(1)_A$ with respect to QCD. Such treatment yields for example reasonable leading-order approximations for $\eta, \eta' \to \gamma\gamma$. Why is such an extra mass term not considered here?

4] In the discussion of the chiral Lagrangian with spin-1 resonances (section 3.3), I think there may be an issue with parameter normalizations. Equation (3.22) and those below it have identical form to those in [41], but there $F_\pi = \sqrt{2} f_\pi^{\rm here}\approx 132$ MeV. In addition, I believe there $g$ is also different from $g_{\rho \pi\pi}^{\rm here}$. I noticed this because plugging the KSRF relation into Eq. (3.23) does not seem to give $Z^2 = 1/2$ as it should. In the definition of $\rho_{\mu\nu}$ after equation (3.20), the sign convention for the gauge coupling seems to be opposite compared to the choice made in section 2.

5] Some suggestions about notation. a) I think it would be useful to give explicitly, in section 2, the general form $M = \mathrm{diag}\,(m_u, m_d, m_u, m_d)$ which is currently defined only implicitly. b) I found equation (3.5) slightly confusing, in the sense that it seems $\mathcal{L}_{\rm mass}$ is defined as $+(\mu^3/2) ( \mathrm{Tr}[ M \Sigma] + \mathrm{h.c.} )$, i.e. with the opposite sign compared to the preceding equation. Perhaps in Eq. (3.5) the LHS could just be removed? c) In equation (3.21), I suggest to replace $[ ... ]$ with $( ... )$. d) In Eq. (4.9) it is presumably assumed that $\mathcal{Q}$ is diagonal (in general, the last instance would be $\mathcal{Q}^T$). It would be helpful to state it explicitly.

6] In the last paragraph of page 25 it is stated that the choices $m_\rho/m_\pi \approx 1.15, 1.25, 1.4$ 'are suggested by existing phenomenological investigations of such theories as dark matter candidates [33]'. However, in Ref. [33] the chosen benchmarks ranged from $1.6$ to $1.9$. Later, on page 30, it is mentioned that ensembles with $m_\rho / m_\pi \gg 1.4$ are not available for computational limitations. This is of course very reasonable, but I think it would help the reader to point it out earlier, already on page 25.

7] On page 8, lines 4 to 7, one reads 'Third, we explore the various options for gauging ...', but isn't this done in section 4 actually, rather than in section 3?

8] The paper contains a rather large number of typos. I mention here those that I caught, and recommend the authors perform a further check. Pages 2, 11, 'Golstone' or 'Golstones'; page 3, 'are in the pseudoreal' $\to$ 'are in a pseudoreal'; page 7, 'all members of meson multiplet' $\to$ 'all members of a meson multiplet'; same page, 'see Tab. 2.3' presumably should be 'see Tab. 1'; page 9, 'the a five-dimensional'; page 12, 'iso-siglet'; page 13, 'heaver'; page 19, 'withh'; page 20, 'Comparison with (4.14)' $\to$ 'Comparison with (4.15)'?; page 24, 'axialcurrents'; page 25, 'remaing', 'degenaracy', 'predctions'; page 26, 'chosing'; page 27, 'mutliplicative'; same page, GMOR has not been defined yet; page 28, 'the the'; page 32, 'Note that for with'; page 33, 'unambgious'; page 36, 'off-diagional'; page 37, 'given in by'; page 42, 'projecting to zero momentum and study' $\to$ 'projecting to zero momentum and studying'; page 43, 'efffects', 'the the'; page 45, should it be '$m(\rho^M)/m_{\pi}^{\rm deg} > 2$' ? Also, 'wheres'; page 48, 'they are can be'.

In summary, I am of the opinion that once these aspects are addressed, the paper will be acceptable for publication in SciPost Physics.

---

## Round 2 · Referee Report · Anonymous (Referee 1) · 2022-9-22

Report

I am satisfied with the changes made to the text by the authors. I think that in its current form this work meets the criteria of the SciPost and can be published.

---

## Round 2 · Referee Report · Ennio Salvioni (Referee 2) · 2022-10-16

Report

In the revised v2, the authors have carefully addressed all the comments contained in my previous report. They have even gone beyond this, by adding the new section 5.4 (and associated appendix C.3) where the $\eta^\prime$ in the mass non-degenerate case is discussed. As a result, I recommend publication in SciPost Physics.

Requested changes

A few typos remain, which can be fixed when preparing the final version:

  • In the newly added footnote 6, I wonder if $f_{\eta'}$ should appear instead of $f_\pi$ in the expression of $\det \Sigma$ (or perhaps here the redefinition $\eta' \to (f_{\eta'}/f_\pi) f_\pi$ has already been made?).
  • In the newly added eq. (5.20), the second term on the RHS is missing the appropriate derivatives.
  • Just below eq. (5.26), $\Delta m^2_\eta \to \Delta m^2_{\eta'}\;$.
  • Language typos: page 9, "Stoke's " $\to$ ``Stokes' "; page 17, third-to-last line: "exploration of for"; page 27, "fermions masses" $\to$ "fermion masses"; page 36, second line: "control then previously".

---

## Round 2 · Author Response

Dear Editor,

we thank both referees for preparing extensive reports which helped to improve the manuscript and iron out some shortcomings and typos. We address all comments by the referees and hope that the revised version is now ready to be published.

---

## Round 2 · List of Changes

Warnings issued while processing user-supplied markup:

  • Inconsistency: Markdown and reStructuredText syntaxes are mixed. Markdown will be used.
    Add "#coerce:reST" or "#coerce:plain" as the first line of your text to force reStructuredText or no markup.
    You may also contact the helpdesk if the formatting is incorrect and you are unable to edit your text.

Reply to Referee 1:

1.At the same time, the “vector portal” phenomenology is long and well studied in a bulk of papers.

Answer: We agree that vector portal or dark photon phenomenology in connection with SIMPs has been studied before, and in a non-SIMP context indeed most extensively. Our aim is to develop a self-consistent treatment of a non-SU(N) dark matter theory including low energy degrees of freedom beyond the Goldstone bosons. The definitions and the charge assignments when we introduce the vector portal in this context are important for self-consistency and notational uniformity of our framework. We now better refer to existing literature to make it clear that aspects of vector portal phenomenology we discuss have also been discussed elsewhere.

More concretely, we are studying for the first time the low-energy Lagrangian of a non-SU(N) theory from first principles, using lattice calculations at NLO and constrain the actual parameters of the vector portal in this context ($m_V$, $\kappa$, $m_\rho$), thus reducing the 3-parameter space to a 2-parameter space (see Fig. 4). We thereby demonstrate that a dark SU(3) and a dark Sp(4) theory yield similar low-energy results for the vector portal.

2.The effective theory, which is discussed in the first five sections, is standard for this science. Apart from taking into account the effects of explicit symmetry breaking by the splitting of quark masses, nothing new has been done here. Exactly the same nonlinear chiral effective theory was built before. (Pseudo)vector mesons were taken into account in the same way in the framework of this scheme; pseudo-goldstones and eta meson - even more so. The classification of states is common for any implementation of symmetry. Accounting for the anomaly through the WZW interaction was done. The new vector field also does not represent a big innovation, since technically, for the effective Lagrangian there is no difference what kind of abelian field is there - new or the same as that already exists in the SM.

Answer: For the mass-degenerate case, most of the presented results are standard, as we indeed cite many times over, not least within the framework of [6-8], on which we heavily rely. However, the mass non-degenerate case is new and has not been investigated before, especially in conjunction with lattice investigations that clarifies the reach of validity in this case. Moreover, the charge textures and their implications have also not been studied in similar detail before, nor its constraints using lattice results.

Of course, the construction of the chiral Lagrangian is general and follows standard procedures found on many occasions in the literature. We do not claim that we present fundamentally new aspects in this respect. Our aim is to present a self-contained and complete construction of the low energy effective theory. To study the mass non-degnerate case---which the referee agrees is novel---we first have to lay out the degenerate case. We highlight our aims both in introduction and conclusions, as well as within the respective sections.

Regarding the WZW-term, we refer the referee to the next answer.

3.Although the authors refer to work [7], but also in [11] the WZW-term was already considered, as in other papers, so it should be noted the obvious negligence of the authors in providing links and discussing results in similar scenarios obtained by other authors.

Answer: On multiple occasions we refer to the works of Hochberg et al., previous Refs. [11,12,19]. With regard to the WZW-term we explicitly refer to these works in the introduction. Upon better review of the literature, however, we identified additional works on the gauged WZW term which are now cited. We want to point out that effective Sp(4) theories and gauged WZW theories have been considered separately many times in the literature. However, we are unaware that gauged WZW terms (including rho's and dark photons) were studied for Sp(4) theories before.

4.Similarly, the lack of completeness of the reference list is also evident when reading peppy, but somewhat premature statements about future research on phenomenology in scenarios of this type, despite the fact that specific predictions of dark matter effects have long been studied in the Sp(4) extension of the SM. However, there are no references to these works (except for review [32]) in the article.

Answer: Upon a better literature review, we have now added a number of works to our reference list. We hope that this presents now a fair cross section of pre-existing original works. If we are still missing additional relevant works, we are of course happy to correct ourselves.

Finally, we believe that a particular strength of our framework is the tailored combination with lattice calculations. This is what we meant with preparing the grounds for phenomenological exploration in "future research". We have improved our conclusions to highlight this as the special approach taken here.

I hope that the authors will be able to conduct a more detailed and accurate analysis of the results obtained by other authors and expand the list of references. They will also be able to indicate, at least in advance, which physical effects and manifestations of the dark matter candidates they propose can be observed in direct experiments or indirect data in various energy regions. It would also be interesting to see the effects of mixing new composite scalars with the Higgs taken into account within this scheme. The article as a whole satisfies the criteria for publication in this journal and can be published after the specified additions and revision of the text.

Answer: As discussed above, we have substantially expanded the list of references, better detailing previous original contributions. It is not the purpose of the paper to go into phenomenological exploration of the kind the referee suggests. Rather, coupling lattice with the chiral construction will allow to do this in the future.

Finally, studying Higgs-portal phenomenology is orthogonal to what we have done in the present paper. This deserves a dedicated study and goes well beyond the scope of this work. We nevertheless acknowledge that these are additional avenues one can take and comment on the possibility of Higgs-portal couplings in the preface to Section 4, footnote 7.

As already noted, it is recommended to carry out a more complete analysis of the results obtained earlier 1. both in the study of the mathematical structure of symplectic extensions with a vector portal 2. and in the discussion of the phenomenological manifestations of dark matter and the possibility of their direct or indirect detection. So, the list of references should be extended. 3. Some preliminary estimations of possible phenomenological manifestations of the DM candidates are also desirable, as is the analysis of scalar-Higgs boson mixing.

Answer: The last comment by the referee summarizes their critical points stated earlier so that we refer to the respective answers that we have provided above.

Reply to Referee 2

1] One question with an eye to SIMP phenomenology. One of the main results of this work is the detailed study of the $m_{u} \neq m_{d}$ scenario, where the flavor symmetry is explicitly broken to $S U(2){u} \times S U(2)$, under which the diagonal Goldstone $\pi^{C}$ transforms as a singlet. The $\pi^{C}$ is also the lightest meson, as shown in section 6 . From the discussion in Sec. 4 (page 17 ) I understand that $\pi^{C}$ is always unstable once a dark $U(1)^{\prime}$ is gauged, regardless of the choice of the charge matrix $\mathcal{Q}$. If this is correct, then rendering such scenario compatible with cosmological/astrophysical constraints would seem to be an important challenge, deserving to be at least mentioned here. Do the authors agree with my reading?

Answer: We agree with the reviewer's reading. Once the diagonal Goldstone $\pi^C$ transforms as a singlet it will be challenging to make it sufficiently long-lived once any connection is made to the observable sector. We have now highlighted this in the main text.

2] A few comments regarding references. a) On the last line of page 1, [22] is cited for 'including dark vector resonances into the SIMP paradigm', however it seems to me that [33] deserves to be included here as well. b) On page 9, when introducing the $\pi^{5}$ interaction only the Witten paper [39] is cited, but I think it would be fair to also include the classic work by Wess and Zumino. c) While the authors have in mind the application to (SIMP) dark matter, in the broader context of the Introduction it may be worthwhile to reference also previous papers that studied the $S U(4) / S p(4)$ coset in the context of electroweak symmetry breaking, such as $0902.1483$ by Gripaios et al. and $1001.1361$ by Galloway et al. This last point is merely a suggestion, which the authors are free to accept or decline.

Answer: a) We now added references to (previous reference [33], along with [22]), which currently read as reference [33, 35]. b) We thank the referee for this suggestion. We now also cite the classic works by Wess and Zumino, which are now references [68,69] along with reference [67]. c) We once again thank referee for this suggestion. Along the lines of suggestions also by referee 1, we now have substantially extended the overall reference list. This also includes $0902.1483$ by Gripaios et al. and $1001.1361$ by Galloway et al. along with other references on the $SU(4)/Sp(4)$ coset works.

3] I was somewhat confused by the treatment of the pseudoscalar singlet $\eta$ in section 3.2. In the QCD case one can add the $\eta^\prime$ along similar lines, by extending ChPT to $U(3){L} \times U(3)$, however a 'hard breaking' mass term $\delta \mathcal{L}=-m_{0}^{2} \eta^{2} / 2$ is also included, to account for the effect of the anomaly of $U(1)_{A}$ with respect to QCD. Such treatment yields for example reasonable leading-order approximations for $\eta, \eta^{\prime} \rightarrow \gamma \gamma$. Why is such an extra mass term not considered here?

Answer: Since we are working in the two-flavor theory, we only have the state $\eta^\prime$ as a consequence of the $U(1)_A$ anomaly. The state $\eta$ does not exist in this case. In order to avoid further confusion we have changed our notation from $\eta$ to $\eta'$. Indeed, the pseudoscalar singlet receives contributions from the $U(1)_A$ anomaly and not from the construction of the chiral Lagrangian itself. Hence, we have added an explicit mass term and decay constant for the $\eta'$ to account for that.

4] In the discussion of the chiral Lagrangian with spin-1 resonances (section 3.3), I think there may be an issue with parameter normalizations. Equation (3.22) and those below it have identical form to those in [41], but there $F_{\pi}=\sqrt{2} f_{\pi}^{\text {here }} \approx 132 \mathrm{MeV}$. In addition, I believe there $g$ is also different from $g_{\rho \pi \pi}^{\text {here }}$. I noticed this because plugging the KSRF relation into Eq. (3.23) does not seem to give $Z^{2}=1 / 2$ as it should. In the definition of $\rho_{\mu \nu}$ after equation (3.20), the sign convention for the gauge coupling seems to be opposite compared to the choice made in section $2 .$

Answer: We agree with the referee's assertions and thank them for their careful eye. We have unified our definitions and ensured that they are now consistent with the rest of the paper. We have also renamed our parameter $g_{\rho \pi \pi}$ to $g_\rho$ to avoid confusion with definitions in the literature.

5] Some suggestions about notation. a) I think it would be useful to give explicitly, in section 2 , the general form $M=\operatorname{diag}\left(m_{u}, m_{d}, m_{u}, m_{d}\right)$ which is currently defined only implicitly. b) I found equation (3.5) slightly confusing, in the sense that it seems $\mathcal{L}_{\text {mass }}$ is defined as $+\left(\mu^{3} / 2\right)(\operatorname{Tr}[M \Sigma]+$ h. c.), i.e. with the opposite sign compared to the preceding equation. Perhaps in Eq. (3.5) the $L H S$ could just be removed? c) In equation (3.21), I suggest to replace $[\ldots]$ with $(\ldots)$. d) In Eq. (4.9) it is presumably assumed that $\mathcal{Q}$ is diagonal (in general, the last instance would be $\mathcal{Q}^{T}$ ). It would be helpful to state it explicitly.

Answer: Again, those are most helpful comments, which we address in the following way: a) The diagonal mass matrix is now given explicitly in section 2. b) The LHS was removed. In order to obtain the vacuum we minimize the potential term $V$, which is defined with an opposite sign in the Lagrangian. c) As the referee suggested, we replace in Eq. (3.21) the brackets. d) We changed the Eq. (4.9) such that a general charge assignment is used.

6] In the last paragraph of page 25 it is stated that the choices $m_{\rho} / m_{\pi} \approx 1.15,1.25,1.4$ 'are suggested by existing phenomenological investigations of such theories as dark matter candidates [33]'. However, in Ref. [33] the chosen benchmarks ranged from $1.6$ to $1.9$. Later, on page 30 , it is mentioned that ensembles with $m_{\rho} / m_{\pi} \gg 1.4$ are not available for computational limitations. This is of course very reasonable, but I think it would help the reader to point it out earlier, already on page $25 .$

Answer: We now point out computational limitation already at the end of section 6.1.

7] On page 8 , lines 4 to 7 , one reads 'Third, we explore the various options for gauging ...', but isn't this done in section 4 actually, rather than in section 3 ? Answer: The reviewer is correct and corresponding section label has been changed.

8] The paper contains a rather large number of typos. I mention here those that I caught, and recommend the authors perform a further check. Pages 2,11 , 'Golstone' or 'Golstones'; page 3, 'are in the pseudoreal' $\rightarrow$ 'are in a pseudoreal'; page 7 , 'all members of meson multiplet' $\rightarrow$ 'all members of a meson multiplet'; same page, 'see Tab. $2.3$ ' presumably should be 'see Tab. 1 '; page 9 , 'the a fivedimensional'; page 12, 'iso-siglet'; page 13, 'heaver'; page 19, 'withh'; page 20, 'Comparison with (4.14)' $\rightarrow$ 'Comparison with (4.15)'?; page 24, 'axialcurrents'; page 25, 'remaing', 'degenaracy', 'predctions'; page 26, 'chosing'; page 27, 'mutliplicative'; same page, GMOR has not been defined yet; page 28 , 'the the'; page 32 , 'Note that for with'; page 33 , 'unambgious'; page 36 , 'off-diagional'; page 37, 'given in by'; page 42 , 'projecting to zero momentum and study' $\rightarrow$ 'projecting to zero momentum and studying'; page 43 , 'efffects', 'the the'; page 45 , should it be ' $m\left(\rho^{M}\right) / m_{\pi}^{\text {deg }}>2$ ' ? Also, 'wheres'; page 48, 'they are can be'.

Answer: We thank the referee for spotting those typos, which we corrected.

Additional major changes to the manuscript:

In an effort to better capitalize on our original results concerning the role of $\eta'$ we have introduced new Sec. 5.4 where we study $\eta'$ in the mass non-degenerate case including the WZW interactions. Finally, we point out that we changed the overall normalization of the kinetic term in the ansatz for the chiral Lagrangian in the non-degenerate case (5.5) and modified all dependent equations accordingly.

---

## Editorial Decision

published